



# Seasonality of the particle number concentration and size distribution: a global analysis retrieved from the network of Global Atmosphere Watch (GAW) near-surface observatories

Clémence Rose[1], Martine Collaud Coen[2], Elisabeth Andrews[3,4], Yong Lin[5], Isaline Bossert[1,6], Cathrine Lund Myhre[5], Thomas Tuch[7], Alfred Wiedensohler[7], Markus Fiebig[5], Pasi Aalto[8], Andrés Alastuey[9], Elisabeth Alonso-Blanco[10], Marcos Andrade[11], Begoña Artíñano[10], Todor Arsov[12], Urs Baltensperger[13], Susanne Bastian[14], Olaf Bath[15], Johan Paul Beukes[16], Benjamin T. Brem[13], Nicolas Bukowiecki[13,*], Juan Andrés Casquero-Vera[17,18], Sébastien Conil[19], Konstantinos Eleftheriadis[20], Olivier Favez[21], Harald Flentje[22], Maria I. Gini[20], Francisco Javier Gómez-Moreno[10], Martin Gysel-Beer[13], A. Gannet Hallar[23], Ivo Kalapov[12], Nikos Kalivitis[24], Anne Kasper-Giebl[25], Melita Keywood[26], Jeong Eun Kim[27], Sang-Woo Kim[28], Adam Kristensson[29], Markku Kulmala[8], Heikki Lihavainen[30,31], Neng-Huei Lin[32,33], Hassan Lyamani[17,18], Angela Marinoni[34], Sebastiao Martins Dos Santos[35], Olga L. Mayol-Bracero[36], Frank Meinhardt[15], Maik Merkel[7], Jean-Marc Metzger[37], Nikolaos Mihalopoulos[24,38], Jakub Ondracek[39], Marco Pandolfi[9], Noemi Pérez[9], Tuukka Petäjä[8], Jean-Eudes Petit[40], David Picard[1], Jean-Marc Pichon[1], Veronique Pont[41], Jean-Philippe Putaud[35], Fabienne Reisen[26], Karine Sellegri[1], Sangeeta Sharma[42], Gerhard Schauer[43], Patrick Sheridan[4], James Patrick Sherman[44], Andreas Schwerin[15], Ralf Sohmer[15], Mar Sorribas[45], Junying Sun[46], Pierre Tulet[47], Ville Vakkari[16,30], Pieter Gideon van Zyl[16], Fernando Velarde[11], Paolo Villani[48], Stergios Vratolis[20], Zdenek Wagner[39], Sheng-Hsiang Wang[32], Kay Weinhold[7], Rolf Weller[49], Margarita Yela[45], Vladimir Zdimal[39] and Paolo Laj[50,34,8]

[1]Université Clermont Auvergne, CNRS, Laboratoire de Météorologie Physique (LaMP), F-63000 Clermont-Ferrand, France.
[2] Federal Office of Meteorology and Climatology, MeteoSwiss, Payerne, Switzerland
[3]Cooperative Institute for Research in Environmental Sciences, University of Colorado, Boulder, CO, USA
[4]NOAA Global Monitoring Laboratory, Boulder, CO, USA
[5]NILU-Norwegian Institute for Air Research, Kjeller, Norway
[6]Université Bourgogne Franche Comté, Besançon, France
[7]Leibniz Institute for Tropospheric Research, Leipzig, Germany
[8]Institute for Atmospheric and Earth System Research, University of Helsinki, Helsinki, Finland
[9]Institute of Environmental Assessment and Water Research (IDAEA), Spanish Research Council (CSIC), Barcelona, Spain
[10]CIEMAT, Center for Research on Energy, Environment and Technology, Joint Research Unit CSIC-CIEMAT, Madrid, Spain
[11]Laboratorio de Fisica de la Atmosfera, Universidad Mayor de San Andres, La Paz, Bolivia
[12]Institute for Nuclear Research and Nuclear Energy, Bulgarian Academy of Sciences, Sofia, Bulgaria
[13]Laboratory of Atmospheric Chemistry, Paul Scherrer Institute, Villigen PSI, Switzerland
[14]Saxon State Office for Environment, Agriculture and Geology (LfULG), Dresden, Germany
[15]German Environment Agency (UBA), Zugspitze, Germany
[16]Atmospheric Chemistry Research Group, Chemical Resource Beneficiation, North-West University, Potchefstroom, 2520, South Africa
[17]Department of Applied Physics, University of Granada, Granada, Spain
[18]Andalusian Institute for Earth System Research (IISTA-CEAMA), University of Granada, Autonomous Government of Andalusia, Granada, Spain
[19]ANDRA DRD/GES Observatoire Pérenne de l'Environnement, 55290 Bure, France
[20]ERL, Institute of Nuclear and Radiological Science & Technology, Energy & Safety N.C.S.R. "Demokritos", Attiki, Greece



[21]Institut National de l'Environnement Industriel et des Risques (INERIS), Verneuil-en-Halatte, France
[22]German Weather Service, Meteorological Observatory Hohenpeissenberg, Hohenpeißenberg, Germany
[23]Department of Atmospheric Sciences, University of Utah, Salt Lake City, UT 84112, USA
[24]Environmental Chemical Processes Laboratory (ECPL), University of Crete, Heraklion, Crete, 71003, Greece
[25]TU Wien - Institute of Chemical Technlogies and Analytics, Vienna, Austria
[26]CSIRO Oceans and Atmosphere, PMB1 Aspendale, VIC, Australia
[27]Global Atmosphere Watch Team, Innovative Meteorological Research Department, National Institute of Meteorological Sciences, Seogwipo-si, Jeju-do, Korea
[28]School of Earth and Environmental Sciences, Seoul National University, Seoul, Korea
[29]Lund University, Department of Physics, Division of Nuclear Physics, Lund, Sweden
[30]Atmospheric composition research, Finnish Meteorological Institute, Helsinki, Finland
[31]Svalbard Integrated Arctic Earth Observing System, Longyearbyen, Svalbard, Norway
[32]Department of Atmospheric Sciences, National Central University, Taoyuan, Taiwan
[33]Center for Environmental Monitoring Technology, National Central University, Taoyuan, Taiwan
[34]Institute of Atmospheric Sciences and Climate, National Research Council of Italy, Bologna, Italy
[35]European Commission, Joint Research Centre (JRC), Ispra, Italy
[36]University of Puerto Rico, Rio Piedras Campus, San Juan, Puerto Rico
[37]Observatoire des Sciences de l'Univers de La Réunion (OSUR), UMS3365, Saint-Denis de la Réunion, France
[38]Institute of Environmental Research & Sustainable Development, National Observatory of Athens, Palea Penteli, 15236, Greece
[39]Department of Aerosol Chemistry and Physics, Institute of Chemical Process Fundamentals, CAS, Prague, Czech Republic
[40]Laboratoire des Sciences du Climat et de l'Environnement, LSCE/IPSL, UMR 8212 CEA-CNRS-UVSQ, Université Paris-Saclay, Gif-sur-Yvette, France
[41]Laboratoire d'Aérologie, CNRS-Université de Toulouse, CNRS, UPS, Toulouse, France
[42]Environment and Climate Change Canada, Toronto, ON, Canada
[43]ZAMG – Sonnblick Observatory, 5020 Salzburg, Austria
[44]Department of Physics and Astronomy, Appalachian State University, Boone, NC, USA
[45]Atmospheric Sounding Station, El Arenosillo, Atmospheric Research and Instrumentation Branch, INTA, 21130, Mazagón, Huelva, Spain
[46]State Key Laboratory of Severe Weather & Key Laboratory of Atmospheric Chemistry of CMA, Chinese Academy of Meteorological Sciences, Beijing 100081, China
[47]Laboratoire de l'Atmosphère et des Cyclones (LACy), UMR8105, Université de la Réunion – CNRS – Météo-France, Saint-Denis de La Réunion, France
[48]4S Company, 63000 Clermont Ferrand, France
[49]Alfred-Wegener-Institut, Helmholtz-Zentrum für Polar- und Meeresforschung, 27570 Bremerhaven, Germany
[50]Univ. Grenoble-Alpes, CNRS, IRD, Grenoble-INP, IGE, 38000 Grenoble, France
*now at University of Basel, Department of Environmental Sciences, Basel, Switzerland

*Correspondence to*: c.rose@opgc.fr

## Abstract

Aerosol particles are a complex component of the atmospheric system that influences climate directly by interacting with solar radiation, and indirectly by contributing to cloud formation. The variety of their sources, as well as the multiple transformations they may undergo during their transport, result in significant spatial and temporal variability of their properties. Documenting this variability is essential to provide a proper representation of aerosols and cloud condensation nuclei (CCN) in climate





models. Using measurements conducted in 2016 or 2017 at 62 ground based stations around the world, this study provides the most up-to-date picture of the spatial distribution of particle number concentration (Ntot) and number size distribution (PNSD, from 39 sites). A sensitivity study was first performed to assess the impact of data availability on Ntot's annual and seasonal statistics, as well as on the analysis of its diel cycle. Thresholds of 50% and 60% were set at the seasonal and annual scale, respectively, for the study of the corresponding statistics, and a slightly higher coverage (75%) was required to document the diel cycle.

Although some observations are common to a majority of sites, the variety of environments characterizing these stations made it possible to highlight contrasting findings, which, among other factors, seem to be significantly related to the level of anthropogenic influence. The concentrations measured at polar sites are the lowest (~$10^2$ cm$^{-3}$) and show a clear seasonality, which is also visible in the shape of the PNSD, while diel cycles are in general barely marked, due notably to the absence of a regular day-night cycle in some seasons. In contrast, the concentrations characteristic of urban environments are the highest (~$10^3$-$10^4$ cm$^{-3}$) and do not show pronounced seasonal variations, whereas diel cycles tend to be very regular over the year at these stations. The remaining sites, including mountain and non-urban continental and coastal stations, do not exhibit as obvious common behaviour as polar and urban sites and display, on average, intermediate $N_{tot}$ (~$10^2$-$10^3$ cm$^{-3}$). Particle concentrations measured at mountain sites, however, are generally lower compared to nearby lowland sites, and tend to exhibit somewhat more pronounced seasonal variations as a likely result of the strong impact of the atmospheric boundary layer (ABL) influence in connection with the topography of the sites. ABL dynamics also likely contribute to the diel cycle of $N_{tot}$ observed at these stations. Based on available PNSD measurements, CCN-sized particles (i.e. > 50 – 100 nm) can represent from a few percent to almost all of $N_{tot}$, corresponding to seasonal medians in the order of ~10 to 1000 cm$^{-3}$, with seasonal patterns and a hierarchy of the site types broadly similar to those observed for $N_{tot}$.

Overall, this work illustrates the importance of in-situ measurements, in particular for the study of aerosol physical properties, and thus strongly supports the development of a broad global network of near surface observatories to increase and homogenize the spatial coverage of the measurements, and guarantee as well data availability and quality. The results of this study also provide a valuable, freely available and easy to use support for model comparison and validation, with the ultimate goal of contributing to improvement of the representation of aerosol-cloud interactions in models, and, therefore, of the evaluation of the impact of aerosol particles on climate.

## 1. **Introduction**

Atmospheric aerosol particles are an essential component of the climate system. They affect the Earth's radiation balance directly by interacting with solar radiation, and indirectly by contributing to cloud formation. These effects, and in particular the latter, are widely recognized as one of the largest sources of uncertainty in climate change projections (IPCC, 2013), further reflecting the difficulty of obtaining an accurate representation of aerosols and cloud condensation nuclei (CCN, i.e. one of the critical elements in the evaluation of cloud aerosol interactions) in climate models. In addition to the large diversity of their





sources (primary or secondary, natural or anthropogenic), particles undergo transformations that lead to changes in their properties during transport. Also, in contrast with greenhouse gases, they have a short lifetime, which results in a highly heterogeneous distribution in space and time. Providing reliable observations of aerosol properties at appropriate spatial and temporal scales is therefore essential, and requires combined approaches adapted to the diversity of these scales and the

information they can provide for climate studies. Satellite observations can document extensive aerosol properties with significant geographic coverage, but they have only limited temporal resolution and are only partially adapted to the study of some aerosol properties such as the size distribution. Also, due to atmospheric boundary layer (ABL) structure segregation of vertical air masses and evolution of such structures on a daily basis (e.g. Gierens et al., 2019), it is currently very difficult to attribute aerosol properties measured with satellite observations to defined depths in the ABL. In contrast, in-situ measurements

performed at ground-level stations are often representative of limited geographical areas and do not allow assessment of vertical variability, but they do allow a more detailed characterization of the aerosol, at a fine temporal resolution.

The Geophysical Monitoring for Climate Change (GMCC) program, established by NOAA in the early 1970's, was the first network dedicated to long-term measurements of climate-relevant aerosol properties. The particle number concentration, considered to be a primary indicator of human impact on atmospheric composition, was the first aerosol property measured at

the GMCC stations (e.g. Bodhaine, 1983). Since then, the number of measured properties has increased and measurement of the particle number size distribution (PNSD) is now quite common. In comparison to the total number concentration alone, the knowledge of the PNSD offers additional information on particle formation processes, transport and type, and, more broadly, on their potential climatic impact. As well summarized by Asmi and coworkers (2013), the effect particles may have on climate is indeed not necessarily proportional to their total number concentration. This effect is, in fact, highly variable

across the particle size spectrum, as both the potential of aerosols to act as CCN and their ability to efficiently scatter or absorb light depends not only on their chemical composition but on their size as well. Among other examples, the importance of measuring the PNSD over long enough time periods in contrasting environments is also well illustrated in the more recent study by Schmale et al. (2018) for the understanding of aerosol-cloud interactions and, ultimately, the improvement of their representation in models. Finally, as a clear sign of its value, the PNSD was recently proposed as an aerosol essential climate

variable (ECV) for climate monitoring in the Global Climate Observing System (GCOS, https://gcos.wmo.int/en/networks). In addition, while these aspects are behind the scope of the present study, the knowledge of the particle size is also essential to assess the effects aerosols may have on human health, as the size constrains in particular the ability of the particles to enter the respiratory system. The health effect of ultrafine particles (<100 nm) is for instance discussed and compared to that of fine (<2.5µm) and larger (<10µm) particles in the recent review by Schraufnagel (2020).

In order to meet the need to document as broad a variety of conditions as possible, the number of stations for systematic monitoring of aerosols has also increased over the past 50 years. Although some sites remain independent, at present measurements are mainly organized within networks that ensure the homogeneity of protocols used for data acquisition, quality control and provision, and also promote the continuity of the measurements. The GAW (Global Atmospheric Watch) aerosol network, initiated in 1997 under the leadership of the GAW Scientific Advisory Group (SAG) for aerosols, brings together a





significant number of sites, which at the same time belong to regional networks such as ACTRIS (Aerosols, Clouds and Trace gases Research Infrastructure, https://www.actris.eu/) or the NOAA Federated Aerosol Network (NOAA-FAN) (Andrews et al., 2019). Although there is still a bias in the world data coverage, the growing number of sites has made it possible to study the spatial variability of aerosol properties and/or their long-term evolution at regional and even global scale.

Taking advantage of the existing monitoring networks (and/or research projects), seven companion studies dedicated to aerosol phenomenology have been conducted in Europe since 2004 (Van Dingenen et al., 2004; Putaud et al., 2004; Putaud et al., 2010; Cavalli et al., 2016; Zanatta et al., 2016; Pandolfi et al., 2018; Bressi et al., in review). Up to 60 sites have contributed to this project involving observations of physical, optical and chemical aerosol properties. In parallel, Asmi et al. (2011) reported on the variability of the PNSD, also in Europe, based on measurements collected at 24 sites, and, shortly after, the

first long-term trend analyses of aerosol optical properties, number concentration and PNSD were performed (Asmi et al., 2013; Collaud Coen et al., 2013). The characteristics of specific processes such as new particle formation (NPF), which is thought to be responsible for a major fraction of the particle number at the global scale (Spracklen et al., 2006, 2008; Merikanto et al., 2009; Gordon et al., 2017), could also be investigated and compared in various environments (Kerminen et al, 2018; Nieminen et al., 2018). Analyses dedicated to specific environments were also carried out. As an example, Sellegri et al.

(2019), Andrews et al. (2011) and Collaud Coen et al. (2018) all concentrated on measurements performed at mountain sites, and focussed respectively on NPF, on aerosol optical properties and on the influence of the ABL on the observations made at these high altitude sites. The monitoring of an increasing number of variables finally made it possible to explore the link between the different properties of the particles and to carry out closure studies, such as that performed by Schmale et al. (2017, 2018) using long-term measurements of CCN number concentrations, particle number size distributions and chemical

composition from 12 ACTRIS sites.

The present work is part of the SARGAN (in-Situ AeRosol GAW observing Network) initiative, which has been introduced in Laj et al. (2020) and aims at supporting a global aerosol monitoring network to become a GCOS associated network. The most complete and up-to-date analysis of the trends and variability of aerosol optical properties measured worldwide was recently reported within the framework of this project (Collaud Coen et al., 2020). Two other studies involving observations

and outputs from the AeroCom models (Aerosol Comparisons between Observations and Models, https://aerocom.met.no/) were also carried out: Gliβ et al. (2020) assessed the ability of global models to reproduce present day aerosol optical properties and Mortier et al. (2020) performed a multi-parameter analysis of the trends of optical, chemical-composition and mass aerosol properties over the last 2 decades.

A preliminary view of the variability of the particle number concentration was reported in Laj et. al (2020), using measurements

performed at 57 sites in 2016 or 2017. This study was however limited to basic statistics, and also did not include any description of the PNSD. The present work aims to complement the analysis initiated in Laj et al. (2020) in order to 1) provide the most up-to-date information on the spatial and temporal variability of the particle number concentration worldwide and discuss what determines this variability, and 2) extend the analysis to the PNSD. This new study, based on observations collected at 62 sites around the world in 2016 or 2017, also complement the previous work of Asmi et al. (2011), which focused





on measurements collected in 2008-2009 in Europe only. Although the findings of the two studies are naturally compared in this paper, there is, however, no detailed analysis of the changes or differences observed for the sites they have in common, since both studies are based on limited measurement periods (1-2 years) that do not allow the evaluation of possible trends; these aspects will be addressed in a separate paper. The first part of the present paper is dedicated to a sensitivity study aimed

at assessing the impact of data availability on the total particle number concentration annual and seasonal statistics, as well as on the analysis of its diel cycle (Sect. 4). The seasonality of the particle number concentration and PNSD are then investigated (Sect. 5). Finally, two shorter sections are dedicated to the analysis of the diel cycle of the total particle number concentration (Sect. 6), and to the study of the CCN-sized fraction of the aerosol spectrum (Sect. 7).

## 2. Measurement sites and data handling

Data collected at 62 sites contributing to SARGAN in 2016 or 2017 (see more details about data availability and coverage criteria in Sect. 4) were included in the present work, among which 57 were already involved in the short analysis of the total number concentration reported in Laj et al. (2020). As indicated in Table 1 and further illustrated in Fig. 1, the majority of these sites are located in the Northern Hemisphere, with, in particular, 39 stations in Europe and 10 in North America, among which 5 are located above the polar circle. Polar regions are fairly well represented in the Southern Hemisphere as well, with

3 sites in Antarctica, but other parts of the world tend to be underrepresented, with only 2 sites in Africa, 4 in Asia, 1 in South America and 3 in the South-West Pacific. In spite of this inhomogeneous distribution, a multitude of conditions are however represented in the combined dataset. The stations are classified based on the combination of a geographical (continental, coastal, mountain, or polar) and footprint (rural background, forest, (sub)-urban, pristine or mixed) criteria as introduced in Laj et al. (2020). Note that the classification of mountain sites does not solely rely on elevation, but also requires that the

station is located higher than the neighbouring environment. As shown in Fig. 1, the spatial distribution of the sites in relation to their classification again reveals certain limitations. For instance, all urban stations are located in Europe, and there is a clear lack of data from desert areas. A final bias concerns the type of data collected at these sites. Specifically, the stations equipped with mobility particle size spectrometers (MPSS) for the monitoring of the PNSD are mainly located in Europe (34 out of 39 sites), while other sites operate condensation particle counters (CPC), which retrieve measurements of the total particle number

concentration only.

As previously implied, most of the stations listed in Table 1 are regional or global GAW sites (https://gawsis.meteoswiss.ch), and belong to regional (mainly ACTRIS and NOAA-FAN) and/or national networks, such as the German Ultrafine Aerosol Network (GUAN; Birmili et al., 2009), or the Spanish Network of Environmental DMAs (REDMAAS; Gómez-Moreno et al., 2015; Alonso-Blanco et al., 2018). Hourly means of the particle number concentration and/or PNSD are available for all these

sites on the database EBAS (http://ebas.nilu.no), which is managed by the Norwegian Institute for Air Research (NILU) and which hosts the World Data Center for Aerosol (WDCA, http://www.gaw-wdca.org) data repository. The inversion of MPSS data was performed by the institutes operating the instruments before submission to the database, and, for both CPC and MPSS,





particle number concentrations were reported in particles per cubic centimetre at STP (T = 273.15 K and P = 101 325 Pa), following the recommendations from Wiedensohler et al. (2012). As reported in Laj et al. (2020), the diameters associated with MPSS data correspond to the geometric mean mobility diameter of the size intervals used in the inversion. MPSS measurements are usually representative of dry aerosol properties, as the relative humidity of the sampled air is recommended

5  to be kept below 40% (Wiedensohler et al., 2012). To ensure the quality of the analysis, only the data marked as valid were used, similar to Asmi et al. (2011). Additional check was performed in collaboration with each instrument's principal investigator to ensure the homogeneity of the dataset. Specifically, negative concentrations arising from inversion issues in certain conditions (e.g. presence of large particles such as dust or sea salt) were filtered out.

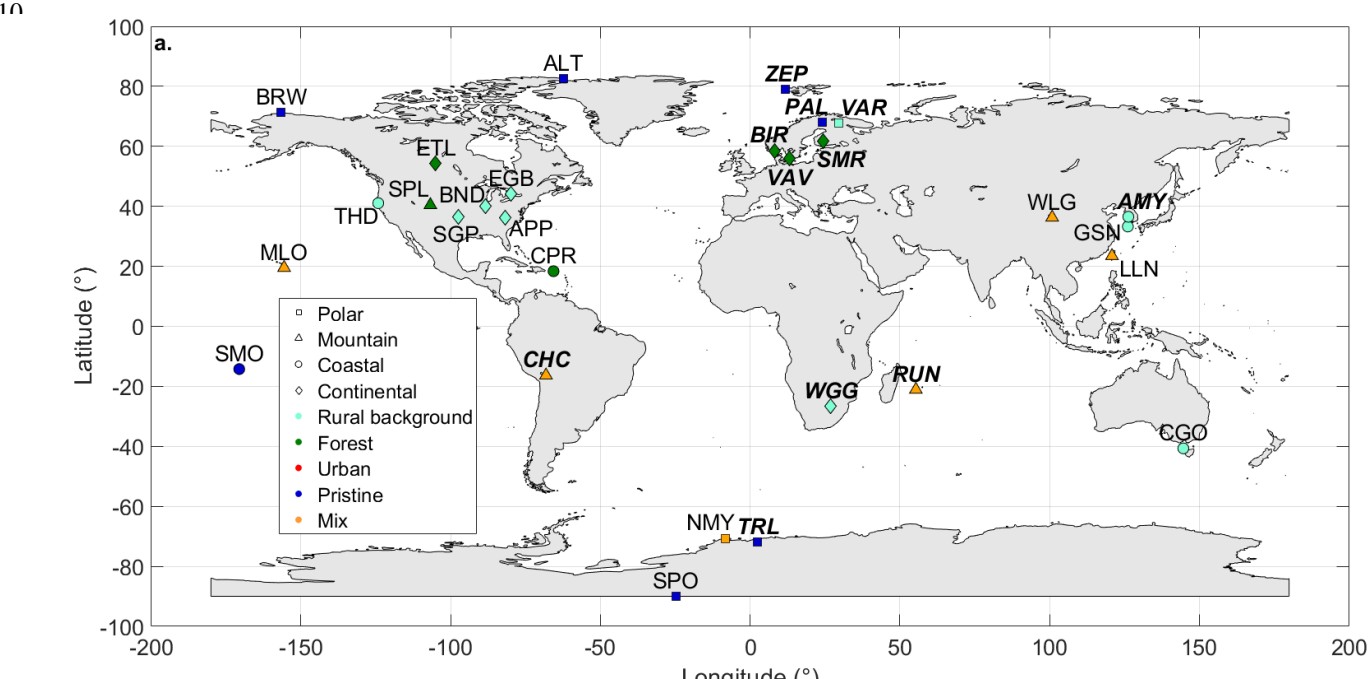



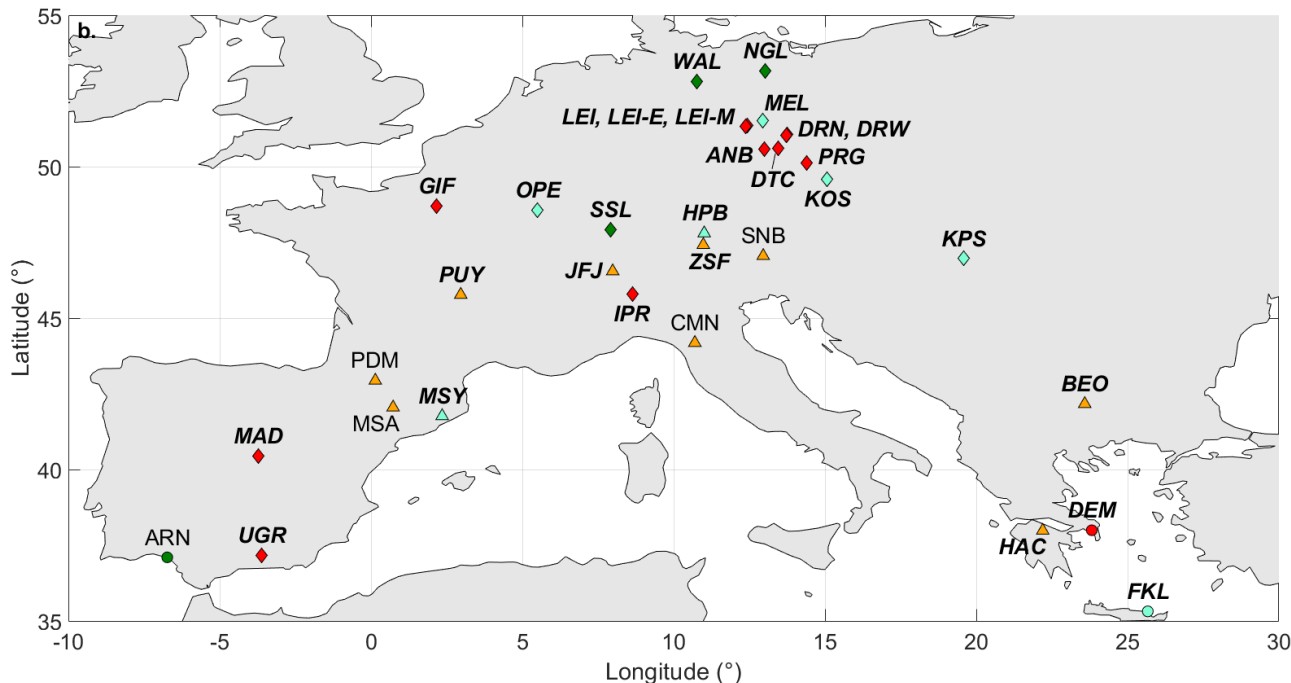

Fig. 1 Geographical distribution of the stations with their acronyms (see Table 1) a. at the global scale and b. specifically over central and Southern Europe. The shape and colour of the markers indicate geographical and footprint categories, respectively. The sites operating a MPSS are additionally marked in italic bold.

Table 1 List of SARGAN stations included in the present study. The geographical (with the following abbreviations: Mt for mountain, P for polar, Con for continental, and Coast for coastal) and footprint (RB for rural background, F for forest, U for



urban, P for pristine, and Mix for mixed) categories are indicated for each site, together with the year considered in the analysis (2016 or 2017), the type of instrument operated at the site (CPC or MPSS) and the corresponding cut-point or diameter range.

| Station name | Acronym | Country | GPS coordinates | Site charact. | Year | Instr. | Lower cut-point / diam. range (nm) |
|---|---|---|---|---|---|---|---|
| WMO I, Africa | | | | | | | |
| La Réunion – Maïdo atmospheric observatory | RUN | FR | 21°4'S, 55°22'E, 2160 m | Mt, Mix | 2017 | MPSS | 10.0 – 600.0 |
| Welgegund | WGG | ZA | 26°34'S, 26°56'E, 1480 m | Con, RB | 2017 | MPSS | 11.8 – 843.9 |
| WMO II, Asia | | | | | | | |
| Anmyeon-do | AMY | KR | 36°32'N, 126°19'E, 46 m | Coast, RB | 2017 | MPSS | 10.6 – 982.2 |
| Gosan | GSN | KR | 33°16'N,126°10'E, 72 m | Coast, RB | 2016 | CPC | 2.5 |
| Lulin | LLN | TW | 23°28'N, 120°52'E, 2862 m | Mt, Mix | 2017 | CPC | 10 |
| Mt. Waliguan | WLG | CN | 36°17'N, 100°54'E, 3810 m | Mt, Mix | 2016 | CPC | 10 |
| WMO III, South America | | | | | | | |
| Mount Chacaltaya | CHC | BO | 16°21'S, 68°8'W, 5240 m | Mt, Mix | 2017 | MPSS | 10.0 – 500.0 |
| WMO IV, North America, Central America and the Caribbean | | | | | | | |
| Alert | ALT | CA | 82°29'N, 62°20'W, 210 m | P, Coast, P | 2017 | CPC | 10 |
| Appalachian State University | APP | US | 36°12'N, 81°42'W, 1100 m | Con, RB | 2017 | CPC | 10 |
| Bondville | BND | US | 40°2'N, 88°22'W, 213 m | Con, RB | 2017 | CPC | 11 |
| Barrow | BRW | US | 71°19'N, 156°36'W, 11 m | P, Coast, P | 2017 | CPC | 10 |
| Cape San Juan | CPR | PR | 18°22'N, 65°37'W, 65 m | Coast, F | 2016 | CPC | 7 |
| Egbert | EGB | CA | 44°13'N,79°47'W, 255 m | Con, RB | 2017 | CPC | 4 |



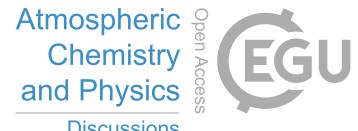

| | | | | | | | |
|---|---|---|---|---|---|---|---|
| East Trout Lake | ETL | CA | 54°21'N, 104°59'W, 500 m | Con, F | 2017 | CPC | 2.5 or 4 |
| Southern Great Plains | SGP | US | 36°36'N, 97°29'W, 318 m | Con, RB | 2016$ | CPC | 10 |
| Storm Peak Laboratory | SPL | US | 40°26'N, 106°44'W, 3220 m | Mt, F | 2016$ | CPC | 10 |
| Trinidad Head | THD | US | 41°3'N, 124°9'W, 107 m | Coast, RB | 2016 | CPC | 11 |
| WMO V, South-West Pacific | | | | | | | |
| Cape Grim | CGO | AU | 40°40'S, 144°41'E, 94 m | Coast, RB | 2017 | CPC | 10 |
| Mauna Loa | MLO | US | 19°32'N, 155°34'W, 3397 m | Mt, Mix | 2017 | CPC | 11 |
| Samoa | SMO | US | 14°14'S, 170°33'W, 77 m | Coast, P | 2016 | CPC | 10 |
| WMO VI, Europe | | | | | | | |
| Annaberg-Buchholz | ANB | DE | 50°34'N, 12°59'E, 545 m | Con, U | 2017 | MPSS | 10.0 – 800.0 |
| El Arenosillo | ARN | ES | 37°6'N, 6°43'W, 41 m | Coast, F | 2017 | CPC | 2.5 |
| Birkenes II | BIR | NO | 58°23'N, 8°15'E, 219 m | Con, F | 2017 | MPSS | 10.0 – 800.0 |
| BEO Moussala | BEO | BG | 42°10'N, 23°34'E, 2971 m | Mt, Mix | 2016 | MPSS | 10.0 – 800.0* |
| Mt Cimone | CMN | IT | 44°10'N, 10° 41'E, 2165 m | Mt, Mix | 2017 | CPC | 10 |
| DEM_Athens | DEM | GR | 37°59'N,23°48'E, 270 m | Coast, U | 2017 | MPSS | 10.0 – 550.0 |
| Dresden-Nord | DRN | DE | 51°3'N, 13°44'E, 116 m | Con, U | 2016 | MPSS | 5.1 – 800.0 |
| Dresden-Winckelmannstrasse | DRW | DE | 51°2'N, 13°43'E, 120 m | Con, U | 2017 | MPSS | 10.0 – 800.0 |
| Deutschneudorf | DTC | DE | 50°36'N, 13°27'E, 660 m | Con, U | 2017 | MPSS | 10.0 – 800.0 |
| Finokalia | FKL | GR | 35°19'N, 25°40'E, 250 m | Coast, RB | 2017 | MPSS | 8.7 – 848.1 |





| | | | | | | | |
|---|---|---|---|---|---|---|---|
| SIRTA Atmospheric Research Obs. | GIF | FR | 48°42'N, 2°9'E, 162 m | Con, U | 2017 | MPSS | ~10 – 1000# |
| Helmos Mountain | HAC | GR | 37°59'N, 22°11'E, 2310 m | Mt, Mix | 2017 | MPSS | 10.0 – 550.0* |
| Hohenpeissenberg | HPB | DE | 47°48'N, 11°0'E, 985 m | Mt, RB | 2017 | MPSS | 10.0 – 800.0 |
| Ispra | IPR | IT | 45°47'N, 8°37'E, 209 m | Con, U | 2017 | MPSS | 10.0 – 800.0 |
| Jungfraujoch | JFJ | CH | 46°32'N, 7°59'E, 3578 m | Mt, Mix | 2017 | MPSS | 17.2 – 469.8 |
| Kosetice | KOS | CZ | 49°34'N, 15°4'E, 535 m | Con, RB | 2017 | MPSS | 9.0 – 841.7 |
| K-puszta | KPS | HU | 46°58'N, 19°34'E, 125 m | Con, RB | 2017 | MPSS | 6.3 – 794.0 |
| Leipzig TROPOS | LEI | DE | 51°21'N, 12°26'E, 113 m | Con, U | 2017 | MPSS | 5.1 – 800.0 |
| Leipzig-Eisenbahnstrasse | LEI-E | DE | 51°20'N, 12°24'E, 120 m | Con, U | 2017 | MPSS | 10.0 – 800.0 |
| Leipzig-Mitte | LEI-M | DE | 51°20'N, 12°22'E, 111 m | Con, U | 2016 | MPSS | 5.1 – 800.0 |
| Madrid | MAD | ES | 40°27'N, 3°43'W, 669 m | Con, U | 2016 | MPSS | 14.6 – 661.2 |
| Melpitz | MEL | DE | 51°31'N, 12°56'E, 86 m | Con, RB | 2017 | MPSS | 5.1 – 800.0 |
| Montsec | MSA | ES | 42°3'N, 0°43'E, 1571 m | MT, Mix | 2017 | CPC | 7 |
| Montseny | MSY | ES | 41°46'N, 2°21'E, 700 m | Mt, RB | 2017 | MPSS | 11.6 – 855.8 |
| Neuglobsow | NGL | DE | 53°10'N, 13°1'E, 62 m | Con, F | 2017 | MPSS | 10.0 – 800.0 |
| Obs. Perenne de l'Environnement | OPE | FR | 48°33'N, 5°30'E, 392 m | Con, RB | 2017 | MPSS | 9.8 – 543.7 |
| Pallas (Sammaltunturi) | PAL | FI | 67°58'N, 24°6'E, 565 m | P, P | 2017 | MPSS | 7.1 – 499.4 |
| Pic du Midi | PDM | FR | 42°56'N, 0°8'E, 2877m | Mt, Mix | 2017 | CPC | 10 |
| Prague-Suchdol | PRG | CZ | 50°7'N, 14°23'E, 270 m | Con, U | 2017 | MPSS | 9.5 – 519.4 |
| Puy de Dôme | PUY | FR | 45°46'N, 2°57'E, 1465 m | Mt, Mix | 2016 | MPSS | 10.3 – 580.0# |



| | | | | | | | |
|---|---|---|---|---|---|---|---|
| Hyytiälä | SMR | FI | 61°51'N, 24°16'E, 181 m | Con, F | 2017 | MPSS | 3.2 – 1000.0 |
| Sonnblick | SNB | AT | 47°3'N, 12°57'E, 3106 m | Mt, Mix | 2017 | CPC | 7 |
| Schauinsland | SSL | DE | 47°54'N, 7°54'E, 1205 m | Con, F | 2017 | MPSS | 10.0 – 800.0 |
| Granada | UGR | ES | 37°9'N, 3°36'W, 680 m | Con, U | 2017 | MPSS | 10.9 – 495.8 |
| Värriö | VAR | FI | 67°46'N, 29°34'E, 400 m | P, RB | 2017 | MPSS | 3.2 - 708.0 |
| Vavihill | VAV | SE | 56°1'N, 13°9'E, 175 m | Con, F | 2017 | MPSS | 3.4 – 857.7 |
| Waldhof | WAL | DE | 52°48'N, 10°45'E, 74 m | Con, F | 2017 | MPSS | 10.0 – 800.0 |
| Zeppelin mountain | ZEP | NO | 78°54'N, 11°53'E, 474 m | P, Mt, P | 2017 | MPSS | 10.0 - 800.0 |
| Zugspitze-Schneefernerhaus | ZSF | DE | 47°24'N, 10°58'E, 2671 m | Mt, Mix | 2017 | MPSS | 10.0 – 510.4 |
| WMO VII, Antarctica | | | | | | | |
| Neumayer | NMY | DE | 70°39'S, 8°15'W, 42 m | P, Coast, Mix | 2017 | CPC | 4 |
| South Pole | SPO | US | 89°59'S, 24°47'W, 2841 m | P, P | 2017 | CPC | 11 |
| Trollhaugen | TRL | NO | 72°0'S, 2°32'E, 1553 m | P, P | 2017 | MPSS | 10.0 – 800.0 |

*The first size bin was excluded from the analysis for these sites (frequent negative concentrations). The diameter of the first bin included in the analysis is 11.2 nm for BEO and 11.1 nm for HAC.

#The size range indicated in the data file is larger for these sites (7.9 – 1357.7 nm and 3.0 – 995.0 nm for GIF and PUY, respectively), but measurements are actually conducted on the ranges reported in the table.

5   $2017 data were not available at the time of analysis for these stations.





## 3. Relevant metrics for the description of the total particle number concentration and size distribution

### 3.1 The total particle number concentration ($N_{tot}$)

#### 3.1.1 Definition – sensitivity to instrumental characteristics

While different nomenclatures are commonly used to refer to the particle number concentration (e.g. CN, PNC), the total
particle number concentration will be hereafter referred to as $N_{tot}$ in the present work, for consistency with Laj et al. (2020). Also following the same approach as in Laj et al. (2020), measurements performed with both CPC and MPSS were first analysed together in order to have as large spatial coverage as possible for the study of $N_{tot}$. To allow for the comparison of observations derived from both instrument types, particle concentration in the range between 10 and 500 nm was inferred from MPSS measurements and assimilated to $N_{tot}$. This size range was selected as it is common to most of the MPSS included in
this study, and its lower end is moreover comparable to the lower cut-off diameter of 15 of the 23 CPC involved in the comparison (10 or 11 nm) (Table 1). One should however keep in mind that some of the remaining CPC have significantly lower cut points (e.g. 2.5 nm at ARN, ETL and GSN), and that some MPSS in contrast only detect particles slightly larger than 10 nm (e.g. up to ~ 17 nm at JFJ), as such cut point differences are likely to influence $N_{tot}$. These aspects are discussed in more detail in the supplementary materials.
The relevance of this approach was further assessed by the comparison of $N_{tot}$ derived from collocated CPC and MPSS measurements, since, besides the effect of different lower cut points, differences may also arise from the fact that each of these instruments has its own operational characteristics and data treatment procedures. For example, CPC instruments detect particles smaller than their lower cut point, because the lower cut point corresponds to the diameter at which 50% of the particles are detected. This may have a non-negligible effect on $N_{tot}$ in the presence of a significant amount of small particles,
such as during NPF events. On the other hand, there may be an overestimation of the particle concentration in the nucleation mode (and consequently $N_{tot}$) by the MPSS if background counts of the CPC in the MPSS is too high, and becomes critical during the inversion process. Data from 6 stations (HPB, MSY, PAL, PUY, SMR and VAR), where both instruments are operated with lower cut-off diameters adapted to the comparison (i.e., ~ 10 nm for the CPCs and ≤ 10 nm for MPSSs, to allow proper calculation of $N_{tot}$), were used to assess such issues. As illustrated in Fig. S2, MPSS tend to retrieve slightly lower $N_{tot}$
compared to CPC at 4 sites, while the opposite is seen at the 2 remaining stations. The agreement between the two instruments is nonetheless fair at all sites, as reflected by the slopes relatively close to 1 (0.50 – 1.30) and the rather low y-intercept values (-30 – 1034) obtained for the linear fittings at most of the stations, as well as by the fairly large coefficients of determination ($R^2$>0.74) (Table S1).

#### 3.1.2 Methodology for the study of $N_{tot}$

The seasonal variations of $N_{tot}$ were explored based on the comparison of the seasonal medians. For simplicity, seasons were assigned using the common December–February (DJF), March–May (MAM), June–August (JJA), and September–November (SON) division at all sites, even for the stations where other time divisions would be more appropriate. This is the case, for



instance, at CHC, where the weather is affected by two main seasons (May–August and December–March) with tropical characteristics (i.e., dry and wet, respectively). Such specificities should be kept in mind when interpreting the results.

The diel cycle of $N_{tot}$ was in addition investigated based on the analysis of the autocorrelation and partial autocorrelation functions (ACF and PACF, respectively), using the approach described in the supplementary materials of the study by Collaud

Coen et al. (2018). Briefly, the autocorrelations at 1 hour (first lag) were first removed from the dataset, and the ACF and PACF were then calculated on the resulting whitened time series at each time lag up to lag 36. In the case of ideal diel cycles, one could simply use the PACF at lag 24 as a metric for the strength of the cycle (i.e., to evaluate how regular the cycle amplitude is), hereafter referred to as $D_{cy}$. Similar to Collaud Coen et al. (2018), the sum of the PACF between lags 22 and 26 was used instead, as the diel cycle may not always be found over a 24 hours period due to the variability of both the natural

and anthropogenic factors which determine it. There is no scale as such, or threshold values, that can be used to explain the quantitative meaning of $D_{cy}$, but $D_{cy}$ generally takes on higher values the more regular the diel cycle is over time. Only the PACF values statistically significant at 95 % confidence level were considered, and the diel cycles were calculated at the annual scale only, because the time series were too short (1 year, with limited data availability at some sites) to properly investigate the seasonal change of the diel cycle; this aspect is only briefly addressed through a few case studies. As further

explained in Sect. 4.2, a stricter coverage criterion was in addition imposed in this specific part of the analysis.

### 3.2 Methodology for the analysis of the PNSD

The study of the PNSD was performed based on the seasonal medians of the distribution. In order to help in the evaluation of the seasonal contrasts and in the comparison between the sites, log normal modes were additionally fitted to the median distributions, as described in Eq. 1.

$$\frac{dn}{dlog_{10}d_p} = \sum_i \frac{N_{m,i}}{\sqrt{2\pi}log_{10}(\sigma_{m,i})} exp\left(-\frac{\left(log_{10}(d_p)-log_{10}(D_{m,i})\right)^2}{2\times\left(log_{10}(\sigma_{m,i})\right)^2}\right) \tag{1}$$

where $N_{m,i}$, $D_{m,i}$ and $\sigma_{m,i}$ are the concentration (cm$^{-3}$), the peak mean diameter (nm) and the geometric standard deviation of mode $i$, respectively. The analysis of the PNSD (including the fitting procedure) was restricted to the size range 20-500 nm to avoid possible bias in the comparison of the sites 1) due to differences in lower cut points or 2) related to increased uncertainty in the measurement of sub-20 nm particles (Wiedensohler et al., 2012). This also allowed a relevant description of the PNSD

with only two log normal modes, as previously done by Asmi et al. (2011). With this approach, the first mode is often a combination of the usual nucleation and Aitken modes, as reflected by the relatively high geometric standard deviation compared to that of the second mode (see Table A1 and Fig. S6). Nevertheless, this first mode will be referred to as Aitken mode for simplicity. The bimodal description performs well in reproducing the observations, as illustrated by the relatively large coefficients of determination obtained between measured and fitted PNSD (R²>0.98, Table A1), supporting the relevance

of such approach.



### 3.3 Investigation of the CCN-sized fraction of aerosols

The ability of a particle to act as CCN is determined both by its intrinsic properties (size and chemical composition) and by the surrounding atmospheric conditions (cloud supersaturation). The relative importance of particle size and chemical composition (which determines, in particular, its hygroscopicity) in the activation process has been the subject of multiple studies, sometimes leading to contrasting results (Schmale et al., 2018 and references therein). Some conclude that the particle size is paramount in determining the CCN impact (e.g. Dusek et al., 2006), while the knowledge of its chemical composition, including the size resolved chemical composition and state of mixing, seems more important in other situations, in particular when fresh pollution aerosol is considered (e.g. Ervens et al., 2010).

The spatial and temporal variability of CCN concentrations, as well as the properties of the particles involved in cloud formation, have recently been studied by Schmale et al. (2017, 2018) using long-term measurements of CCN number concentration, particle number size distribution and chemical composition performed at 12 sites representative of various environments. While the value of such collocated observations, even when temporary, is well demonstrated by Schmale and co-workers, there is no such data for all the sites considered in this study. A simpler approach has therefore been adopted here, based on the assumption that all particles larger than a given activation diameter are potential CCN, regardless of their chemical composition. This approach was previously used by Asmi et al. (2011), and also in several studies specifically dedicated to the evaluation of the contribution of NPF to the formation of CCN (Kerminen et al., 2012 and references therein; Rose et al. 2017, 2019). Very good agreement between measured CCN and predictions from size distribution data only was for instance reported from JFJ by Jurány et al. (2011). The relevance of such method was further validated by Hoyle et al. (2016): using activation diameter statistics from multiple campaigns (Hammer et al., 2014), they showed that 79% of the observed variance in cloud droplet numbers at JFJ could be explained by the concentration of particles larger than 80 nm. This threshold diameter was close to the overall median activation diameter (87 nm) reported by Hammer et al. (2014) for an approximate cloud supersaturation of 0.35%. A tight connection between cloud droplet number concentration and the concentration of particles larger than 100 nm, itself very close to the CCN concentration measured at 0.24% supersaturation, was also observed at PUY by Asmi et al. (2012). One should however keep in mind that such approach might be less accurate for the prediction of CCN in the presence of fresh pollution aerosol, whose ability to act as CCN may depend more largely on chemical composition than in the case of aged particles, such as those sampled at PUY or JFJ.

Similar to Asmi et al. (2011), two different activation diameters were considered in the present work, 50 and 100 nm, in order to reflect the abovementioned effects of both the properties of the particle itself and atmospheric conditions in the activation process. These threshold diameters are consistent with the findings of previous studies based on direct CCN measurements, which indicate that the smallest particles involved in the formation of real atmospheric cloud droplets are usually in the range 50–150 nm; those include in particular the results of Schmale et al. (2018), who report that at 0.2% supersaturation, activation diameters have a distribution centered around or slightly larger than 100 nm at most of the sites involved in their analysis. The


number concentrations of particles in the ranges 50-500 nm and 100-500 nm, hereafter referred to as $N_{50}$ and $N_{100}$, were thus inferred from available MPSS measurements and used as proxies for the CCN number concentration.

## 4. Data availability – Coverage criteria

### 4.1 Impact on the annual and seasonal statistics of $N_{tot}$

In the analysis of $N_{tot}$ presented in Laj et al. (2020), annual and seasonal statistics were reported when 75% of the hourly data was available over the statistics reference period (year or season). In cases when the 2017 coverage was not sufficient (i.e. <75% for all seasons) or 2017 data were not available at the time of analysis, the 2016 data were considered instead. Three stations were nevertheless discarded from the analysis (MSA, RUN and VAV) due to not having adequate coverage for either year, and among the 285 medians (annual and seasonal) which could have been expected for the other 57 sites, only 197 (69%)

were effectively calculated due to insufficient data availability in the remaining cases. As illustrated in Fig. 2, long gaps are seen in some datasets, indicating that despite the efforts made to ensure continuous measurements, interruptions (e.g. caused by instrumental failure or malfunctioning, natural disasters) cannot be avoided, and the difficulty of access to some of the sites can further complicate the situation. However, while these long gaps obviously result in reduced data availability at some sites, the 75% coverage required in Laj et al. (2020) may have been too high, also limiting the number of statistics able to be included

in the analysis.

The first aim of the present study was thus to investigate the effect of reduced data availability on the statistics of $N_{tot}$ to evaluate the possibility of lowering the 75% threshold used in Laj et al. (2020) without compromising the relevance of the analysis. For that purpose, the 11 sites with an annual data coverage of more than 95% were selected (ETL, IPR, KOS, LEI-E, NGL, NMY, PAL, SNB, THD, TRL, and VAR) and, for each site, the statistics derived from the original dataset were

compared to those calculated from reduced datasets in which the absence of data was simulated. The selected stations do not represent all geographic and footprint categories, but they remain representative of a variety of environments. Two different approaches were used to investigate how, on top of the data availability itself, the length and configuration of the missing periods were affecting the results. Note that, however, none of these approaches were designed to address the effect of regular/cyclic gaps in the datasets, or corresponding to very specific conditions prone to affect the instrument or the

transmission of the data. They also do not intend to evaluate the effect of intentional data rejection resulting from automatic filtering based on systematic criteria (e.g. wind direction). Such filtering occurs at SPO, BRW and MLO; for these three stations, the coverage criteria discussed here were not applied.

Exclusion of weeks was first performed to replicate long gaps in the data, similar to what can happen in the event of an instrument failure. Note that a week refers here to a block of 7 or 8 days, so that, for the sake of simplicity, each month has 4

weeks and the full year is 48 weeks long in total. The exclusion of 1 to 24 consecutive weeks was tested at the annual scale, and in each case all possible combinations were considered (e.g. there are 47 possibilities to exclude 2 consecutive weeks out of 48). The median and percentiles of $N_{tot}$ were computed for all combinations, and for each combination we calculated the





ratio of the newly derived median of $N_{tot}$ over that derived from the original dataset. In addition, in order to gain more insight into the variability associated with each simulated gap length, the maximum of the 75$^{th}$ percentile of $N_{tot}$ obtained from the different combinations was divided by the 75$^{th}$ percentile of $N_{tot}$ calculated from the original dataset. Similarly, the 25$^{th}$ percentile from the original full $N_{tot}$ dataset was divided by the minimum of the 25$^{th}$ percentile of all the different combinations.

As illustrated in Fig. 3, there is almost no impact on the annual statistics of $N_{tot}$ when the measurement interruption is shorter than 4 – 5 weeks, and the effect remains limited for all types of sites up to ~ 12 weeks missing, with most of the medians computed from the reduced datasets within a factor of 1.5 of that derived from the original datasets. The variability is however more pronounced for the polar sites (NMY, PAL, TRL and VAR), especially as the length of the measurement interruption increases. This observation is consistent with the strong seasonal contrast of $N_{tot}$ highlighted for these sites in Laj et al. (2020)

and further discussed in Sect. 5.2.1. For data gaps of up to 18-19 continuous weeks missing, the medians of the ratios are relatively evenly distributed around 1. In contrast, as the simulated gap in the data gets longer, the distribution of the ratios becomes less symmetric around 1, clearly reflecting the fact that the seasonal cycle of $N_{tot}$ (regardless of its strength) is not represented in the statistics anymore. In fact, the absence of more than 19 consecutive weeks implies that at least part of the period JJA, when either the highest or lowest concentrations are often measured (depending on the hemisphere, see Sect. 5),

is missing, which in turn affects the statistics.

The same analysis was repeated at the seasonal scale, and exclusion of individual hourly averages was finally tested at both scales, annual and seasonal, to reproduce the rejection of sporadic data points as it may occur, for instance, during data quality control. The corresponding results are detailed in the Supplement. For comparable data availability, long interruptions in the datasets tend to have a slightly stronger impact on the statistics compared to the absence of individual data points. As illustrated

in Figs. 2 and S5, such long interruptions are moreover mostly responsible for the low data coverage observed at some sites. Indeed, 9 of the 14 sites which have an annual data availability below 64% have experienced measurement interruptions longer than 90 days, and, more broadly, 29 of the 39 stations which have an annual data availability lower than 88% have missing data over periods longer than 30 days (Fig. S5). The definition of the coverage criteria to be used in this work was in turn based on the results obtained from the simulation of long gaps in the datasets. Based on the observations from Fig. S3, a threshold

of 50% was set at the seasonal scale, and 60% of the data were required at the annual scale to ensure some minimal representativeness of the datasets with respect to seasonal cycles (Fig. 3). Although they are not based on strict statistical criteria, these thresholds seem to allow a reasonable compromise between availability and quality of statistics for the dataset of interest. Following these criteria, the three stations (MSA, RUN and VAV) discarded from the study of $N_{tot}$ reported in Laj et al. (2020) were included in the present work. These looser requirements also allowed the analysis of 53 more summary

statistics for the 57 other sites already included in Laj et al. (2020). Furthermore, unlike in Laj et al. (2020), the data from 2016 were used for THD in order to benefit from greater coverage for this station, which closed in early June 2017. Note that for consistency, in spite of the modified coverage criteria, the 2016 data was still considered for the sites for which this was already the case in Laj et al. (2020); this also made it possible to increase the number of statistics for all these sites (10 in total) except WLG.



## 4.2 Impact on the estimation of the $D_{cy}$

Using the same approach as in Sect. 4.1, the effect of reduced data availability on the autocorrelation patterns and, more importantly on the amplitude of the diel cycle of $N_{tot}$, was investigated. As introduced in Sect. 3.1.2, the $D_{cy}$ was calculated as the sum of the PACF coefficients obtained for the whitened time series of $N_{tot}$ for lags between 22 and 26 hours. The analysis was performed at the annual scale with the threshold data availability of 60% defined in Sect. 4.1 as a starting point, and the sensitivity of $D_{cy}$ to the data coverage was further investigated by also simulating data availability of 75%. These targets were reached in two ways: first by excluding 19 and 12 consecutive weeks, respectively, from the original time series, and second by removing enough randomly selected, non-contiguous individual hourly averages. As with the statistics of $N_{tot}$, all possible combinations of weeks to exclude were considered in the first case, and the second test was repeated 25 times with different sets of randomly selected hourly averages. An overview of the results obtained at all sites is shown in Fig. 4. More specifically, Figs. 4.a and 4.c, show, for each of the reduced datasets, the ratio between the newly derived $D_{cy}$ and the value found in the original time series. In addition, Figs. 4.b and 4.d further illustrate the variability of the $D_{cy}$, calculated for each site as the difference between the maximum and the minimum of the $D_{cy}$ found in the reduced datasets normalized by the $D_{cy}$ of the complete time series.

As illustrated in Fig. 4.a, long interruptions in the time series overall have more significant effect on the $D_{cy}$ than on the statistics of $N_{tot}$ (Sect. 4.1). The exclusion of 12 and 19 weeks nonetheless lead to comparable results, as reflected by the variability of the $D_{cy}$ (Fig. 4.b), which is often close in both cases. On the other hand, this variability is observed to decrease with the magnitude of the $D_{cy}$ in the original dataset, which suggests that the evaluation of the $D_{cy}$ is all the more uncertain in reduced time series as its value is already low in the complete dataset. Although they have a more pronounced effect on the $D_{cy}$ than on the statistics of $N_{tot}$, gaps of longer consecutive periods have, for the same resulting data availability, weaker impact compared to the absence of individual data points. This observation, which contrasts with the findings of the previous section, is expected because the number of value pairs available for the determination of the ACF (and consequently affecting the PACF and $D_{cy}$ calculation) drops significantly when an increasing number of sporadic values are missing, with a likely effect on the significance of the resulting correlations. In such situation, negative $D_{cy}$ may appear, a priori without physical meaning, but rather in response to the decreased amount of data in the reduced datasets, while positive values are associated with the complete datasets. This is the case for all the sites highlighted by a black square at the top of panels a. and, more importantly c., of Fig. 4, and for which such negative $D_{cy}$ are not shown. Note that observations from TRL are not presented since negative $D_{cy}$ is obtained in the original dataset at this site; again this negative value is most likely an artefact, which is thought to arise in this case from the very strong variability of $N_{tot}$ caused by the occurrence of snow storms between April and August at this site. As evidenced in Fig. 4.c, the occurrence of negative $D_{cy}$ is the most frequent when degrading the data availability to 60%, and the variability of the $D_{cy}$ is also the highest, up to almost 300% (Fig. 4.d). When the simulated data availability is raised to 75%, the occurrence of negative $D_{cy}$ is less frequent, but the variability of the $D_{cy}$ remains on average more pronounced than in the case of consecutive missing weeks. As in the case of longer interruptions, the variability of the





$D_{cy}$ resulting from the absence of individual data points seems, however, to decrease with the magnitude of the $D_{cy}$ in the complete timeseries.

Based on these last observations, and even if long interruptions (e.g., due to instrument failure) were the main reason for decreased data availability in the datasets (Fig. S5), the coverage criteria was raised to 75% for the study of the $D_{cy}$, and the

5   main analysis was limited to the annual scale. The seasonal change of the diel cycle was only briefly investigated at few sites with particularly high coverage to give further insight into the findings obtained at the annual scale. All the results presented in Sect. 6 should nonetheless be considered with caution, as the length of the selected datasets remains in any case limited for such application.

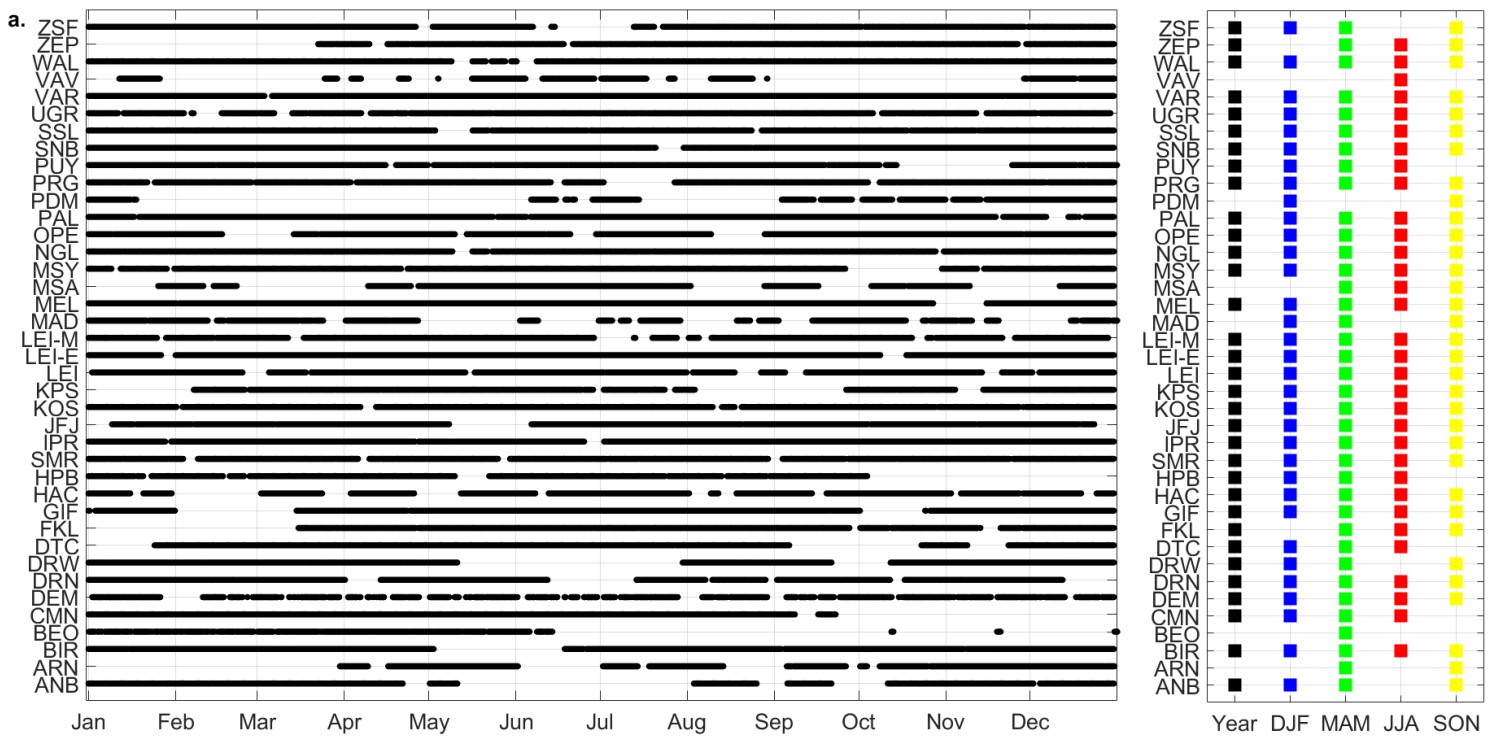


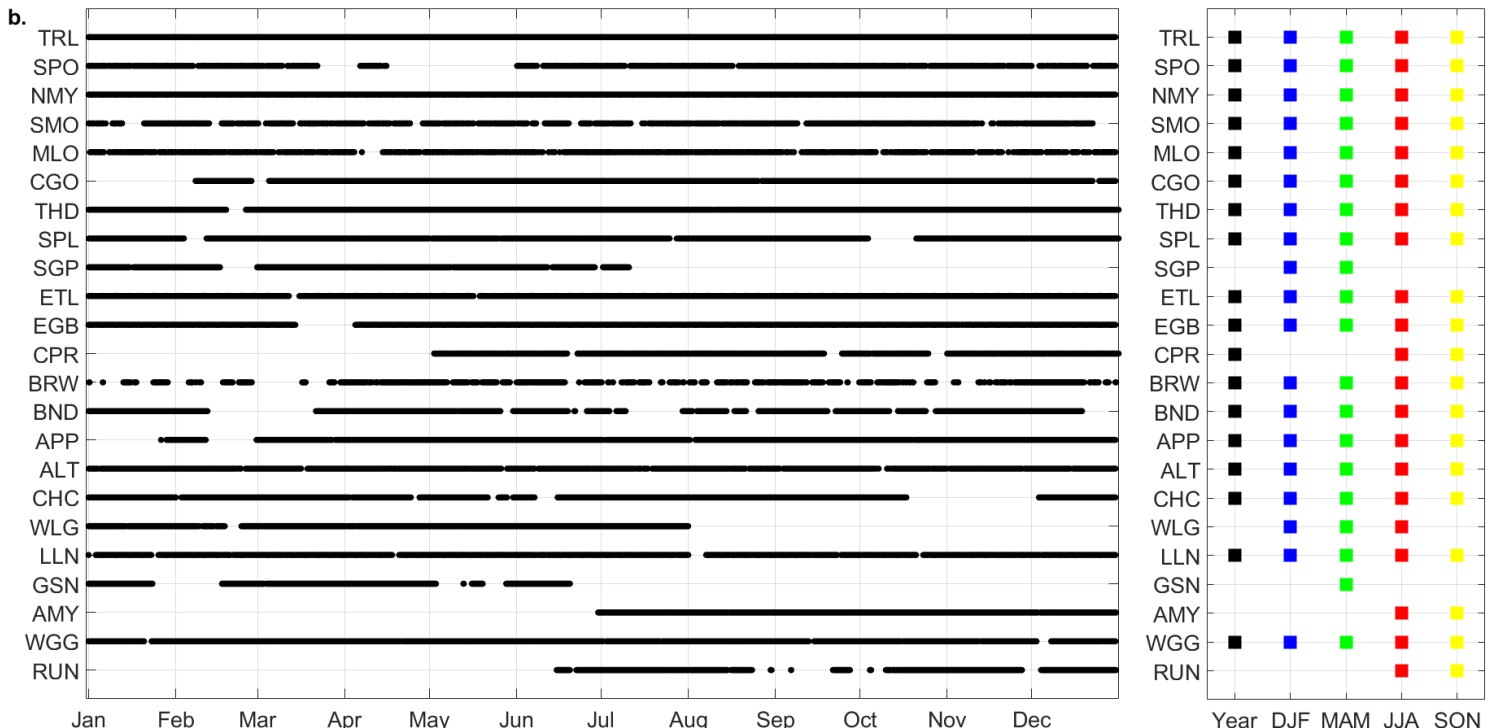

Fig. 2 Data coverage of the sites. For clarity, a. European and b. remaining stations are shown separately. Black dots on the left panel indicate the presence of valid hourly data, and markers on the right panel indicate the periods (year and seasons) for which the corresponding data availability was sufficient to compute statistics (i.e. 50% for seasons and 60% for the full year).




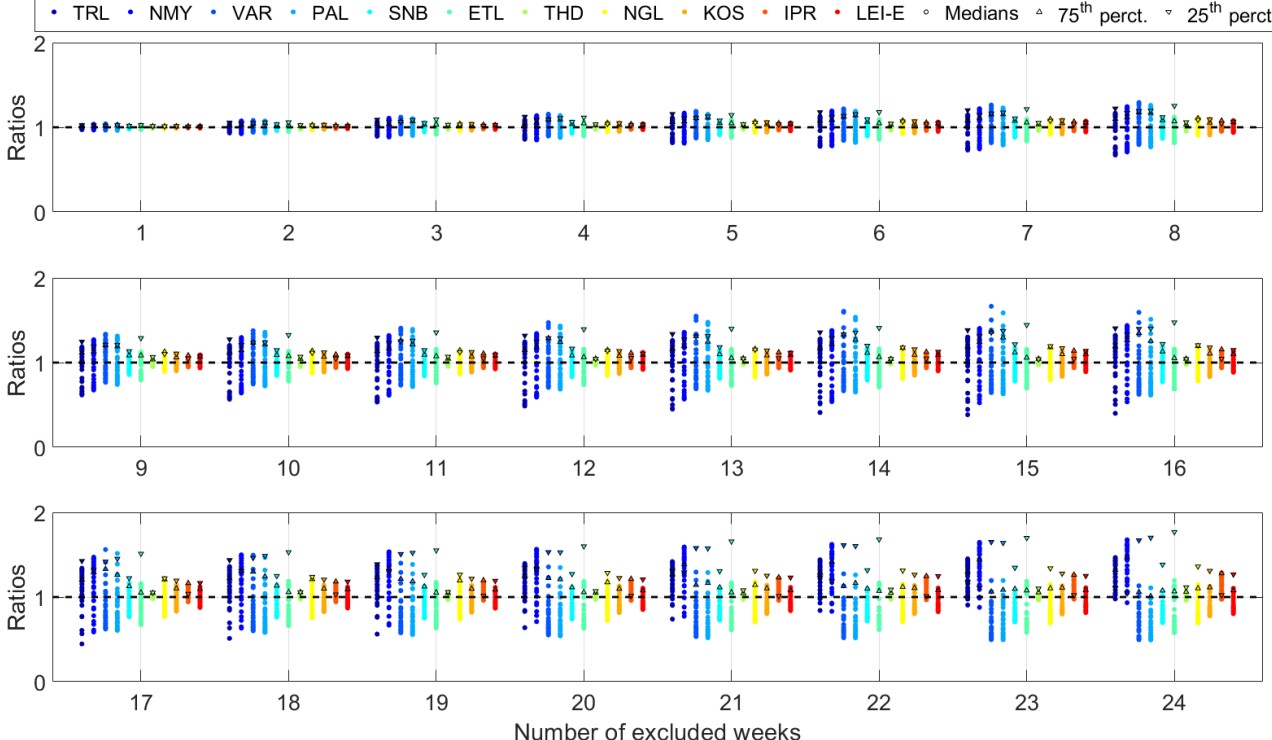

Fig. 3 Variability of $N_{tot}$ annual statistics in reduced datasets. For each investigated gap length (between 1 and 24 consecutive weeks), all possible combinations were tested, and in each case the ratio between the newly derived median of $N_{tot}$ and that derived from the original dataset was calculated (circles). The upward and downward triangles provide insight into the range of variability. The upfacing triangles represent the ratio between the maximum value of the 75th percentile of $N_{tot}$ obtained from the reduced datasets and the 75th percentile calculated from the original time series. The downfacing triangles represent the 25th percentile from the original dataset divided by the minimum of the 25th percentile.

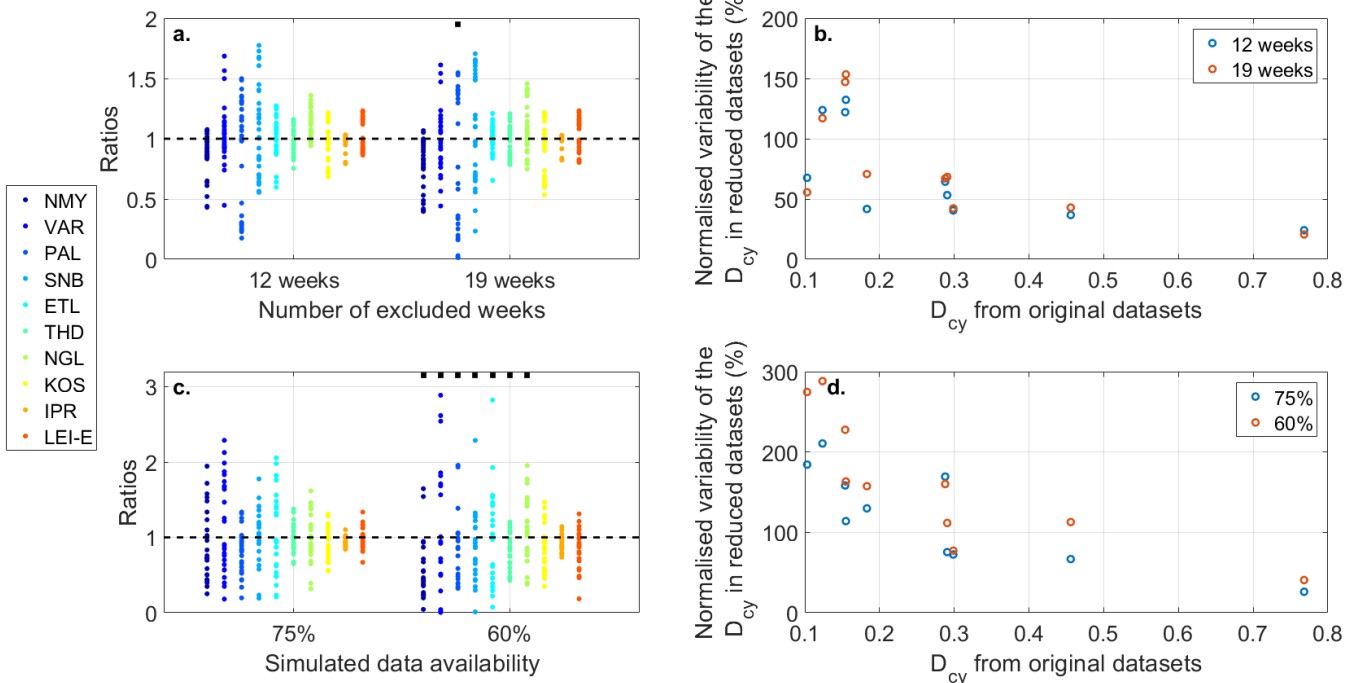

Fig. 4 Variability of the diel cycle ($D_{cy}$) of $N_{tot}$ in datasets with data availabilities degraded to ~ 60% and ~ 75% as a result of the exclusion of 12 and 19 consecutive weeks (all possible combinations tested) (two top panes) and individual hourly averages (test repeated 25 times) (bottom two panes). In a. and c., the ratio between the newly derived $D_{cy}$ and that calculated from the original dataset was calculated for each reduced dataset. Black squares indicate the occurrence of negative $D_{cy}$ (not considered in the calculation of the ratios) at the corresponding sites. Panels b. and d. show the variability of the $D_{cy}$, calculated for each site and each target data availability as the difference between the maximum and the minimum of the $D_{cy}$ derived from the reduced datasets normalized by the $D_{cy}$ calculated from the complete time series, as a function of the original $D_{cy}$.

## 5. Seasonality of the total particle number concentration and size distribution

### 5.1 Structure of the section

The seasonality of $N_{tot}$ was investigated first, together with the PNSD when measurements were available. The results are discussed below, separately for the following station types defined as combinations of geographical and/or footprint criteria among which comparable trends or features could be identified: mountain sites, polar stations, continental and coastal urban stations and remaining lowland sites (i.e., non-urban continental and coastal stations). Note that all polar sites characterized by an additional geographical category (i.e., ALT, BRW and NMY, Table 1) were considered only as polar sites in this analysis. Figure 5 provides an overview of the spatial distribution of $N_{tot}$ based on the medians (annual and/or seasonal) computed for all sites, which are also reported in Table S2 in the Supplement. This overall picture is complemented by the results shown in





Fig. 6, which offers an additional viewpoint based on the ranking of the sites according to 1) the annual median of $N_{tot}$ (Fig. 6.a), in a similar way as in Fig. 8 in Laj et al. (2020), and to 2) the ratio of the maximum and minimum seasonal medians of $N_{tot}$ (Fig. 6.c). This ratio, hereafter referred to as SeasC and used as a metric to evaluate the seasonal variability of $N_{tot}$, was calculated when all seasonal medians were available; the seasons corresponding to the medians used in the calculation of

SeasC are also shown for each site on the right hand side of Fig. 6.c. In addition to Fig. 6.a, which, together with the annual median of $N_{tot}$, also indicates the corresponding 1$^{st}$ and 3$^{rd}$ quartile, Fig. 6.b provides the normalized interquartile range of $N_{tot}$, hereafter referred to as NIQR. The NIQR, calculated as the ratio of the interquartile range over the corresponding median aims to allow a comparison of the variability of $N_{tot}$ independent of the concentration level observed at each site. The NIQR corresponds in other words to the relative variability of $N_{tot}$ expressed as a percentage of the median, which is used as a

reference in this approach. Similar information is also provided at the seasonal scale in Fig. 7 for further investigation of the intra-seasonal variability of the particle concentration. Note that the analyses presented in Fig. 6 are restricted to the stations where data availability was sufficient over the periods of interest.

The study was further limited to the sites where MPSS data was available for the investigation of the PNSD. Median distributions and corresponding modes are shown for each site and season (depending on data availability) in Figs. A1-A6 in

Appendix A, and corresponding characteristics of the modes (i.e. modal concentrations $N_{m,1}$ and $N_{m,2}$, mode peak locations $D_{m,1}$ and $D_{m,2}$ and geometric standard deviations $\sigma_{m,1}$ and $\sigma_{m,2}$) are reported in Table A1. In addition, the modal parameters are shown for all sites as a function of their type in Figs. 8, 9 and S6, which also indicate, for each station, the site-specific variability of each parameter. For a given site, this variability was calculated when at least two seasons were available as the ratio between the standard deviation of the parameter over the corresponding mean (calculated from all available seasonal

values, i.e., between 2 and 4 seasons). Similar to the NIQR, such normalization was adopted to allow for the quantification of the variability regardless of the absolute value of the parameters, and in turn make the comparison between the sites more relevant. One should however keep in mind that the variations of the modal parameters are often connected when interpreting the site-specific variability of each single parameter. As an example, changes in the concentration and width of a mode can be seen concurrently, and the resulting increase or decrease of the modal concentration can contrast with the initial guess one

could make from the visualisation of the median distributions only. This is for instance the case at ZEP, where the MAM to JJA increase of $N_{m,1}$ reported in Table A1 is not as pronounced as expected from the clear enhancement of the sub-50 nm particle concentration visible in Fig. A1 due to the concurrent strong decrease of $\sigma_{m,1}$ (Table A1 and Fig. S6). As evidenced in Fig. S6, site-specific variability of the geometric standard deviation is overall limited for both modes (9% on average), but can nonetheless reach 27-28%, with the highest variability observed at urban sites.





## 5.2 Results from different station types

### 5.2.1 Polar sites

As shown in Figs. 5 and 6.a, the lowest particle concentrations are on average observed at polar stations, where annual medians of $N_{tot}$ are of the order of $10^2$ cm$^{-3}$. Consistent with earlier observations by Asmi et al. (2011), the variability of the particle

number concentration is the most pronounced at these sites, as shown in Fig. 6.a and further reflected by the corresponding NIQR presented in Fig. 6.b (~ 160% on average, up to ~ 240% at PAL and VAR). This variability is primarily related to a remarkably strong seasonal contrast of $N_{tot}$ at most of these stations (SeasC > 7 at 5 of the 7 documented sites, Fig. 6.c), with, in particular, enhanced concentrations observed during local summer which often contrast with winter minima. The exception is BRW, where all seasonal medians are quite similar. The variability of $N_{tot}$ is also influenced by a pronounced intra-seasonal

variability at some stations (Fig. 7), including for instance BRW during JJA (NIQR ~ 250%), and to a slightly lesser extent at TRL and NMY during MAM (~ 210%).

The corresponding PNSD are generally characterized by an Aitken mode located around 42±14 nm and an accumulation mode found, on average, at 149±37 nm (Table A1 and Fig. 9). Similar to $N_{tot}$, the shape of the PNSD is nonetheless highly variable at polar stations (Fig. A1), with the largest site-specific variability observed for $N_{m,1}$ (in the order of 89% on average versus

59% for $N_{m,2}$, Fig. 8). The variability of $N_{m,1}$ is significantly more pronounced at polar stations compared to other station types, and also contrasts with the trend observed at other sites, where $N_{m,2}$ is instead more variable throughout the year. Enhanced concentrations of Aitken mode particles coinciding with the maximum of $N_{tot}$ during local summer more specifically appear as a common feature of the four polar sites equipped with a MPSS (Table A1 and Fig. 8).

Despite their distinctive behaviour, slight differences are noticed among the stations located at high latitudes. This first includes

the tendency of $N_{tot}$ to further decrease towards the poles, under conditions of minimal anthropogenic influence, down to 38 cm$^{-3}$ at SPO during local winter. The PNSD measured at these sites also experience contrasting evolution throughout the year. In fact, the summer PNSD is almost unimodal at PAL and VAR, and differs significantly from the bimodal distributions observed during other seasons. At the Arctic station ZEP, in spite of the strong changes exhibited in Fig. A1 (in particular between MAM and JJA), two distinct modes are clearly seen during all investigated seasons (DJF missing), while this bimodal

feature is in contrast much less pronounced at TRL regardless of the season. While being less obvious, changes in the modes peak location also accompany the evolution of the modal concentrations at the sites located in the Northern Hemisphere (Fig. 9), with the most pronounced site-specific variability again observed for the Aitken mode, in the order of 28% on average (versus 11% for the accumulation mode). Larger diameters are more specifically seen for both modes during MAM at ZEP, and later at PAL and VAR, coinciding with the maximum of $N_{tot}$ observed during JJA, while the modes diameters are in

contrast almost constant at TRL.

While similar processes are certainly contributing to $N_{tot}$ at all these sites, contrasting properties of the PNSD likely result from varying sources and local specificities across the relevant latitude ranges. Transport was for instance reported as an important source of Aitken and accumulation mode particles during summer at Arctic sites such as ZEP and ALT (Croft et al., 2016).





Secondary aerosol formation, including in specific NPF, was furthermore observed at polar stations (Kerminen et al., 2018 and references therein; Nieminen et al., 2018), with slightly different seasonal patterns that presumably result from the diversity of condensing vapours (and their associated concentration) involved in the process at the different sites. For instance, compounds of marine origin that are related to ocean ice cover and biological activity are likely more contributing to aerosol

formation in the pristine conditions found at the sites located at extreme latitudes (Abbatt et al., 2019; Jang et al., 2019) than at sub-Arctic sites such as PAL and VAR. Finally, some specific phenomena have also been previously reported to affect the PNSD. This is for instance the case during episodes of Arctic haze, which causes elevated number concentrations of accumulation mode particles during springtime in the Arctic region (Abbatt et al., 2019 and references therein), as reflected in the measurements collected at ZEP during this time of the year (Fig. A1).

**5.2.2 Urban stations**

In contrast to polar sites, stations located in urban areas, both continental and coastal, exhibit the highest $N_{tot}$, with yearly medians in the range $10^3$-$10^4$ cm$^{-3}$ (Figs. 5 and 6.a). As shown in Figs. 6.a and 6.b, the variability of $N_{tot}$ is also less pronounced in urban conditions, with an average NIQR of ~ 90%. Specifically, these sites, that are all located in Europe, display only limited seasonal variation (SeasC < 2 for the 9 documented sites, Fig. 6.c). Despite the lack of a clear trend in the seasonal

cycle, slightly greater medians are nonetheless observed during summer at 5 stations, while winter concentrations are on average higher at IPR and UGR, where the most pronounced contrast is seen. Intra-seasonal variability is also minimal at urban stations, with NIQR mainly below 100% (Fig. 7).

The weak seasonality of $N_{tot}$ is associated with limited changes of the PNSD, which are almost unimodal throughout the year and shifted towards the lower end of the investigated size range at a majority of urban sites compared to other station types,

with elevated concentrations of sub-100 nm particles (Figs. A2-A3). The distributions are specifically dominated by a wider Aitken mode compared to other station types ($\sigma_{m,1} > 2$) (Table A1 and Fig. S6), which is on average located at 32±11 nm and only experiences a limited seasonal variation of its properties at most of the sites (on average 20% and 22% for the mode diameter and modal concentration, respectively, Table A1 and Figs. 8-9). In contrast, the characteristics of the accumulation mode show more variability for a given site (in the order of 26% and 77% for the mode peak location and concentration,

respectively), but with no clear pattern among the sites. On average, this second mode is positioned at 122±37 nm but is often found below 100 nm, and sometimes even overlaps strongly with the Aitken mode (Table A1 and Fig. 9). Furthermore, the accumulation mode can be relatively wide, as observed for instance at LEI-M and DRN during DJF (Fig. S6). The shape of the PNSD at IPR, while also almost unimodal, is slightly different from those of the other urban sites, with features comparable to those observed for rural background continental sites. As noticed earlier by Asmi et al. (2011), distinctive behaviour at IPR

is in particular observed in DJF, with elevated particle concentrations around 100 nm resulting from the accumulation of aerosols in the lowermost levels of the troposphere (<1000 m) during this time of the year (Barnaba et al., 2010). As mentioned before, increased concentrations of ground level particles are also measured during winter at UGR, in particular in the range





50-100 nm (Fig. A2), and were earlier attributed to the combined effect of several factors including ABL dynamics and enhanced anthropogenic activities (domestic heating) by Lyamani et al. (2010).

More broadly, sub-100 nm particles, which often dominate the urban PNSDs, are emitted directly into the atmosphere from combustion processes related to traffic, industry or residential heating, or from other sources, such as vehicle brakes, and they can be formed as well from gaseous precursors (Rönkkö et al., 2019). As indicated in this recent review, a number of studies have been conducted to investigate the characteristics of urban aerosol, and to assess the relative importance of the abovementioned sources. Different approaches have been used, including simultaneous measurements of the PNSD at different locations in the same urban area (e.g. Harrison et al., 1999; Salma et al., 2014), possibly coupled with laboratory experiments (Rönkkö et al., 2017), or the application of statistical methods for the analysis of data collected at a single site (Pey et al., 2009; Dall'Osto et al., 2012; Al-Dabbous and Kumar, 2015; Brines et al., 2015). All of these studies agree on a very strong contribution of traffic related emissions to the total particle number concentration. More specifically, Pey et al. (2009) indicate that road traffic could explain, on average, 54%, 69%, 74% and 86% of the particle concentration measured at Barcelona (Spain) in the ranges 13 – 20, 20 – 30, 30 – 50 and 50 – 100 nm, respectively, while Rönkkö et al. (2017) and Olin et al. (2020) report the importance of traffic emissions in the sub-3 nm range as well. While traffic related emissions are often subject to daily variation (e.g. increase during morning and evening rush hours), probably affecting the diel cycle of $N_{tot}$ at urban sites (see Sect. 6), they however experience more limited seasonal variation, which likely explains the weak seasonality of $N_{tot}$ in urban areas. The fact that slightly higher particle concentrations are observed during summer at a number of urban stations, when the atmospheric boundary layer (ABL) height is also increased compared to colder months, suggests, however, that there are certainly additional sources of aerosols in summer which compensate for the ABL dilution effect. Increased concentrations of sub-40 nm particles are observed during MAM and more importantly JJA at some stations (PRG, LEI, DRN and GIF, Fig. A2), supporting a probable role of secondary aerosol processes in the build-up of increased summer $N_{tot}$ at these sites. This assumption is supported by the results of Salma et al. (2014) and Brines et al. (2015), who report that NPF can represent a significant source of particles in the urban atmosphere, in particular during spring and summer, and more broadly under high insolation conditions. In addition to supplementary sources, we also cannot exclude an effect of seasonally reduced particle sink on $N_{tot}$ at some sites. Such an effect was for instance reported for Botsalano (semi-clean location) and Marikana (industries and residential area nearby) in South-Africa, where the lack of wet removal during the dry season (from May to September) contributes to higher particle number concentrations during this time of the year, in particular above 100 nm (Vakkari et al., 2013). The studies of Harrison et al. (1999) and Salma et al. (2014) also underline the strong spatial heterogeneity of observations within a given urban area, also visible in our dataset when comparing measurements from LEI and LEI-E, which are separated by ~3 km only. Fresh traffic emissions have a strong impact on the shape of the PNSD, with increased amount of small particles (<10 nm) compared to urban background sites (Harrison et al., 1999; Salma et al., 2014; Rönkkö et al., 2017), and also experiences high-frequency variations, which can be attributed (at least partly) to the wide variety of vehicular sources emission characteristics (Harrison et al., 1999). This in particular the case for roadside samples, such as those collected at DRN, LEI-E and LEI-M in the present study.



### 5.2.3 Non-urban sites and mountain stations

The remaining sites, including mountain and non-urban continental and coastal stations, do not have as clear a common behaviour as polar and urban sites and display, on average, intermediate $N_{tot}$, with yearly medians of the order of $10^2$-$10^3$ cm$^{-3}$ (Figs. 5 and 6.a). As shown in Fig. 6.a, the signature of their dominant footprint is noticeable, with lower concentrations

measured in forested areas, or at stations influenced by air masses of various origins ("mixed"), if compared to rural background sites. However, the distinction between the different geographical categories (i.e. mountain, continental and coastal) is in contrast less pronounced. Nonetheless, as noted in Laj et al. (2020) and in agreement with previous observations from Asmi et al. (2011), particle concentrations measured at mountain sites tend to be lower compared to nearby lowland sites, as observed for instance for SNB (3106 m a.s.l., annual median of $N_{tot}$ ~1027 cm$^{-3}$) and KOS (535 m a.s.l., 2690 cm$^{-3}$). Also,

as discussed below, mountain sites, and specifically those characterized by mixed footprints, tend to exhibit somewhat more pronounced seasonal variations relative to lowland stations. This is likely a result of the strong impact of the ABL height variability (e.g. Herrmann et al., 2015; Rose et al., 2017) in connection with the topography of the sites (Collaud Coen et al., 2018), which largely determines the contribution of long range transport relative to more local sources of particles.

### i.     Non-urban continental and coastal sites

Particle number concentrations measured at non-urban continental and coastal sites are overall lower compared to those observed at urban stations, but similar features are observed among all these lowland sites. Specifically, the variability of $N_{tot}$ is comparable (NIQR ~100%, Fig. 6.b), as a result of both limited intra (Fig. 7) and inter seasonal variability (Fig. 6.c). A slight enhancement of $N_{tot}$ is visible during local spring (6 sites) or summer (9 sites) at all 17 non-urban lowland sites documented in Fig. 6.c except ETL and THD, where higher concentrations are instead found in autumn. Similar to urban sites,

this likely results from the concurrent variability of particle sources and ABL dynamics, as for instance hypothesized for OPE by Farah et al. (2020), who suggested a biogenic secondary source for the extra particles observed in the warmest seasons. Hoewever, as mentioned already, an effect of seasonally reduced sink (mainly from precipitation) on the variations of $N_{tot}$ can also not be excluded at some sites (e.g. Vakkari et al. 2013). As shown in Fig. 6.c, the stations located in forested areas tend to exhibit stronger seasonal variations. This is likely explained by the biogenic nature of at least some of the aerosol sources

at these sites, which are affected by a strong seasonality that is related to the biosphere activity. The distinct nature of these forested sites is also visible in the PNSD, which tend to have a more pronounced bimodal shape compared to rural background stations, where the distributions are, in contrast, more monomodal and similar to those observed at urban sites (Figs. A3-A5). Specifically, the northernmost stations located in forested areas, SMR and BIR, feature similar PNSD variations as the sub-Arctic polar stations PAL and VAR, including a growth of the Aitken mode in summer with greater concentrations and larger

mode diameters (Table A1 and Figs. 8-9). On average, the Aitken and accumulation modes are found at 51±13 and 174±29 nm, respectively, at non-urban sites. These are actually the largest mode diameters among all station types, with the most noticeable shifting (compared to other station types) observed for the first mode at the two coastal sites AMY and FKL (Table


A1 and Fig. 9). Despite being less pronounced compared to urban stations, the site-specific variability for $N_{m,2}$ is also significant at non-urban sites, in the the order of 48% on average (versus 31% for $N_{m,1}$, Fig. 8). In spite of the clear seasonal variations in the PNSD at some of these sites (Fig. A4), the variability of $D_{m,1}$ and $D_{m,2}$ is, on average, also less pronounced than at urban sites (16% and 12% for $D_{m,1}$ and $D_{m,2}$, respectively, Fig. 9).

Despite the differences observed in terms of level of $N_{tot}$ and characteristics of the PNSD, this last analysis highlights similarities between observations conducted from urban and non-urban areas, and particularly between measurements from urban and rural background sites. This result suggests that diluted urban aerosol is likely contributing to the aerosol sampled at a number of non-urban stations, in particular those located in the vicinity or urban areas.

**ii.**     **Mountain stations**

As mentioned earlier, the seasonality of the observations collected at mountain sites is somewhat stronger than at lowland stations (other than polar). This is the case in particular at stations characterized by mixed footprints, where there can be up to a factor of almost 5 difference between the maximum and minimum seasonal medians of $N_{tot}$ (Fig. 6.c). Similar to polar sites, higher $N_{tot}$ are mostly found during local summer (6 sites), and often contrast with winter minima (5 sites). The main exception

is CHC, which sees its highest $N_{tot}$ during JJA, which, as noted in Sect. 3.1.2, coincides with the dry season at this site located in the tropics. This seasonal contrast contributes to an average NIQR of ~117% for mountain sites (Fig. 6.b), which is also explained by the relatively marked intra-seasonal variability of $N_{tot}$ compared to lowland sites (other than polar) (Fig. 7). Note that the particularly low NIQR values observed at MLO (Figs. 6.b and 7, between 38 and 46% in the different seasons) are likely related to the automatic filtering of the data based on wind direction.

The PNSD collected at mountain sites exhibit a stronger bimodal behaviour compared to lowland stations (other than polar), with mean diameters for the two modes close to those obtained for polar sites. These modes are, on average, found at 39±9 nm and 142±25 nm, but, similar to $N_{tot}$, significant variability of the PNSD is observed, both among the sites and seasons. The most significant site-specific variability is observed for $N_{m,2}$ (in the order of 76% versus 36% for $N_{m,1}$, Fig. 8), while, like all other station types except urban, the peak location of the Aitken mode is slightly more variable (20%) than that of the

accumulation mode (13%) (Fig. 9). The contrast between the sites is sometimes striking, as observed for instance for JFJ and CHC, where the medians of $N_{tot}$ differ by one order of magnitude (Fig. 6.a) as a likely result of the contrasting surroundings of these sites. The impact of the emissions from the neighboring urban area of La Paz (~20 km) on the measurements performed at CHC was demonstrated by Chauvigné et al. (2019), while there is no such major source of pollution in the vicinity of JFJ. Similarities among sites can also be seen. For instance, the two mountain stations located below 1000 m a.s.l., MSY and HPB,

feature $N_{tot}$ levels and variability comparable to those of rural background continental sites (Fig. 6), and less obvious bimodal behaviour of the PNSD, particularly for MSY (Figs. A5 and A6).



Following this last observation, the connection between the medians of $N_{tot}$ and the elevation of the sites was further investigated, separately for each season (Fig. 10.a). The linear fit between these two variables is shown in the plot to further guide the eye, but the strength of the correlation was more specifically evaluated by the mean of the Spearman's rank correlation coefficient, which does not require the variables to be normally distributed and assess monotonic relationships,

whether linear or not. Note that in order to include measurements from CHC and RUN (the two mountain sites located in the Southern Hemisphere), local seasons are considered in this part of the study (i.e., for example, DJF data from CHC and RUN contribute to summer data). In addition, in order to include as many sites as possible, we did not limit this analysis to the sites with sufficient data availability over all four seasons, which means that the number of points considered in the search for correlations varies from season to season, from 11 in fall to 16 in spring.

As shown in Fig. 10.a, there is a tendency for $N_{tot}$ to decrease with altitude in all seasons but winter, where the opposite is seen. However, the correlations between $N_{tot}$ and the station elevation are not statistically significant for any season except summer, where the correlation is found to be statistically significant at 95% confidence level. This last observation is consistent with the fact that measurements collected at mountain sites during this time of the year are likely more connected to the lower tropospheric layers due to increased ABL dynamics (including thermally driven wind systems) and height; they are instead

more representative of free tropospheric air masses and long range transport during winter, where a weaker connection between altitude and $N_{tot}$ is thus expected. The results of this correlation study seem, however, to be strongly influenced by the observations from CHC, which is the highest station and where, nonetheless, winter concentrations are for instance much higher compared to other sites. We cannot exclude that the use of the common division DJF-MAM-JJA-SON is not adapted to this station located in the tropics, but, more broadly, this result questions the relevance of using altitude alone to describe

the influence of lower tropospheric levels on measurements performed at mountain sites. Based on Collaud Coen et al. (2018), the meso-scale topographical features around the station should be considered as well; the connection between $N_{tot}$ and the ABL-TopoIndex (Collaud Coen et al., 2018), an index defined to provide a more complete characterization of the ABL influence at high altitude sites, was thus investigated here as well. This topography based index is defined in such a way that the greater the influence of the ABL, the higher the value it takes. As shown in Fig. 10.b, all correlations are statistically

significant at 90% confidence level and positive. This result is consistent with earlier findings by Collaud Coen et al. (2018), who more specifically highlighted a positive correlation between particle concentration and the components of the ABL-TopoIndex describing the ease of local transport of both particles and their precursors to the station. The overall stronger connection observed between $N_{tot}$ and the ABL-Topoindex (compared to the station elevation alone) clearly illustrates the need to take the topography around the sites into account to characterize the ABL influence on observations performed at mountain

stations. In summer, however, the correlation between $N_{tot}$ and the ABL-topoindex appears to be weaker than in the case of altitude, as reflected by the absolute value of the corresponding Spearman's rank correlation coefficients (0.57 versus 0.76). During this time of the year, inputs from the ABL at mountain sites are certainly not only more frequent, but also associated to higher particle loading, in line with increased $N_{tot}$ observed in the lower layers (Sects. 5.2.1-5.2.3.i). We may hypothesize that this combined effect has a strong impact on the connection between $N_{tot}$ and altitude, while it could be in contrast less





prevalent when the configuration of the site and its environs is taken into account; this would explain the lower Spearman's rank correlation coefficient obtained in the correlation between $N_{tot}$ and the ABL-Topoindex. Repeating the same approach with the modal concentrations instead of $N_{tot}$ would have probably provided more insight into these aspects, but such analysis was not performed for the present work because of the limited data availability in some seasons.

Overall, the topography and environs of the sites (which determine the ABL influence) combined with the variations of the ABL height strongly affect the seasonal cycles of the particle number concentration and size distribution observed at mountain stations. At JFJ, for instance, the greatest variability is observed for $N_{m,2}$, the median of which is increased by almost one order of magnitude between local winter and summer (Table A1 and Fig. 8). This results from the increased frequency of ABL injections during summer, which are the main source of accumulation mode particles at this site (Herrmann et al., 2015). Such

significant variability of $N_{m,2}$ is also seen at CHC, where it is accompanied by a widening of the accumulation mode and decrease of its mean diameter, reflecting the overall shifting of the whole distribution towards the lower end of the investigated size range during JJA (Table A1 and Figs. 8, 9 and S6). The concentration of sub-40 nm particles is clearly enhanced during this time of the year at CHC (Fig. A6), coinciding with elevated NPF frequency observed at the site (Rose et al., 2015). Additional insight into the occurrence and role of NPF at mountain stations is more broadly considered in the recent review

by Sellegri et al. (2019).

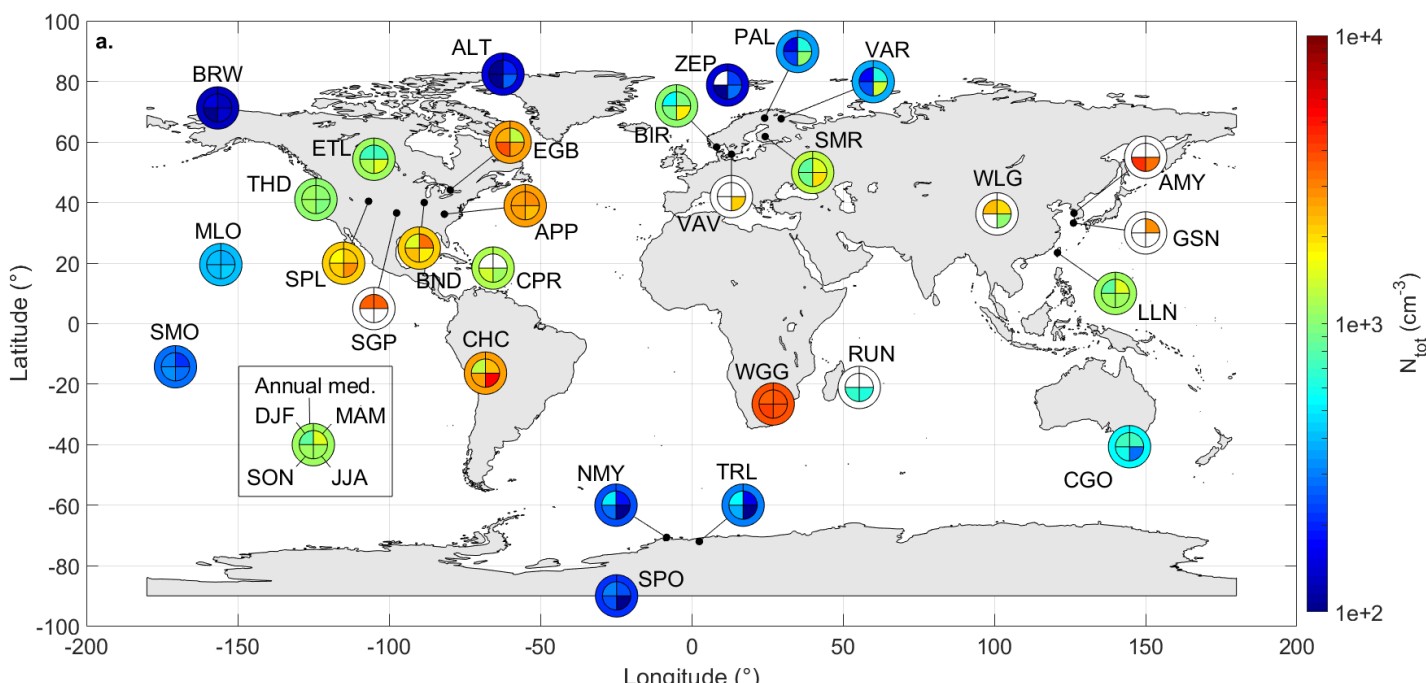





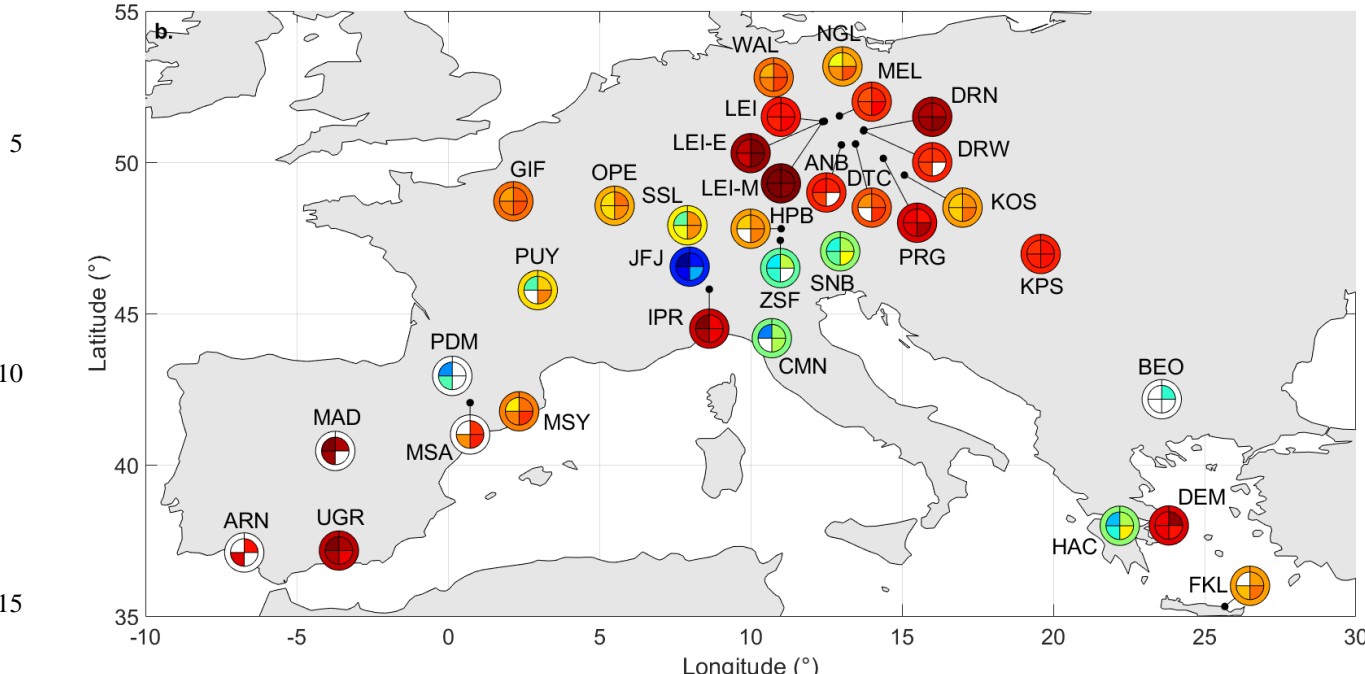

Fig. 5 Geographical distribution of $N_{tot}$ a. at the global scale and b. specifically over Europe. As indicated in the legend, for each station, seasonal medians are shown in the inner part of the corresponding pie chart and the annual median is represented by the outer ring. The symbols are left empty when data availability was not sufficient over the corresponding period.



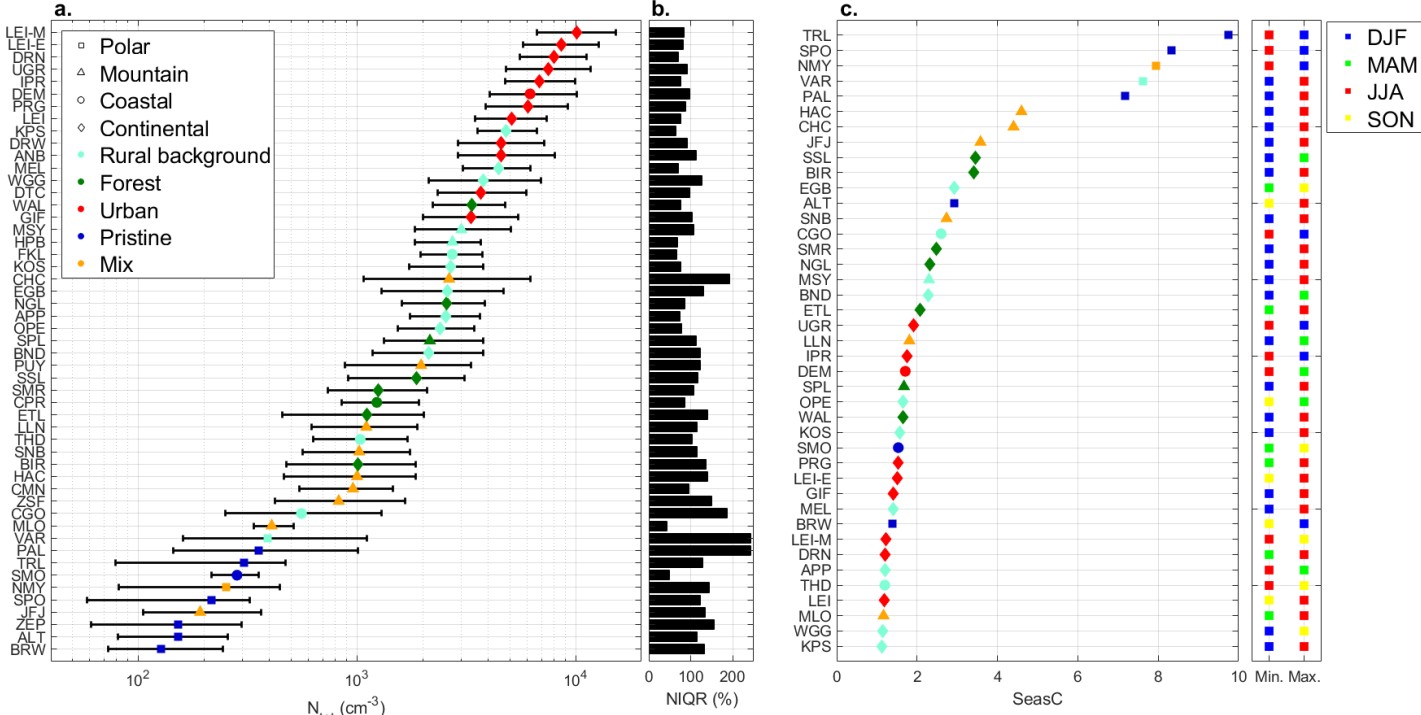

Fig. 6 Ranking of the stations based on a. the annual median of $N_{tot}$ and c. the variable SeasC, used as a metric to evaluate the
magnitude of the seasonal contrast of $N_{tot}$. In a., the markers represent the median of $N_{tot}$, and the left and right end of the error
bars represent the 1$^{st}$ and 3$^{rd}$ quartile of the data, respectively. The annual variability of $N_{tot}$ is again shown in b. by means of
the NIQR calculated for each site displayed in a.. Different symbols and colours in a. and c. indicate geographical and footprint
categories, respectively. The right hand side of c. also shows the seasons corresponding to the minimum and maximum seasonal
medians of $N_{tot}$ used in the calculation of SeasC. Additional explanations regarding the calculation and interpretation of SeasC
and NIQR are available in Sect. 5.1.

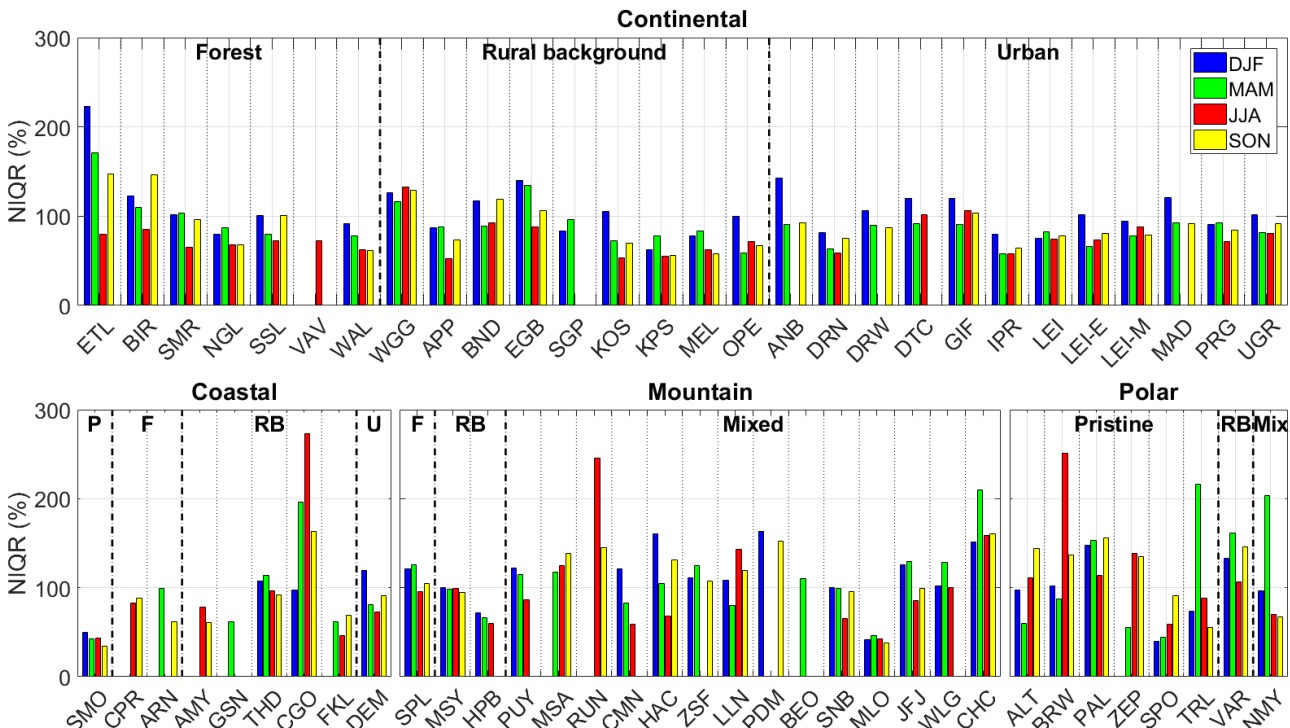

Fig. 7 Intra seasonal variability of $N_{tot}$. Stations are sorted based on the classification reported in Table 1. The meaning of the abbreviations used for the footprint is the following: P for pristine, F for forest, RB for rural background, U for urban and Mix for mixed. Explanations regarding the calculation and interpretation of the NIQR are available in Sect. 5.1.

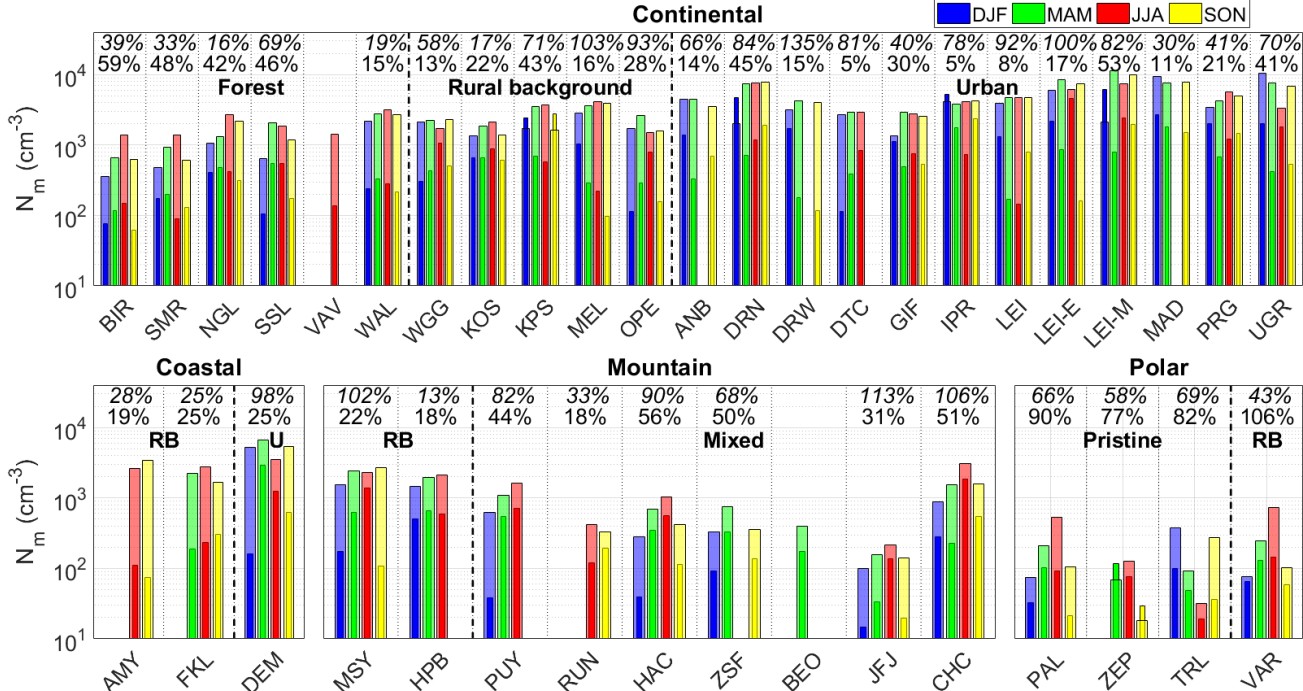

Fig. 8 Modal concentrations. For each site and season, the thicker bar represents the modal concentration of the Aitken mode

($N_{m,1}$) and the thinner one that of the accumulation mode ($N_{m,2}$). The values at the top of each panel indicate the site-specific variability of the modal concentrations, with the italicized text corresponding to $N_{m,2}$. The meaning of the abbreviations used for the footprint is the following: RB for rural background and U for urban. Details regarding the calculation of the site-specific variability of the modes characteristics are available in Sect. 5.1.





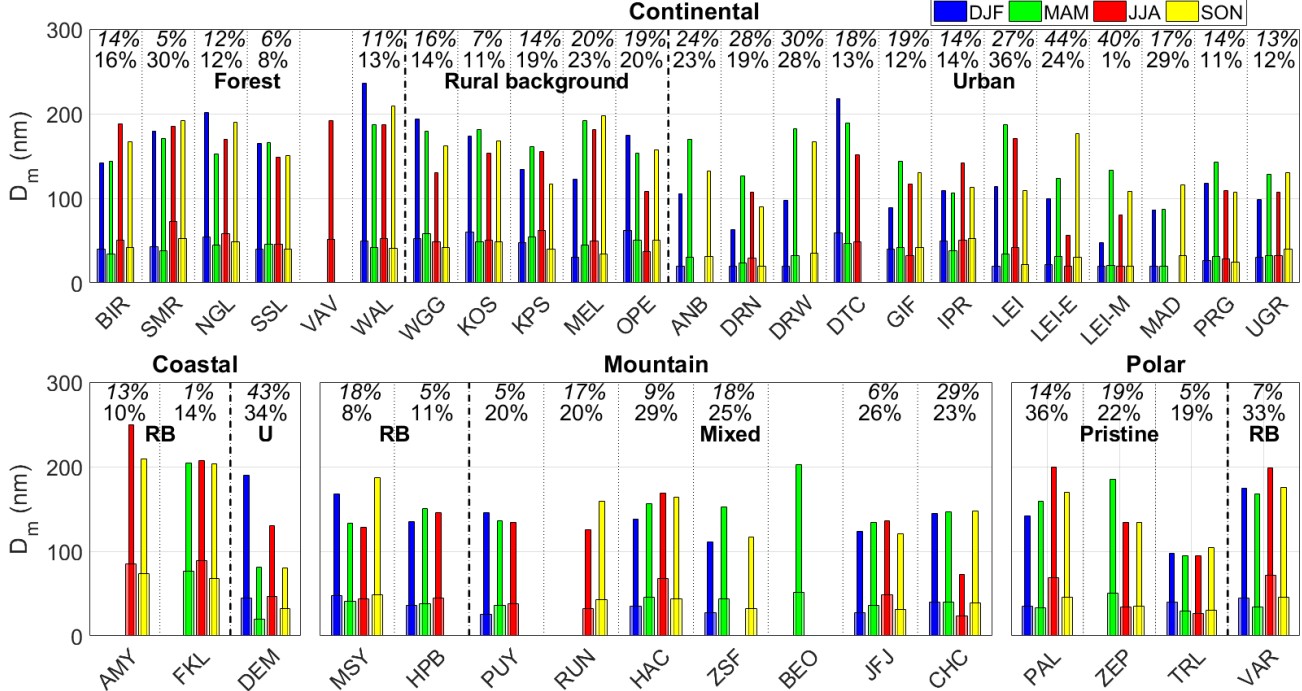

Fig. 9 Modes peak location. For each site and season, the thicker bar represents the mean diameter of the Aitken mode ($D_{m,1}$) and the thinner one that of the accumulation mode ($D_{m,2}$). The values at the top of each panel indicate the site-specific variability of the position of the modes, with the italicized text corresponding to $D_{m,2}$. The meaning of the abbreviations used for the footprint is the following: RB for rural background and U for urban. Details regarding the calculation of the site-specific variability of the modes characteristics are available in Sect. 5.1.





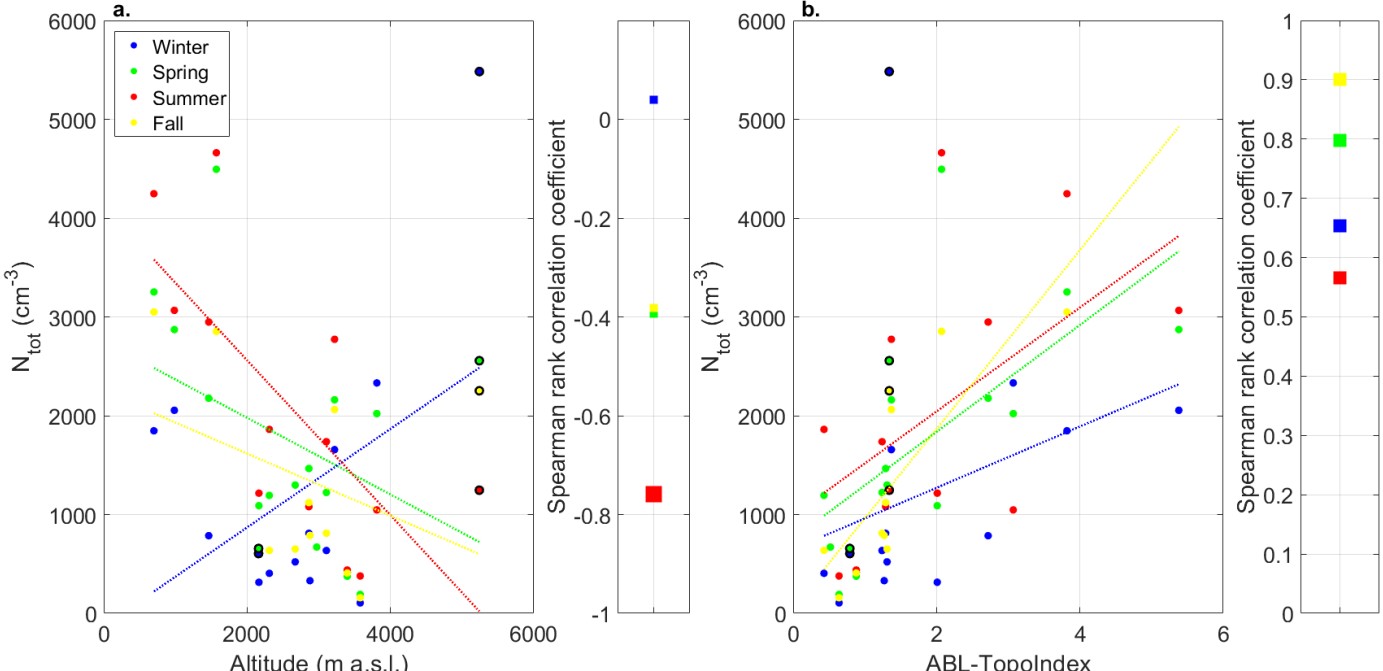

Fig. 10 Connection between $N_{tot}$ and a. the elevation of the mountain sites and b. corresponding ABL-TopoIndex in each season. The linear fit between the two variables is shown in each case to further guide the eye, but the strength of the correlation is evaluated based on the Spearman's rank correlation coefficient. Statistically significant correlations at 95 % and 90 %
confidence levels are marked by large and medium symbol sizes, respectively, while non-statistically significant correlations are represented by small symbols. Observations from RUN and CHC, the two mountains sites located in the Southern Hemisphere, are highlighted by the black circles in the scatter plots.

## 6. Diel cycle of the total particle number concentration

Figure 11.a presents the $D_{cy}$ calculated at the annual scale for the 34 sites that had sufficient data availability (>75%). To help interpret these results, Fig. 11.b additionally shows the seasonal $D_{cy}$ calculated for the 11 sites with the highest coverage (>95% at the annual scale, and in turn sufficient data availability in all seasons) previously involved in the sensitivity studies reported in Sect. 4. For DJF, where the three months considered are not consecutive, the $D_{cy}$ has been calculated in two different ways: first, by proceeding as for the other seasons, as if the three months were consecutive, and second, by excluding the calculation of autocorrelation over non-consecutive periods. The results of these two approaches are presented in Fig. 11.b, DJF V1
corresponding to the first one and DJF V2 to the second. As a reminder, only the PACF coefficients (between lags 22 and 26) statistically significant at 95 % confidence level were used in the calculation, which explains why some $D_{cy}$ are missing in Fig. 11.b, in the absence of significant PACF values. Negative $D_{cy}$ have also been filtered out, which is why, in particular, the





annual $D_{cy}$ obtained for TRL and JFJ, both negative based on the 2017 data, are not shown in Fig. 11.a. As already indicated in Sect. 4.2, these negative values likely have no physical meaning; rather, they most probably result from an alternation of contrasting conditions at the site (e.g. in connection with the dynamics of the ABL at JFJ), or from specific meteorological phenomena (e.g. snow storms at TRL) that impact the average diel cycle of $N_{tot}$. It can also not be excluded that the value

reported for ZSF may be affected by the daily absence of data between 11:00 and 22:00 UTC from July 15[th] onwards at this site.

Contrasting values are observed among polar stations, but the annual $D_{cy}$ is on average weak at these sites (Fig. 11.a), as a likely consequence of the absence of a regular day-night cycle in some seasons, and also because there is no strong anthropogenic activity prone to influence the $D_{cy}$ in these pristine environments. As shown in Fig. 11.b, $D_{cy}$ are in fact mainly

reported during the transition seasons, when there is a day/night distinction and NPF events are also observed (e.g. Nieminen et al., 2018) that can contribute to the identified cycles. The average behaviour described by the annual $D_{cy}$ is therefore of limited value for these polar sites which, in addition to the common characteristics mentioned above, have individual specificities that also affect the diel cycle of $N_{tot}$. As mentioned already, this is for example the case of TRL, where the occurrence of snow storms between April and August have a strong impact on the evolution of $N_{tot}$.

Overall, higher $D_{cy}$ are in contrast found at urban and mountain sites (Fig. 11.a). In urban conditions, the diel cycle is probably largely influenced by anthropogenic factors that have a strong diurnal variability but, on the contrary, limited seasonal variations (e.g. morning and evening traffic rush hours), thus allowing a noticeable regularity of these cycles over the year. Indeed, relatively high $D_{cy}$ are observed in all seasons at IPR and LEI-E (Fig. 11.b). The lower summer values, observed at both sites, are probably related to a decrease in traffic and increase in ABL height, while domestic heating, which is well-

commonly more intense from October to April, certainly contributes to the identification of more pronounced cycles during these months. At mountain sites, diel cycles of $N_{tot}$, like seasonal cycles, are probably largely influenced by ABL dynamics. The continuous influence of the residual or continuous aerosol layer in summer (see Collaud Coen et al. 2018 and references therein for the nomenclature), or, on the contrary, the lower ABL heights observed in winter, may lead to lower $D_{cy}$ during these seasons. This is observed, at least partially, at SNB, where the $D_{cy}$ in SON is higher than the summer and winter $D_{cy}$ (Fig.

11.b). However, this behavior is certainly not universal, and the environmental specificities of certain sites (e.g., island station or coastal zone, complex topography) can certainly also constrain the cycles. For example, given the altitude of LLN and its proximity to the ocean, it is possible that at this station the residual layer does not remain or is dispersed by winds during the night in summer, which could lead to higher $D_{cy}$ values at this time of the year. In addition, we cannot exclude that enhanced photochemical processes in summer, while contributing (together with increased precursor availability) to favour secondary

aerosol formation, might also influence the $D_{cy}$ at these sites.

For the remaining low altitude sites, $D_{cy}$ are observed over a wide range of values (Fig. 11.a), which can probably be explained by the diversity of conditions observed at these sites (e.g., altitude range, nature of the sources, including the proximity to anthropogenic sources). This diversity is reflected by the $D_{cy}$ reported in Fig. 11.b, which show contrasting seasonal cycles from one site to another.





While the latter analysis highlights some additional contrasts among the different station types, it also indicates that the interpretation of the annual $D_{cy}$ must be conducted with caution, in light of the type of station and the possible specificities of certain sites. When the diel cycle is relatively homogeneous throughout the year (e.g., urban sites), the annual $D_{cy}$ describes a real average behavior, whereas when the natural and/or anthropogenic factors that determine the $D_{cy}$ are highly variable from one season to another (e.g., polar sites), the annual $D_{cy}$ has only a limited value. The complete analysis of the $D_{cy}$ therefore requires a detailed seasonal study, taking into account the environmental characteristics of each site, and could be the subject of a future study using the extended time series available for some stations.

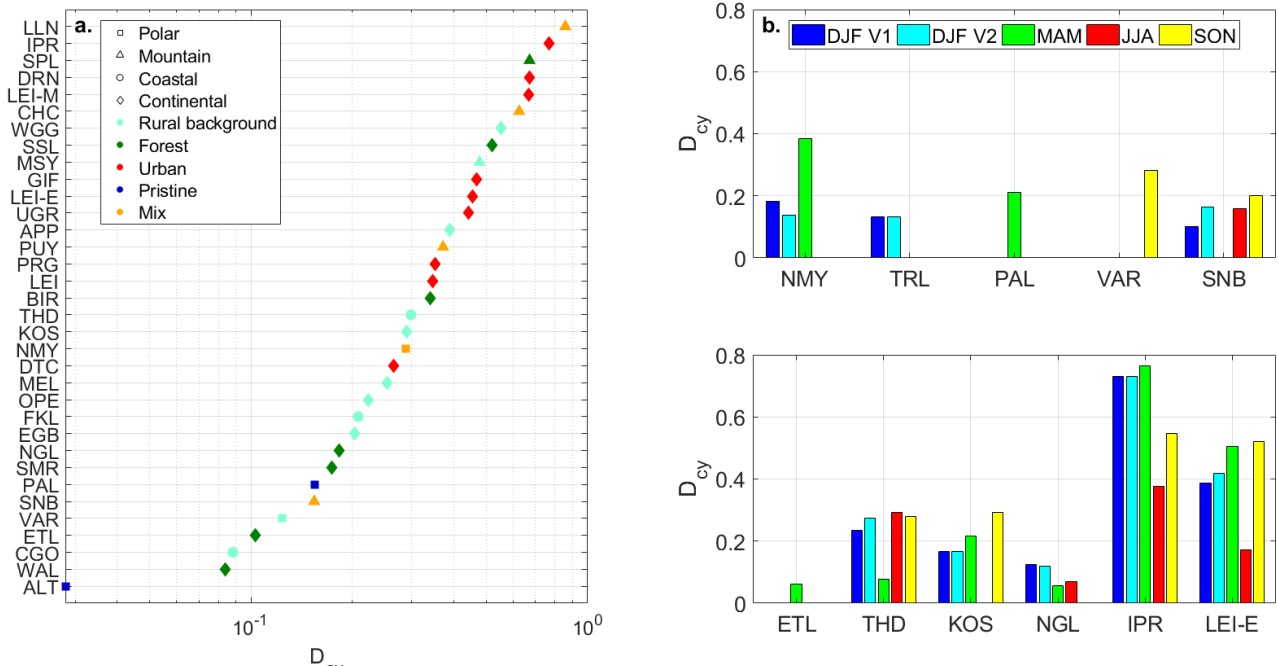

Fig. 11 a. Ranking of the stations based on regularity of the diel cycle ($D_{cy}$) of $N_{tot}$. b. presents the seasonal $D_{cy}$ calculated for the 11 sites with the highest coverage (>95%). Polar and mountain sites are shown in the upper panel, and other lowland stations are shown in the lower panel.

## 7. Focus on CCN-sized particles

As explained in Sect. 3.3, the number concentrations of particles in the ranges 50-500 ($N_{50}$) and 100-500 nm ($N_{100}$) were used as proxies for the number concentration of potential CCN. Since similar trends are obtained for $N_{50}$ and $N_{100}$, only the results corresponding to $N_{100}$ are shown herein (Figs. 12 and 13), while the equivalent observations for $N_{50}$ are shown in the Supplement (Figs. S7 and S8).





Figure 12 (and S7) shows the seasonal medians (as well as the first and third quartiles) of $N_{100}$ (respectively $N_{50}$). The trends observed for the different station types are similar to what was highlighted for $N_{tot}$. The lowest concentrations are again obtained for the polar sites, with medians for $N_{100}$ in the order of ~10 to a few hundred particles, which is on average lower than the values obtained at mountain and other non-urban lowland sites (~100 – 1000 cm$^{-3}$) and, more importantly, at urban

sites (~1000 cm$^{-3}$). Similar orders of magnitude are obtained for $N_{50}$, but with concentrations that are slightly higher due to the contribution of particles between 50 and 100 nm in diameter. As in the case of $N_{tot}$, there is some variability within each station type, and this seems to be more pronounced for mountain (e.g. JFJ vs CHC) and polar sites. Although based on a reduced number of sites, such intra-station type variability has also been shown by the direct measurements of the CCN number concentration reported by Schmale et al. (2018). With respect to the seasonal variations of $N_{50}$ and $N_{100}$, there are again

similarities with what was obtained for $N_{tot}$. In particular, we observe well-marked cycles for polar and mountain sites, almost non-existent cycles for urban sites, and a range of patterns for the remaining sites according to their characteristic footprint (e.g., stronger variations at forest compared to rural background lowland sites). There are, however, small differences with the results obtained for $N_{tot}$, particularly in the magnitude of the contrasts, which are probably related to the variability of the contributions of $N_{100}$ and $N_{50}$ to $N_{tot}$ in the different seasons, as demonstrated for example by Jurányi et al. (2011) at JFJ.

In order to further address this aspect, Fig. 13 (and S8) presents, for the stations which had sufficient data availability in all seasons (i.e. >50%, see Fig. 2), and separately for the 4 station types discussed so far, the relationship between $N_{100}$ (respectively $N_{50}$) and $N_{tot}$. Given the high number of points, raster graphs are used instead of standard correlation plots; on these graphs, the color of each pixel indicates the number of data points (hourly averages) falling into its area (all pixels have equal area on a log-log scale). The linear fit performed on the logarithm of the variables is also presented for the whole data

set and for each season separately. The logarithm is used here because it allows a more immediate visualization of the contribution of $N_{100}$ (or $N_{50}$) to $N_{tot}$ and its variability; the fit equations and corresponding coefficients of determination are reported in Table 2 (respectively S3). Statistics of the ratio between $N_{100}$ (respectively $N_{50}$) and $N_{tot}$ are in addition reported for each station type and period (year and seasons). Note that in order to allow and facilitate comparison of sites located in different hemispheres, local seasons are considered in this final analysis. Finally, as a complement to the distinction between

seasons, Fig. S9 presents the scatter plots of $N_{100}$ and $N_{50}$ as a function of $N_{tot}$ for polar, mountain and the remaining non-urban sites, this time highlighting the different footprints present in each class of sites.

As shown in Fig. 13, $N_{100}$ represents from a few tenths of percent to almost all of $N_{tot}$. The median annual contributions of $N_{100}$ to $N_{tot}$ are comparable at polar, urban and mountain sites (~19%), while being slightly higher at other lowland sites (~26%). The lowest contributions are observed during fall at polar sites, particularly at the two sites PAL and VAR located in the

Northern Hemisphere, and during winter at mountain sites (Fig. 13.a and d). These observations might be, at least partly, related to increased frequency of cloud occurrence during these seasons. This is for instance the case at PAL, where low clouds (below 1000 m) are more often seen during fall (Komppula et al., 2005), or at PUY, where the frequency of cloud occurrence is in the order of 60% in winter, compared to 24% in summer (Baray et al., 2019). In cloudy conditions, the sampling efficiency of activated particles may be lower than that of smaller intersticial particles, or even not possible in absence of  a whole air


inlet (e.g. PAL), thus leading to an artificial shift of the PNSD towards lower sizes. Highest ratios between $N_{100}$ and $N_{tot}$ are in contrast observed during summer at these sites, when clouds are less prevalent and the transport (in connection to ABL dynamics at mountain sites) of CCN sized particles is the most favoured (e.g. Croft et al., 2016; Herrmann et al., 2015). At lowland sites other than polar, higher contributions of $N_{100}$ to $N_{tot}$ occur during winter, when the presence of small particles in

connection with NPF is the less frequent and additional sources of larger particles, such as resential heating, are in contrast more intense. At urban stations, the $75^{th}$ percentile of the ratio between $N_{100}$ and $N_{tot}$ is on average lower compared to other station types, likely reflecting the significant contribution of traffic related sub-100 nm particles to $N_{tot}$ in all seasons (see. Sect. 5.2.2).

The contributions of $N_{50}$ to $N_{tot}$ are logically higher than those of $N_{100}$, systematically above a few percent and up to ~100%

for all station types, being on average twice as high at the annual scale (Fig. S8). Similar trends to those obtained for $N_{100}$ are observed, with, in particular, close median annual contributions for polar, urban and mountain stations (43 – 48%), and slightly higher contributions at other lowland sites (~55%). We also find the same hierarchy of footprints within a station type (Fig. S9) as well as the same seasonal characteristics for the different station types. The winter maximum of the ratio between $N_{50}$ and $N_{tot}$ is however less marked than in the case of $N_{100}$ at lowland sites other than polar, supporing the existence of an

additional source of particles larger than 100 nm in winter at these stations. The signature of traffic, which is a permanent source of sub-100 nm, and in particular, sub-50 nm particles (e.g. Pey et al., 2009), is again visible at urban sites, with the $75^{th}$ percentile of the ratio between $N_{50}$ and $N_{tot}$ being lower than for the other sites. The stronger connection between $N_{50}$ and $N_{tot}$ is also reflected in the higher coefficients of determination associated with the linear fits (Tables 2 and S3).

A feature common to all types of sites is the almost constant contribution of $N_{100}$ and $N_{50}$ over the whole range of $N_{tot}$ in winter

and fall, reflected by the slopes close to 1 obtained for all the corresponding fits (slopes between 0.86 and 1.05 for $N_{100}$, and between 0.92 and 1.03 for $N_{50}$, see Tables 2 and S3, respectively). For all the lowland sites, the contribution of $N_{100}$ to $N_{tot}$ is generally lower for the highest $N_{tot}$ values in spring (slopes between 0.64 and 0.78), with the strongest contrast observed for the polar sites. This is also the case in summer for lowland stations other than polar, and is probably related to the more important contribution to $N_{tot}$ of small particles originating from NPF, particularly favoured during these seasons (Nieminen

et al., 2018). Logically, the same trend is observed for $N_{50}$, but in a less marked way (slopes between 0.70 and 0.90 at lowland sites during spring), since the probability that NPF particles contribute to $N_{50}$ is higher than $N_{100}$. The fits obtained for polar stations in summer indicate a behaviour close to that described for the colder months, with slopes approaching 1 (0.91 for $N_{100}$ and 0.99 for $N_{50}$), and this is the case as well for mountain sites, where, both during spring and summer, the slopes of the corresponding fits are even closer to 1 than during winter and fall (0.94 and 0.97 for $N_{100}$, for spring and summer respectively,

0.95 and 0.94 for $N_{50}$).

This last analysis, focused on the largest particles of the spectrum, makes it possible to obtain an estimate of the concentration of potential CCN based solely on the knowledge of the PNSD. According to the previous results, an estimate of the CCN-sized particle concentration may even be deduced from the knowledge of $N_{tot}$ only in some seasons, when the contributions of $N_{100}$ and $N_{50}$ are observed to be constant over the whole range of $N_{tot}$. However, while such simple approach assuming that all





particles larger than a given activation diameter are potential CCN was reported to lead to reasonable results at JFJ (Jurányi et al., 2011), a more precise analysis would require information on the hygroscopicity of the sampled particles for each site, which probably varies seasonally according to the nature of the particles, since it will impact their activation diameter (Schmale et al., 2018).

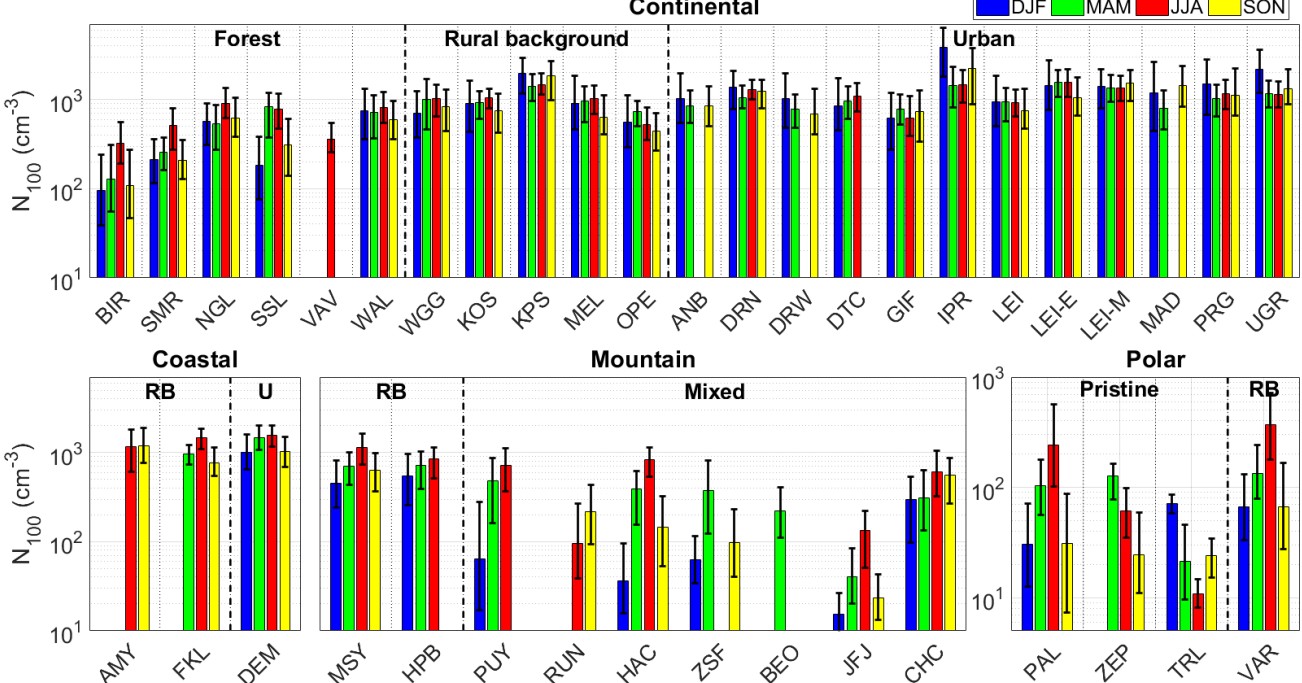

Fig. 12 Seasonal statistics of $N_{100}$, the particle number concentration in the range 100-500 nm, used as a proxy for potential CCN population. The bars represent the median of $N_{100}$, and the lower and upper end of the error bars represent the 1st and 3rd quartile of the data, respectively. Stations are sorted based on the classification reported in Table 1. The meaning of the abbreviations used for the footprint is the following: RB for rural background and U for urban.



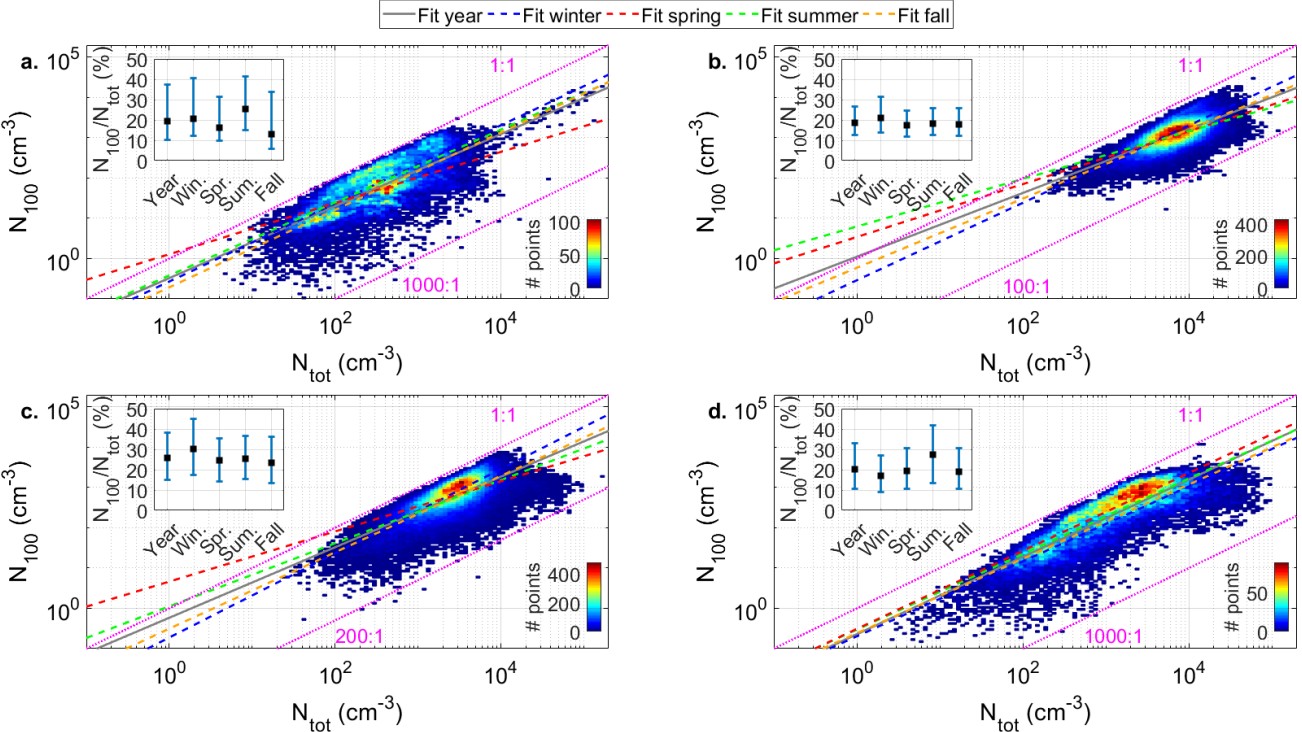

Fig. 13 Scatter plots of $N_{100}$ as a function of $N_{tot}$ (hourly averages) for the different station types: a. polar sites, b. urban sites, c. other lowland sites and d. mountain sites. The color of each pixel indicates the number of data points (hourly averages) falling into its area (all pixels have equal area on a log-log scale). The linear fit performed on the logarithm of the data, separately for each period (year and seasons), is also presented. The statistics of the ratio between $N_{100}$ and $N_{tot}$ calculated for each or these periods are in addition shown for each station type in the insert of the corresponding panel; the markers represent the median of the ratios, and the lower and upper limits of the error bars indicate the 1[st] and 3[rd] quartile, respectively.



Table 2 Connection between $N_{100}$, the particle number concentration in the range 100-500 nm, used as a proxy for the CCN population, and $N_{tot}$. For each station type and season, the equation of the linear fit performed on the logarithm of the data is reported in the second column, and the corresponding coefficient of determination in the third column. Note that based on corresponding p-values, all correlations were found significant at 95% confidence level ($p < 0.05$).

| Station type / season | Fit equation | $R^2$ |
|---|---|---|
| **Polar sites** | | |
| Year | $\log_{10}(N_{100}) = 0.89 \times \log_{10}(N_{tot}) - 0.48$ | 0.59 |
| Winter | $\log_{10}(N_{100}) = 0.97 \times \log_{10}(N_{tot}) - 0.57$ | 0.66 |
| Spring | $\log_{10}(N_{100}) = 0.64 \times \log_{10}(N_{tot}) + 0.11$ | 0.30 |
| Summer | $\log_{10}(N_{100}) = 0.91 \times \log_{10}(N_{tot}) - 0.42$ | 0.46 |
| Fall | $\log_{10}(N_{100}) = 0.96 \times \log_{10}(N_{tot}) - 0.71$ | 0.53 |
| **Urban sites** | | |
| Year | $\log_{10}(N_{100}) = 0.79 \times \log_{10}(N_{tot}) + 0.05$ | 0.50 |
| Winter | $\log_{10}(N_{100}) = 0.96 \times \log_{10}(N_{tot}) - 0.54$ | 0.64 |
| Spring | $\log_{10}(N_{100}) = 0.66 \times \log_{10}(N_{tot}) + 0.54$ | 0.42 |
| Summer | $\log_{10}(N_{100}) = 0.59 \times \log_{10}(N_{tot}) + 0.80$ | 0.38 |
| Fall | $\log_{10}(N_{100}) = 0.86 \times \log_{10}(N_{tot}) - 0.23$ | 0.53 |
| **Other lowland sites** | | |
| Year | $\log_{10}(N_{100}) = 0.88 \times \log_{10}(N_{tot}) - 0.23$ | 0.53 |
| Winter | $\log_{10}(N_{100}) = 1.05 \times \log_{10}(N_{tot}) - 0.73$ | 0.67 |
| Spring | $\log_{10}(N_{100}) = 0.78 \times \log_{10}(N_{tot}) + 0.05$ | 0.44 |
| Summer | $\log_{10}(N_{100}) = 0.62 \times \log_{10}(N_{tot}) + 0.66$ | 0.31 |
| Fall | $\log_{10}(N_{100}) = 0.95 \times \log_{10}(N_{tot}) - 0.51$ | 0.55 |
| **Mountain sites** | | |
| Year | $\log_{10}(N_{100}) = 0.96 \times \log_{10}(N_{tot}) - 0.64$ | 0.72 |
| Winter | $\log_{10}(N_{100}) = 0.93 \times \log_{10}(N_{tot}) - 0.70$ | 0.77 |
| Spring | $\log_{10}(N_{100}) = 0.94 \times \log_{10}(N_{tot}) - 0.53$ | 0.74 |
| Summer | $\log_{10}(N_{100}) = 0.97 \times \log_{10}(N_{tot}) - 0.51$ | 0.69 |
| Fall | $\log_{10}(N_{100}) = 0.92 \times \log_{10}(N_{tot}) - 0.62$ | 0.72 |



## 8. Summary and conclusion

This study, based on data collected at 62 sites around the world, provides the most up-to-date picture of the spatial distribution of aerosol concentration and particle number size distribution. Specifically, 38 more stations than previously considered in Asmi et al. (2011) were included, and all WMO regions were covered. However, as noted earlier in Laj et al. (2020), there is a strong bias in the world data coverage, with a majority of stations located in Europe (39 sites) and North America (10), and a lack of observations in other regions, in particular in Africa (2), Asia (4) and South America (1). Analysis of the spatial distribution of the sites in relation to their classification also reveals certain limitations. For instance, all urban stations are located in Europe, and there is a clear lack of data on deserts; considering oceans cover >70% of Earth, it can certainly be considered that there is a lack of marine observations as well. A final bias concerns the type of data collected at these sites, with most of the MPSS allowing PNSD monitoring located in Europe (34 sites out of 39) while elsewhere CPC is the dominant instrument.

The first objective of this study was to assess the impact of data availability on $N_{tot}$'s annual and seasonal statistics (median, $25^{th}$ and $75^{th}$ percentiles), in order to determine a threshold for a reasonable compromise between the number of statistics included and their quality. To do this, the absence of data was simulated in the $N_{tot}$ time series of the stations with data availability greater than 95% over the year (11 sites). It appears that the lack of individual hourly averages has, for comparable coverage, less impact on the statistics than long periods of missing data. However, although there are differences from one station to another, in particular with a more pronounced effect at polar sites, and also from one season to another, it appears overall that seasonal statistics are only slightly impacted when the corresponding data availability remains above 50% in the reduced data sets. At the annual level, a slightly higher coverage, 60%, is necessary to maintain the representativeness of the statistics. An availability of 75% year-round was required for the study of the diel cycle of $N_{tot}$, which appears to be more sensitive to the data coverage, and also to missing individual data points (as opposed to long consecutive data gaps).

Firstly, the analysis of $N_{tot}$ reveals few common behaviours amongst all sites. In particular, it appears that higher concentrations are often observed in spring and summer, as a likely result of enhanced emission sources and/or favoured formation processes (in connection with ABL dynamics at mountain stations), and possibly, reduced particle sinks at some sites. Also, the first log-normal mode fitted to the PNSD, which is a combination of the usual nucleation and Aitken modes, is wider than the second (accumulation) mode at all sites, and most of the time dominates the distribution. With the exception of polar sites, where the characteristics of the Aitken mode show a particularly pronounced variability, the concentration of this first mode is also less variable from one season to another than that of the second mode; its location is in contrast more variable for all station types except urban. Beyond these common features, however, there are notable differences among sites. Among other factors (including the nature and the proximity of the aerosol sources), the level of anthropogenic influence seems to strongly impact the observations, and contributes significantly to the contrasting patterns observed for the different station types:





1.  The lowest concentrations, on the order of $10^2$ cm$^{-3}$, are observed at polar sites, but with significant annual variability resulting from both marked seasonal contrasts and significant intra-seasonal variability at some sites. The PNSD is mostly bimodal, especially in the Northern Hemisphere, but also shows a strong seasonal contrast and reflects the specificities of each site (e.g. impact of Arctic haze on summer measurements at ZEP). The diel cycle is, on average, weak at polar sites, probably as a consequence of the absence of a regular day-night cycle in some seasons, and also because there is no strong anthropogenic activity likely to influence it in these pristine environments.

2.  In contrast to the polar sites, stations located in urban areas, both continental and coastal, exhibit the highest $N_{tot}$, with yearly medians in the range $10^3$-$10^4$ cm$^{-3}$. Due to limited intra-seasonal variability and low seasonal contrast, the variability of $N_{tot}$ is overall less pronounced at these sites. The weak seasonality of $N_{tot}$ is associated with minimal changes of the PNSD, which are almost unimodal throughout the year and shifted towards the lower end of the investigated size range at a majority of stations, with elevated concentrations of sub-100 nm particles. In contrast, the diel cycle of $N_{tot}$ is marked for these sites, reflecting the significant impact of anthropogenic activities on the measurements.

3.  The remaining sites, including mountain and non-urban continental and coastal stations, do not exhibit as clear a common behaviour as polar and urban sites and display, on average, intermediate $N_{tot}$, with yearly medians of the order of $10^2$-$10^3$ cm$^{-3}$. Particle concentrations measured at mountain sites tend to be lower compared to nearby lowland sites, with more pronounced seasonal variations, but there is overall little difference between the geographical categories. The signature of the dominant footprints is in contrast more pronounced, with lower concentrations measured in forested areas or at stations influenced by air masses of various origins ("mixed") compared to rural background sites.

    3.1 Particle number concentrations measured at non-urban continental and coastal sites are overall lower compared to those observed at urban stations, but exhibit comparable variability, as a result of both limited intra- and inter seasonal variability. The stations located in forested areas, however, show more pronounced variations, and are also distinguished by the shape of their PNSD, which tend to have a more pronounced bimodal behaviour compared to rural background stations. The modes representative of the distributions measured at non-urban sites peak at the largest diameters among all station types, with the most important shift to larger diameters being observed at coastal sites (AMY and FKL). The diel cycle of $N_{tot}$ is overall less marked at these sites compared to urban stations.

    3.2 Observations from mountain stations are influenced by the site topography and environs, which, coupled with the variations of the ABL height, largely explain the significant intra- and inter seasonal contrasts observed at



these sites, as well as the pronounced diel cycle of $N_{tot}$. The PNSD measured at mountain sites exhibit a stronger bimodal behaviour compared to lowland stations (other than polar), but with noticeable differences from site to site. Features comparable to those of lowland rural background continental sites are observed for the two mountain stations located below 1000 m a.s.l. (MSY and HPB).

Furthermore, the specific analysis of the CCN-sized particle number concentration (i.e. > 50 – 100 nm, referred to as $N_{50}$ and $N_{100}$) indicates that these particles of climatic importance can represent between a few percent and almost all of $N_{tot}$, with seasonal medians of the order of ~10 to 1000 cm$^{-3}$ depending on the site and season. The trends observed for $N_{50}$ and $N_{100}$, including the classification of the station types according to concentration levels and the existence of seasonal contrasts, are

overall similar to those observed for $N_{tot}$. Slight differences are however observed, particularly in the magnitude of the contrasts, due to the variability of the contributions of $N_{100}$ and $N_{50}$ to $N_{tot}$, itself tightly connected to the variability of the particle sources in the different seasons.

By comparing and contrasting observations that characterize the different station types, this study shows the importance of collecting data in various environments, and therefore highlights the need to increase the monitoring spatial coverage in certain

regions and / or environments in the future. The need for harmonized protocols for data acquisition and quality control, as well as ease of access and availability, clearly indicates the interest in developing these observations within networks and/or distributed research infrastructures. Operating in the context of a network may also promote the sustainability of the observations, necessary to capture the seasonal contrasts characteristic of certain station types, or, more importantly, for the evaluation of long-term trends. Such a trend study of $N_{tot}$ will be carried out for the sites with sufficiently long time series (>

10 years) and reported in a separate paper.

The results of this study, which cover a variety of environments across all WMO regions, also provide a valuable, freely available and easy to use support to the modeling community to perform model comparison and validation with respect to particle number concentration and size distribution. A sufficiently accurate description of these aerosol properties is, in particular, a crucial step towards an improved representation of aerosol-cloud interactions in models, and therefore, better

evaluation of their effect on climate.

## Appendix A

Table A1. Parameters of the modes identified for the description of the median particle number size distributions measured at the stations equipped with a MPSS. $N_m$, $\sigma_m$ and $D_m$ are the number concentration, the geometric standard deviation and the

geometric mean dry diameter of the mode, respectively. R² is the coefficient of determination between observed and fitted size distributions. The results are reported separately for each season.





a)   DJF

| Station | Mode 1 | | | Mode 2 | | | R² |
|---|---|---|---|---|---|---|---|
| | $N_{m,1}$ | $\sigma_{m,1}$ | $D_{m,1}$ | $N_{m,2}$ | $\sigma_{m,2}$ | $D_{m,2}$ | |
| RUN | - | - | - | - | - | - | - |
| WGG | 2130 | 1.92 | 53 | 306 | 1.44 | 194 | 1.00 |
| AMY | - | - | - | - | - | - | - |
| CHC | 893 | 1.77 | 40 | 278 | 1.45 | 145 | 1.00 |
| ANB | 4498 | 3.00 | 20 | 1370 | 1.94 | 105 | 1.00 |
| BIR | 356 | 1.90 | 40 | 76 | 1.45 | 142 | 1.00 |
| BEO | - | - | - | - | - | - | - |
| DEM | 5227 | 2.28 | 45 | 159 | 1.28 | 190 | 1.00 |
| DRN | 2018 | 1.52 | 20 | 4673 | 2.36 | 63 | 1.00 |
| DRW | 3130 | 2.81 | 20 | 1729 | 2.11 | 98 | 1.00 |
| DTC | 2670 | 2.61 | 59 | 114 | 1.30 | 218 | 1.00 |
| FKL | - | - | - | - | - | - | - |
| GIF | 1333 | 2.05 | 40 | 1128 | 1.79 | 89 | 1.00 |
| HAC | 280 | 1.55 | 35 | 39 | 1.45 | 138 | 0.99 |
| HPB | 1465 | 2.46 | 36 | 500 | 1.67 | 136 | 1.00 |
| IPR | 4170 | 2.34 | 49 | 5301 | 1.68 | 109 | 1.00 |
| JFJ | 101 | 2.08 | 27 | 15 | 1.45 | 124 | 1.00 |
| KOS | 1351 | 2.16 | 60 | 667 | 1.65 | 173 | 1.00 |
| KPS | 1712 | 1.97 | 48 | 2450 | 1.71 | 134 | 1.00 |
| LEI | 3951 | 3.00 | 20 | 1307 | 2.14 | 114 | 0.99 |
| LEI-E | 5904 | 2.82 | 21 | 2168 | 2.17 | 99 | 1.00 |
| LEI-M | 2103 | 1.46 | 20 | 6179 | 2.80 | 47 | 1.00 |
| MAD | 9378 | 2.01 | 20 | 2659 | 1.84 | 87 | 1.00 |
| MEL | 2837 | 2.84 | 30 | 1043 | 2.16 | 123 | 1.00 |
| MSY | 1537 | 2.33 | 47 | 175 | 1.41 | 168 | 1.00 |
| NGL | 1057 | 2.03 | 54 | 404 | 1.65 | 202 | 1.00 |
| OPE | 1692 | 2.19 | 62 | 113 | 1.40 | 175 | 1.00 |
| PAL | 74 | 1.71 | 35 | 32 | 1.59 | 142 | 1.00 |
| PRG | 3436 | 2.58 | 26 | 2000 | 1.86 | 118 | 1.00 |
| PUY | 630 | 2.24 | 26 | 38 | 1.47 | 146 | 1.00 |
| SMR | 478 | 1.80 | 43 | 173 | 1.41 | 180 | 0.99 |
| SSL | 645 | 2.27 | 40 | 104 | 1.42 | 165 | 1.00 |
| UGR | 10443 | 2.75 | 30 | 1986 | 1.55 | 99 | 1.00 |
| VAR | 76 | 1.69 | 44 | 66 | 1.51 | 175 | 0.99 |
| VAV | - | - | - | - | - | - | - |
| WAL | 2150 | 2.70 | 50 | 238 | 1.50 | 236 | 1.00 |
| ZEP | - | - | - | - | - | - | - |
| ZSF | 326 | 1.84 | 27 | 92 | 1.60 | 111 | 1.00 |
| TRL | 374 | 1.81 | 40 | 100 | 1.52 | 98 | 1.00 |





b) MAM

| Station | Mode 1 | | | Mode 2 | | | $R^2$ |
|---|---|---|---|---|---|---|---|
| | $N_{m,1}$ | $\sigma_{m,1}$ | $D_{m,1}$ | $N_{m,2}$ | $\sigma_{m,2}$ | $D_{m,2}$ | |
| RUN | - | - | - | - | - | - | - |
| WGG | 2214 | 1.93 | 58 | 434 | 1.48 | 180 | 1.00 |
| AMY | - | - | - | - | - | - | - |
| CHC | 1557 | 1.73 | 39 | 229 | 1.40 | 146 | 1.00 |
| ANB | 4500 | 2.77 | 30 | 328 | 1.47 | 170 | 1.00 |
| BIR | 656 | 1.78 | 34 | 118 | 1.45 | 144 | 1.00 |
| BEO | 401 | 1.72 | 52 | 174 | 1.49 | 203 | 1.00 |
| DEM | 6629 | 2.71 | 20 | 2952 | 2.01 | 81 | 1.00 |
| DRN | 7463 | 2.77 | 24 | 711 | 1.80 | 127 | 1.00 |
| DRW | 4238 | 2.84 | 32 | 181 | 1.35 | 182 | 1.00 |
| DTC | 2927 | 2.32 | 47 | 386 | 1.46 | 189 | 1.00 |
| FKL | 2207 | 2.00 | 76 | 187 | 1.32 | 204 | 1.00 |
| GIF | 2931 | 2.15 | 42 | 496 | 1.58 | 144 | 1.00 |
| HAC | 691 | 1.81 | 46 | 347 | 1.48 | 156 | 1.00 |
| HPB | 1938 | 2.03 | 38 | 653 | 1.58 | 150 | 1.00 |
| IPR | 3799 | 2.27 | 38 | 1750 | 1.83 | 107 | 1.00 |
| JFJ | 156 | 2.08 | 36 | 34 | 1.49 | 134 | 1.00 |
| KOS | 1870 | 1.98 | 49 | 653 | 1.52 | 182 | 1.00 |
| KPS | 3557 | 2.13 | 55 | 699 | 1.47 | 161 | 1.00 |
| LEI | 4680 | 2.92 | 34 | 168 | 1.32 | 187 | 1.00 |
| LEI-E | 8433 | 2.69 | 32 | 855 | 1.93 | 123 | 1.00 |
| LEI-M | 11227 | 3.00 | 20 | 803 | 1.89 | 133 | 1.00 |
| MAD | 7690 | 1.90 | 20 | 1819 | 1.81 | 87 | 1.00 |
| MEL | 3619 | 2.32 | 45 | 290 | 1.39 | 192 | 1.00 |
| MSY | 2439 | 1.96 | 40 | 630 | 1.60 | 133 | 1.00 |
| NGL | 1330 | 1.80 | 45 | 475 | 1.62 | 152 | 1.00 |
| OPE | 2620 | 2.11 | 51 | 286 | 1.42 | 154 | 1.00 |
| PAL | 209 | 1.61 | 33 | 102 | 1.50 | 159 | 0.99 |
| PRG | 4275 | 2.67 | 31 | 682 | 1.63 | 143 | 1.00 |
| PUY | 1091 | 1.87 | 36 | 540 | 1.59 | 136 | 0.99 |
| SMR | 928 | 1.80 | 38 | 200 | 1.43 | 170 | 0.99 |
| SSL | 2085 | 2.12 | 45 | 547 | 1.54 | 166 | 1.00 |
| UGR | 7615 | 2.46 | 33 | 414 | 1.40 | 128 | 1.00 |
| VAR | 248 | 1.77 | 34 | 129 | 1.46 | 168 | 0.98 |
| VAV | - | - | - | - | - | - | - |
| WAL | 2778 | 2.14 | 42 | 331 | 1.47 | 187 | 1.00 |
| ZEP | 68 | 2.13 | 50 | 116 | 1.45 | 185 | 0.99 |
| ZSF | 759 | 1.93 | 44 | 332 | 1.50 | 152 | 1.00 |
| TRL | 93 | 1.65 | 29 | 48 | 1.47 | 95 | 0.99 |

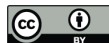



c) JJA

| Station | Mode 1 | | | Mode 2 | | | R² |
|---|---|---|---|---|---|---|---|
| | $N_{m,1}$ | $\sigma_{m,1}$ | $D_{m,1}$ | $N_{m,2}$ | $\sigma_{m,2}$ | $D_{m,2}$ | |
| RUN | 424 | 1.65 | 32 | 121 | 1.54 | 125 | 1.00 |
| WGG | 1703 | 2.21 | 48 | 1070 | 1.78 | 130 | 1.00 |
| AMY | 2600 | 1.70 | 85 | 110 | 1.25 | 250 | 1.00 |
| CHC | 3042 | 1.72 | 23 | 1879 | 1.88 | 73 | 1.00 |
| ANB | - | - | - | - | - | - | - |
| BIR | 1399 | 1.77 | 50 | 147 | 1.34 | 188 | 1.00 |
| BEO | - | - | - | - | - | - | - |
| DEM | 3519 | 2.46 | 46 | 1244 | 1.67 | 130 | 1.00 |
| DRN | 7512 | 2.44 | 30 | 1168 | 1.82 | 107 | 1.00 |
| DRW | - | - | - | - | - | - | - |
| DTC | 2881 | 1.89 | 48 | 829 | 1.51 | 151 | 1.00 |
| FKL | 2773 | 1.89 | 89 | 236 | 1.31 | 207 | 1.00 |
| GIF | 2737 | 2.07 | 32 | 752 | 1.65 | 117 | 1.00 |
| HAC | 1045 | 2.12 | 68 | 568 | 1.52 | 169 | 0.99 |
| HPB | 2118 | 2.11 | 45 | 599 | 1.50 | 146 | 1.00 |
| IPR | 4151 | 2.25 | 50 | 736 | 1.51 | 142 | 1.00 |
| JFJ | 215 | 1.78 | 48 | 137 | 1.53 | 136 | 1.00 |
| KOS | 2114 | 1.84 | 50 | 886 | 1.53 | 154 | 1.00 |
| KPS | 3736 | 2.06 | 62 | 581 | 1.43 | 155 | 1.00 |
| LEI | 4687 | 2.49 | 42 | 145 | 1.28 | 171 | 1.00 |
| LEI-E | 6065 | 2.98 | 20 | 4550 | 2.32 | 56 | 1.00 |
| LEI-M | 7363 | 2.73 | 20 | 2436 | 2.07 | 80 | 1.00 |
| MAD | - | - | - | - | - | - | - |
| MEL | 4109 | 2.21 | 50 | 223 | 1.32 | 181 | 1.00 |
| MSY | 2314 | 1.80 | 43 | 1370 | 1.66 | 128 | 1.00 |
| NGL | 2691 | 1.80 | 59 | 421 | 1.41 | 170 | 1.00 |
| OPE | 1507 | 1.81 | 37 | 794 | 1.61 | 108 | 1.00 |
| PAL | 528 | 1.74 | 69 | 93 | 1.34 | 200 | 0.99 |
| PRG | 5671 | 2.46 | 29 | 1200 | 1.69 | 110 | 1.00 |
| PUY | 1617 | 2.13 | 38 | 715 | 1.63 | 134 | 0.99 |
| SMR | 1400 | 1.82 | 73 | 90 | 1.38 | 185 | 0.99 |
| SSL | 1872 | 2.18 | 45 | 547 | 1.55 | 149 | 1.00 |
| UGR | 3325 | 1.80 | 32 | 1813 | 1.63 | 108 | 1.00 |
| VAR | 737 | 1.73 | 72 | 144 | 1.36 | 199 | 1.00 |
| VAV | 1416 | 1.88 | 52 | 139 | 1.36 | 192 | 0.98 |
| WAL | 3129 | 1.95 | 52 | 280 | 1.37 | 187 | 1.00 |
| ZEP | 128 | 1.52 | 34 | 77 | 1.52 | 134 | 1.00 |
| ZSF | - | - | - | - | - | - | - |
| TRL | 32 | 1.70 | 26 | 19 | 1.53 | 95 | 1.00 |





d) SON

| Station | Mode 1 | | | Mode 2 | | | $R^2$ |
|---|---|---|---|---|---|---|---|
| | $N_{m,1}$ | $\sigma_{m,1}$ | $D_{m,1}$ | $N_{m,2}$ | $\sigma_{m,2}$ | $D_{m,2}$ | |
| RUN | 330 | 1.90 | 43 | 195 | 1.46 | 159 | 1.00 |
| WGG | 2322 | 2.21 | 42 | 509 | 1.53 | 162 | 1.00 |
| AMY | 3385 | 1.97 | 73 | 74 | 1.25 | 209 | 1.00 |
| CHC | 1587 | 1.89 | 39 | 546 | 1.67 | 148 | 1.00 |
| ANB | 3479 | 2.65 | 31 | 693 | 1.81 | 132 | 1.00 |
| BIR | 629 | 1.79 | 42 | 62 | 1.34 | 166 | 0.99 |
| BEO | - | - | - | - | - | - | - |
| DEM | 5310 | 3.00 | 32 | 617 | 1.83 | 81 | 1.00 |
| DRN | 7781 | 2.68 | 20 | 1888 | 1.95 | 90 | 1.00 |
| DRW | 4005 | 2.71 | 35 | 118 | 1.35 | 167 | 1.00 |
| DTC | - | - | - | - | - | - | - |
| FKL | 1677 | 1.86 | 68 | 306 | 1.35 | 204 | 1.00 |
| GIF | 2550 | 2.21 | 42 | 526 | 1.55 | 130 | 1.00 |
| HAC | 414 | 1.82 | 44 | 114 | 1.44 | 164 | 1.00 |
| HPB | - | - | - | - | - | - | - |
| IPR | 4177 | 2.09 | 52 | 2329 | 1.74 | 113 | 1.00 |
| JFJ | 141 | 2.08 | 31 | 20 | 1.40 | 121 | 1.00 |
| KOS | 1384 | 1.94 | 48 | 615 | 1.60 | 168 | 1.00 |
| KPS | 1612 | 1.74 | 40 | 2800 | 1.70 | 117 | 1.00 |
| LEI | 4750 | 2.88 | 21 | 795 | 2.03 | 109 | 1.00 |
| LEI-E | 7312 | 2.80 | 31 | 162 | 1.45 | 176 | 1.00 |
| LEI-M | 9903 | 2.58 | 20 | 1969 | 1.96 | 108 | 1.00 |
| MAD | 7847 | 2.11 | 32 | 1494 | 1.68 | 116 | 1.00 |
| MEL | 3956 | 2.74 | 34 | 97 | 1.30 | 198 | 1.00 |
| MSY | 2721 | 2.25 | 49 | 108 | 1.32 | 187 | 1.00 |
| NGL | 2171 | 1.90 | 49 | 316 | 1.43 | 190 | 1.00 |
| OPE | 1570 | 2.28 | 50 | 155 | 1.47 | 157 | 1.00 |
| PAL | 106 | 1.60 | 45 | 22 | 1.37 | 170 | 0.98 |
| PRG | 4907 | 2.69 | 24 | 1465 | 1.88 | 107 | 1.00 |
| PUY | - | - | - | - | - | - | - |
| SMR | 609 | 1.77 | 52 | 129 | 1.36 | 192 | 0.98 |
| SSL | 1178 | 2.31 | 40 | 175 | 1.48 | 151 | 1.00 |
| UGR | 6856 | 2.31 | 40 | 527 | 1.51 | 130 | 1.00 |
| VAR | 102 | 1.62 | 46 | 59 | 1.43 | 175 | 0.99 |
| VAV | - | - | - | - | - | - | - |
| WAL | 2666 | 2.31 | 40 | 213 | 1.39 | 209 | 1.00 |
| ZEP | 18 | 1.51 | 35 | 30 | 1.62 | 134 | 0.99 |
| ZSF | 362 | 1.66 | 32 | 136 | 1.55 | 117 | 1.00 |
| TRL | 270 | 1.63 | 30 | 37 | 1.47 | 104 | 1.00 |



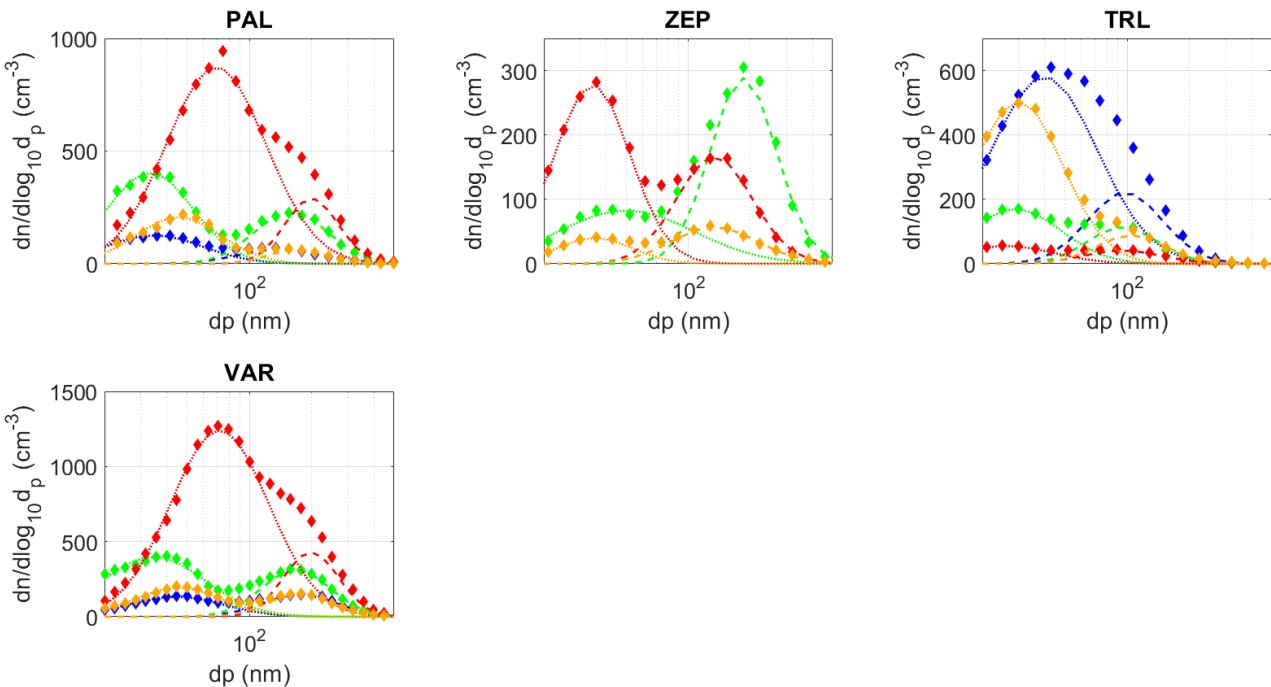

Fig. A1 Median particle size distributions (diamonds) and corresponding modes (dotted lines) at polar sites. Blue lines and markers correspond to DJF, green lines and markers to MAM, red lines and markers to JJA and orange lines and markers to SON.





Fig. A2 Median particle size distributions (diamonds) and corresponding modes (dotted lines) at urban continental sites. See Fig. A1 for an explanation of the symbols.





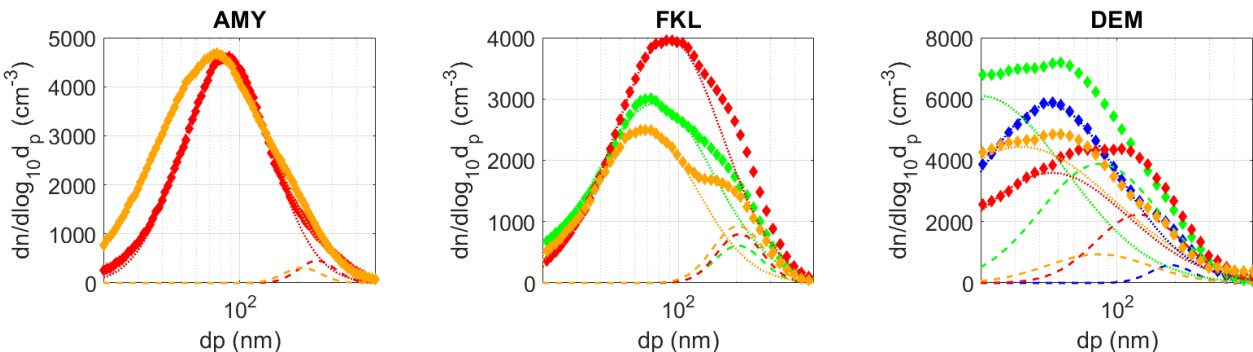

10    Fig. A3 Median particle size distributions (diamonds) and corresponding modes (dotted lines) at coastal sites. See Fig. A1 for
an explanation of the symbols.

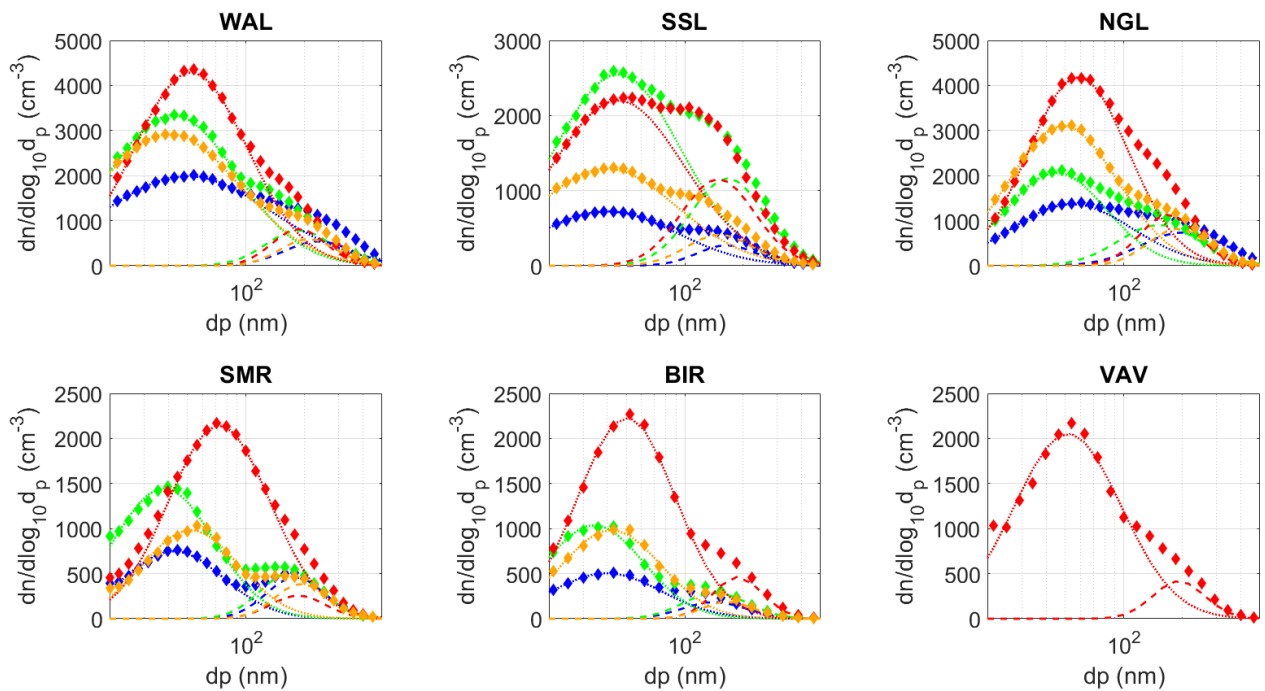

Fig. A4 Median particle size distributions (diamonds) and corresponding modes (dotted lines) at forest continental sites. See

30    Fig. A1 for an explanation of the symbols.





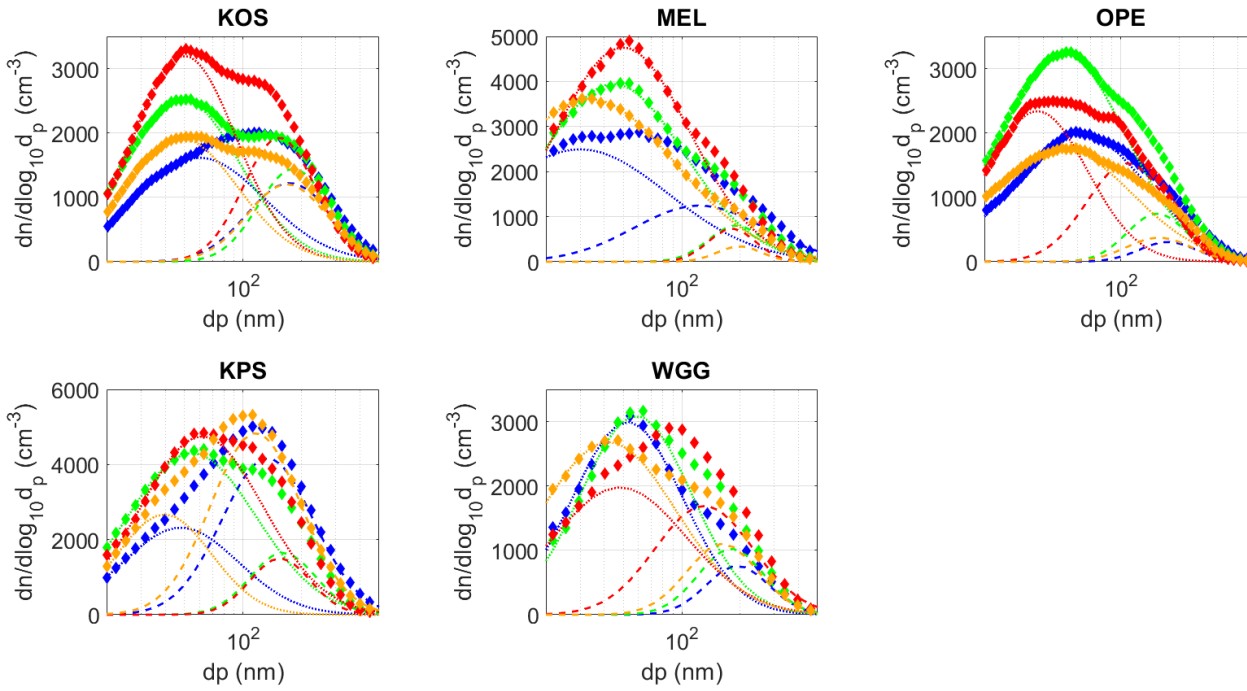

Fig. A5 Median particle size distributions (diamonds) and corresponding modes (dotted lines) at rural background continental sites. See Fig. A1 for an explanation of the symbols.

Fig. A6 Median particle size distributions (diamonds) and corresponding modes (dotted lines) at mountain sites. See Fig. A1 for an explanation of the symbols.





**Data availability**

Data will be made freely available under a specific doi after final publication of the paper.

**Author contributions**

CR, MCC, EA and PL defined the concept and methodology of the paper. Curation of data was done by CLM, YL, and MF as

part of the duties of the World Data Center for Aerosol. IB provided support in data analysis, and the formal analysis of datasets
was done by CR, MCC and EA. The original draft was written by CR, while EA, MCC, PL, KS, JPP, MG, OF, JPB, VV, EAB,
JACV, MS, SWK, JS, HL, YL and IB  participated in the scientific discussion and contributed to review and edit the paper.
All the authors have contributed to the necessary funding for the provision of data used in the paper.

**Competing interest**

The authors declare they have no competing interest.

**Acknowledgements**

NOAA base funding supports the following observatories: BRW, BND, MLO, SMO, SPO, and THD, where efforts of the
dedicated observatory staff and of programmer Derek Hageman are appreciated.

Measurements at Welgegund are supported by North-West University, the University of Helsinki, and the Finnish

Meteorological Institute. This publication also forms part of the output of the Biogeochemistry Research Infrastructure
Platform (BIOGRIP) of the Department of Science and Innovation of South Africa.

Pallas and SMEAR II are grateful for the support of the Academy of Finland Centre of Excellence programme (project no.
272041), the Academy of Finland project Greenhouse gas, aerosol and albedo variations in the changing Arctic (project no.
269095), and the Novel Assessment of Black Carbon in the Eurasian Arctic: From Historical Concentrations and Sources to

Future Climate Impacts (NABCEA, project no. 296302).

Aerosol measurements at Anmyon-do were supported by Korea Meteorological Administration Research and Development
Program "Development of Monitoring and Analysis Techniques for Atmospheric Composition in Korea" under grant
KMA2018-00522. Measurements at Gosan were supported by the National Research Foundation of Korea
(2017R1D1A1B06032548) and the Korea Meteorological Administration Research and Development Program under grant

KMI2018-01111.

The Lulin station is operated under the grants funded by the Taiwan Environmental Protection Administration.

WLG is supported by China Meterological Administration, where efforts of the dedicated observatory staff are appreciated.



Sites PDM, PUY, GIF, CHC, and RUN are partially operated with the support of CNRS-INSU under the long-term observation programme and the French Ministry for Research under the ACTRIS-FR national research infrastructure. PDM and GIF received specific support of the French Ministry of the Environment. ATMO Occitanie is mentioned for sampling operations at PDM. Measurements at SIRTA are hosted by CNRS and by the alternative energies and atomic energy commission (CEA)

with additional contributions from the French Ministry of the Environment through its funding to the reference laboratory for air quality monitoring (LCSQA). PUY is grateful for support from ATMO Auvergne Rhône Alpes for sampling operations and the support from the personnel of the Observatoire de Physique du Globe de Clermont-Ferrand (OPGC). The specific support of the Institut de Recherche et Développement (IRD) in France and the Universidad Mayor de San Andres in Bolivia support operations at CHC operations.

The Steamboat Ski Resort provided logistical support and in-kind donations for SPL. The Desert Research Institute is a permittee of the Medicine-Bow Routt National Forests and an equal opportunity service provider and employer. SPL appreciates the extensive assistance of the NOAA/ESRL Federated Aerosol Network, of Ian McCubbin, site manager of SPL, and of Ty Atkins, Joe Messina, Dan Gilchrist, and Maria Garcia, who provided technical assistance with the maintenance and data quality control for the aerosol instruments.

SGP measurements/mentorship were supported by DOE-7F-30118 and staff onsite.

Cape Grim Baseline Air Pollution Monitoring Station is grateful to the Australian Bureau of Meteorology for their long-term and continued support and all the staff from the Bureau of Meteorology and CSIRO, who have contributed to the generation of records reported here.

The aerosol measurements at the Jungfraujoch were conducted with financial support from MeteoSwiss (GAW-CH aerosol

monitoring programme) and from the European Union as well as the Swiss State Secretariat for Education, Research and Innovation (SERI) for the European Research Infrastructure for the observation of Aerosol, Clouds and Trace Gases (ACTRIS). The International Foundation High Altitude Research Station Jungfraujoch and Gornergrat (HFSJG) is mentioned for providing the research platform at the Jungfraujoch.

The aerosol measurements at Kosetice received funding from the European Union's Horizon 2020 research and innovation

programme under grant agreement no. 654109 and from the project for support of national research infrastructure ACTRIS – participation of the Czech Republic (ACTRIS-CZ – LM2015037) supported by the Ministry of Education, Youth and Sports of CR within National Sustainability Program I (NPU I), grant agreement no. LO1415. The measurements were also supported by ERDF "ACTRIS-CZ RI" (no. CZ.02.1.01/0.0/0.0/16_013/0001315).

Measurements at the Madrid site were funded by the following projects: CRISOL (CGL2017–85344-R

MINECO/AEI/FEDER, UE), TIGAS-CM (Madrid Regional Government Y2018/EMT5177), AIRTEC-CM (Madrid Regional Government P2018/EMT4329), REDMAAS2020 (RED2018-102594- T CIENCIA) and Red de Excelencia ACTRIS-ESPAÑA (CGL2017-90884-REDT). Measurements at Montsec and Montseny were supported by the Spanish Ministry of Economy, Industry and Competitiveness and FEDER funds under project HOUSE (CGL2016-78594-R), and by the Generalitat de Catalunya (AGAUR 2017 SGR41 and the DGQA). Aerosol measurements at El Arenosillo Observatory are



supported by the National Institute for Aerospace Technology and by different R&D projects of the Ministerio Español de Economía, Industria y Competitividad (MINECO). Aerosol measurements at UGR are supported by the Spanish Ministry of Economy and Competitiveness through projects no. CGL2016-81092-R, CGL2017-90884-REDT, RTI2018-097864-B-I00 and PGC2018-098770-B-I00, and by the Andalusia Regional Government through project no. P18-RT-3820.

FKL, HAC and DEM are grateful for funding by project PANhellenic infrastructure for Atmospheric Composition and climate change (MIS 5021516) which is implemented under action Reinforcement of the Research and Innovation Infrastructure, funded by operational programme Competitiveness, Entrepreneurship and Innovation (NSRF 2014-2020) and co-financed by Greece and the European Union (European Regional Development Fund).

CPC measurements at Sonnblick are supported by the Climate and Air Quality Commission of the Austrian Academy of
Sciences and the office of the provincial government Salzburg, Unit 5/02.

At CMN, aerosol measurements were partially supported by the Italian Ministry of Research and Education.

Measurements at Birkenes II are financed by the Norwegian Environment Agency.

VAV is grateful for various Swedish FORMAS, Swedish Research Council (VR) grants and the Magnus Bergvall and Märta och Erik Holmberg foundations and Swedish EPA for making the research possible at the VAV site.

NMY wishes to thank the many technicians and scientists of the Neumayer overwintering crews, whose outstanding commitment enabled continuous, high-quality aerosol records over many years.

Gunter Löschau is acknowledged for his contribution to the data acquisition at ANB, DTC and DRN.

**Financial support**

This research was supported by the European Commission's Horizon 2020 Framework Programme (ACTRIS2 (grant
agreement no. 654109)); the University of Helsinki; the Finnish Meteorological Institute; Department of Science and Innovation of South Africa; the Academy of Finland Centre of Excellence programme (project no. 272041); the Academy of Finland project Greenhouse gas, aerosol and albedo variations in the changing Arctic (project no. 269095); the Novel Assessment of Black Carbon in the Eurasian Arctic: From Historical Concentrations and Sources to Future Climate Impacts (NABCEA, project no. 296302); the Korea Meteorological Administration Research and Development Program
"Development of Monitoring and Analysis Techniques for Atmospheric Composition in Korea" (grant no. KMA2018-00522); the National Research Foundation of Korea (grant no. 2017R1D1A1B06032548); the Korea Meteorological Administration Research and Development Program (grant no. KMI2018-01111); the Taiwan Environmental Protection Administration; China Meteorological Administration; National Scientific Foundation of China (41675129, 41875147); National Key R&D Program of the Ministry of Science and Technology of the People's Republic of China (grant no. 2016YFC0203305 and
2018YFC0213204); Chinese Academy of Meteorological Sciences (2020KJ001); Innovation Team for Haze-fog Observation and Forecasts of MOST and CMA; CNRS-INSU; French Ministry for Research under the ACTRIS-FR national research infrastructure; French Ministry of the Environment; MeteoSwiss (GAW-CH aerosol monitoring programme); the Swiss State



Secretariat for Education, Research and Innovation (SERI); Ministry of Education, Youth and Sports of CR within National Sustainability Program I (NPU I, grant no. LO1415); ERDF "ACTRISCZ RI" (grant no. CZ.02.1.01/0.0/0.0/16_013/0001315); CRISOL (CGL2017-85344-R MINECO/AEI/FEDER, UE); TIGAS-CM (Madrid Regional Government Y2018/EMT-5177); AIRTEC-CM (Madrid Regional Government P2018/EMT4329); REDMAAS2020 (RED2018-102594-T CIENCIA); Red de Excelencia ACTRIS-ESPAÑA (CGL2017-90884-REDT); the Spanish Ministry of Economy, Industry and Competitiveness; FEDER funds (project HOUSE, grant no. CGL2016-78594-R); the Generalitat de Catalunya (AGAUR 2017 SGR41 and the DGQA); the National Institute for Aerospace Technology; the Ministerio Español de Economía, Industria y Competitividad (MINECO); Spanish Ministry of Economy and Competitiveness (projects no. CGL2016-81092-R, CGL2017-90884-REDT, RTI2018-097864-B-I00 and PGC2018-098770-B-I00); Andalusia Regional Government (project no. P18-RT-3820); PANhellenic infrastructure for Atmospheric Composition and climate change (MIS 5021516); Research and Innovation Infrastructure; Competitiveness, Entrepreneurship and Innovation (grant no. NSRF 2014-2020); the Italian Ministry of Research and Education; the Norwegian Environment Agency; Swedish FORMAS; Swedish Research Council (VR); the Magnus Bergvall foundation; the Märta och Erik Holmberg foundation; the Swedish EPA.

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
