# Peer review of "Seasonality of the particle number concentration and size distribution: a global analysis retrieved from the network of Global Atmosphere Watch (GAW) near-surface observatories"

_Atmospheric Chemistry and Physics, 2020_

## Author Comment (AC1)

We would like to thank Referee #1 for their very positive feedback on the paper and their comments, which are addressed point by point below. Note that in addition to the marked edits, the paper has undergone English editing resulting in minor changes that are not systematically highlighted in the revised manuscript.

Comment 1: Page 5, lines 10: There have been many long-term trend analyses of optical properties, number concentration, and PNSD. Does this refer to analyses involving several sites? Please clarify in the text.

Reply 1: If the Reviewer refers to the studies by Asmi et al. (2013) and Collaud Coen et al. (2013), they in fact involve several sites. This is now explicitly mentioned in the text:
*"Moreover, Asmi et al. (2011) reported on the variability of the PNSD, also in Europe, based on measurements collected at 24 sites; shortly after, the first multi-site long-term trend analyses of aerosol optical properties (Collaud Coen et al., 2013), number concentration and PNSD (Asmi et al., 2013) were performed using measurements conducted at stations located in Europe, North America, Antarctica and on Pacific Ocean islands."*

Comment 2: Page 6, line 10: Why 2016 and 2017? Please explain in the text.

Reply 2: The present study is focussed on 2016 and 2017 for consistency with the analysis of Laj et al. (2020), which our work aims to complement. These specific years were selected as they provided the most updated view of measurements worldwide when the study of Laj et al. (2020) was initiated. The text was modified as follows:
*"Data collected at 62 sites contributing to SARGAN in 2017 or 2016, i.e. the reference years as chosen in Laj et al. (2020) (see more details about data availability and coverage criteria in Sect. 4), were included in the present work, among which 57 were already involved in the short analysis of the total number concentration reported in that study."*

Comment 3: Figure 1 and throughout: The symbols in the legend box should be larger.

Reply 3: The size of the symbols has been increased in the legends. The triangles in figures 3, S3 and S4 have also been slightly enlarged to ease the distinction between downward and upfacing triangles.

Comment 4: Figure 1 caption: It would be helpful if MPSS were spelled out.

Reply 4: The last sentence of the figure caption was changed to account for this suggestion:
*"The sites operating a mobility particle size spectrometer (MPSS) are additionally marked in italic bold."*

Comment 5: Page 6, line 18: What is meant by "(sub)-urban"? Does it include both suburban and urban? As far as I can tell neither it or "suburban" are used anywhere else in the paper.

Reply 5: In fact, the expression "(sub)-urban" was meant to refer to urban and suburban sites but, as they are all referred to as urban in the rest of the paper and also in Laj et al. (2020), this expression was removed to avoid confusion:
*"The stations are classified based on the combination of a geographical [...] and footprint (rural background, forest, urban, pristine or mixed) criteria as introduced in Laj et al. (2020)."*

Comment 6: Page 7, line 9: What is the definition of "large" particles as used here?

Reply 6: "large" refers here to particles that are above the measurement range of the instrument. The sentence was slightly change to clarify this point and two references were added:

*"As part of this quality control process, negative concentrations arising from inversion issues in certain conditions (e.g. presence of particles above the size range covered by the MPSS, such as dust or sea salt; Pfeifer et al., 2014; Wiedensohler et al., 2018) were filtered out"*.

Comment 7: Figure 10: It is difficult to tell the difference between the size of the symbols. Please change so that the size difference is clear. The yellow symbols are difficult to see. Can these be changed to another color?

Reply 7: To avoid any confusion, the different marker sizes have been replaced by different shapes (triangle and square), and yellow was changed to light orange to improve readability.

Please note that while adjusting the symbols, we realised that the size of the marker was not consistent between the two panels of Fig. 10. In Fig. 10.b, all correlations are in fact statistically significant at the 95% confidence level (instead of 90%); this has also been corrected in the text (P29, L25).

Comment 8: Also – up to here the coefficient of determination (r^2) has been used to describe the significance of the correlation between two parameters. Why is the Spearman's rank correlation coefficient being used here?

Reply 8: We used the coefficient of determination to evaluate the agreement between observed and fitted PNSD (i.e. same variable), following the same approach as Asmi et al. (2013). In the case of Fig. 10, as indicated P29, L4-5, the Spearman's rank correlation was used instead, similar to Collaud-Coen et al. (2018), as we evaluate the correlation between two different variables (particle concentration and elevation of the site or ABL TopoIndex) which are not necessarily normally distributed, and for which we cannot anticipate a linear relationship.

Comment 9: Page 40, line 5: change to "RESIDENTIAL"

Reply 9: Thank you for noticing the typo!

---

## Author Comment (AC2)

We would like to thank Referee #2 for their comments, which are addressed point by point below. Note that in addition to the marked edits, the paper has undergone English editing resulting in minor changes that are not systematically highlighted in the revised manuscript.

**General comment:**
My apologies to all for the tardy review. I view the data analyses in this paper as strong, but some of the interpretations and conclusions derived from the analysis are not nearly so rigorous as the methods. The introduction is nicely written, and the analysis of the fraction of available data that is important to adequately represent the statistics appears to be very well done. At about 31 pages of text and 13 figures, the paper is too long.

Section 5.2 is a series of cursory discussions about what the variations in Ntot may represent. The discussion here does not offer a lot, and some of the inferences concerning POLAR sites are a little misleading. I t might be helpful if the authors told us how these data should be used to help global models: e.g., which plots might you think the most useful for model evaluation?

Reply:
The paper is indeed relatively long, but we believe that this length is justified both by the number of sites considered (62) and by the number of variables analysed; furthermore, as the reviewer points out, this paper includes, as part of the statistical approach that is adopted, a section dedicated to an original analysis of the impact of data availability.

Section 5.2 is in fact very descriptive but presents "numbers" that we believe are useful to the modelling community, both in terms of particle concentrations (Table S2) and the parameters that describe log-normal aerosol number size distributions (Table A1); this information is complemented by the data in Figures 12 and 13 and Table 2 (and Figures S7 and S8 and Table S3) in Section 7 with respect to the study of the potential CCN population. The files made available by Rose et al. (2021) provides in addition qualified time series of the particle concentration in the size ranges of interest ($N_{tot}$, $N_{50}$ and $N_{100}$) as well as the seasonal medians of the particle number size distributions when available. This is now clearly mentioned at the end of Sect. 5.1:

"*In addition to Tables A1 and S2 (as well as Figs 12-13 and S7-S8 and Tables 2 and S3 discussed in Sect. 7), which provide ready-to-use information for the modelling community on both particle number concentrations in the size ranges of interest for this work and the parameters used to describe the PNSD, qualified time series of $N_{tot}$ (as well as $N_{50}$ and $N_{100}$) and seasonal medians of the PNSD are available in Rose et al. (2021).*"

More broadly, Section 5.2 provides a perspective on the observations of a large number of sites with different characteristics, allowing differences and similarities to emerge that are otherwise difficult to identify in individual site studies, or in studies focused on a particular site type. Processing of data from a significant number of stations belonging to the same "class" can also provide a more complete picture of these sites, such as for instance high altitude sites, whose representation is particularly complex in models. With respect to polar sites, we hope that the changes and clarifications made in response to the detailed comments will clarify the corresponding observations.

**Detailed comments:**

Comment 1: Page 2, lines 43-44 – It is unclear whether multiple transformations include wet and dry deposition. Explicitly mention deposition.

Reply 1: As suggested, deposition is now explicitly mentioned: *"The variety of their sources, as well as the multiple transformations they may undergo during their transport (including wet and dry deposition), result in significant spatial and temporal variability of their properties."*

Comment 2: Page 3, line 10 – maybe "less evident" rather than "barely marked".

Reply 2: Change was made.

Comment 3: Page 3, line 18 – ">50-100 nm" is unclear. Instead, maybe "(considered here as either >50 nm or >100 nm)".

Reply 3: The suggestion was taken into account.

Comment 4: Page 5, lines 17-20 – The wording here suggests that such closure studies have never been conducted until recently, which is not the case. Please add something like "at the above-mentioned network sites". The ACTRIS reference relates to the Schmale study, not this generalized statement.

Reply 4: Precision was added to avoid any confusion:
*"The monitoring of an increasing number of variables […] and to carry out closure studies at the above mentioned network sites, such as that performed by Schmale et al. (2017, 2018) […] from 12 ACTRIS sites"*.

Comment 5: Page 5, lines 31-33 – It is very disappointing to see no PSND data from Alert and no data at all from the Whistler Mountain site (both of which have many related publications) not included in the "most up-to-date information … worldwide…".

Reply 5: As mentioned in Sect. 2 (P6, L29-31), this study includes stations with data available in the EBAS/World Data Centre for Aerosol database (WDCA), which was unfortunately not the case for PNSD data from the above mentioned sites or for $N_{tot}$ at WHI. An exception was made for WGG as it was the only station in Africa and for WLG, because of the relatively low coverage in Asia (and because political constraints in China do not allow for submission of the WLG data to EBAS), but otherwise this criterion was scrupulously applied for regions of the globe where the measurement network (regardless of the type of instrument used) was denser, such as Europe and North America. This is a strategy that we need to enforce to make sure all information is made accessible through the adequate repositories, in that case EBAS/WDCA. We have made a clarification in Sect. 2: "*With the exception of WGG and WLG*, hourly means of the particle number concentration and/or PNSD are available for all these sites on the database EBAS..."

Comment 6: Figure 1a – "Pristine" is defined as "belonging to the earliest period or state", "not spoiled, corrupted, or polluted (as by civilization)", "fresh and clean as or as if new". In terms of the aerosol, this definition means that a site is pristine if it is not influenced by anthropogenic sources, rather only by natural sources. Parts of the Arctic can reach a pristine condition during the summer, but it is incorrect to broadly represent the Arctic stations as pristine. Arctic Haze is not a pristine condition. The general use of 'pristine' to describe the polar sites is unjustified.

Reply 6: It is true that the conditions encountered at Arctic sites are not always pristine in the strict sense of the definition recalled by the reviewer, but we believe that, overall, these sites provide a picture of environments that can certainly be considered pristine in comparison to more anthropogenically influenced regions. Furthermore, this nomenclature has been used and accepted in recent publications related to the SARGAN initiative that involve some or all of the sites discussed in this study (e.g., Collaud Coen et al., 2020; Laj et al., 2020). Clarification on the use of the term "pristine" in this work has been added in Sect. 2:
 "*Regarding the "pristine" class, it includes stations that sample background air in comparison to more anthropically influenced locations, but this classification does not imply, however, that these sites are completely free of anthropogenic interference. In particular, while the Arctic is a pristine region from an aerosol source perspective, anthropogenic influence through long-range transport can be substantial (particularly during winter and spring when the polar vortex extends and includes more polluted area; e.g. Abbatt et al., 2019 and references therein)*".

Comment 7: Section 3.2 – A. You state that you fit two modes to the particles between 20 nm and 500 nm, and the first mode is comprised of Aitken particles (maybe nucleation particles, depending on one's definition). Is there a typical size range for the minimum between the two modes, and can this minimum

be related in some cases to cloud processing (i.e., Hoppel minimum)? B. Also, modes of some urban and rural distributions can be singular. Is that allowed in your analyses? C. Lastly, is 100 nm your definition of the split between Aitken and accumulation, and would you be explicit about that, please.

Reply 7:
A.    The minimum between the two modes has not been specifically studied. It can however be expected that the Hoppel minimum will vary depending on the supersaturation experienced by the air mass reaching a certain site. So, yes, this aspect would be highly interesting but is beyond the scope of this work and would further increase the paper length. Rather, this would deserve a paper on its own.
B.    The analysis was conducted in such a way that 2 modes are systematically searched for. This approach was motivated by the aim of applying the same analysis approach to all the sites, and further supported by the work of Asmi et al. (2011), who show that the use of 2 modes often provides a better description of the distributions than 1 mode. When a distribution has a strong monomodal feature, this translates into a significant domination of one mode over the other.
C.    There is no prescribed diameter for the distinction between the two modes. The fitting process looks for two modes whose characteristic diameters are between 20 and 500 nm; specific values (50 nm for the first mode and 130 nm for the second) are only used to initialise the procedure (conducted with the Matlab fit function). The fitting procedure is mentioned in Sect. 3.2 and the fact that the 2 modes are derived from the fitting procedure is now recalled in Sect 5.1:
*"Median distributions and corresponding modes (defined through the separation by the bimodal fit procedure, see Sect. 3.2) are shown for each site and season…".*

Comment 8: Section 3.3 – Two points are missing here: 1) CCN number concentrations do not necessarily translate to cloud droplet number concentrations, particularly at higher concentrations, due to competition for water vapour by droplets growing at cloud base; 2) particles at 50 nm can only be effective if the cloud base supersaturation reaches a sufficiently high value. The latter happens in one of two ways: relatively few particles larger than 200 nm, such that the rate of water uptake by the >200 nm particles does not dominate the total uptake; very rapid cooling rates, typically associated with relatively extreme updraft speeds.

Reply 8: The role of surrounding atmospheric conditions on the droplet activation process, including in particular cloud supersaturation, is already acknowledged in Sect. 3.3. We note that also updraft velocity plays an important role. Additional elements clarifying the distinction between CCN and cloud droplet number concentrations have been added at the end of the section:
*"It should be noted that estimates of CCN number concentrations based on particle characteristics such as size do not necessarily translate to cloud drop number (CDN) concentrations. Other factors also need to be considered, such as updraft velocity or if there is competition for water vapor due to high CCN concentrations or, as alluded to above, whether supersaturation values reach a sufficiently high value to enable the CCN to CDN transformation. Analysis of this last process and evaluation of CDN concentrations are, however, beyond the scope of the present work."*
In addition we added the following text in Sect. 3.3:
*"This threshold diameter was close to the overall median activation diameter (87 nm) reported by Hammer et al. (2014) for an approximate cloud supersaturation of 0.35%, although the activation diameter at this site was occasionally as low as 40 nm with a supersaturation of 0.86% (Motos et al., 2019)."*

Comment 9: Page 16, lines 18-30 – A. There are others in Figure 2 that appear to fall into the category of 95% or more: ZEP and ALT, LLN, WGG. Why were they not used? B. Also, I find it odd that data coverage can be essentially 100%, as it appears to be for TRL and NMY. Do these sites not perform checks on the operation of the instrument, zeroes, etc.? C. The filtering for wind direction, referred to for SPO, BRW and MLO, also happens for the ALT data, and it has been shown that that such filtering of the Alert data has little impact on the observations at Alert, except for a near-negligible impact on Aitken-size particles.

Reply 9:
A.      Data availability is reported for all the stations in Fig. S5 in the Supplement; the above mentioned sites (ZEP, ALT, LLN and WGG), although they have significant data coverage (between 74 and 93%), do not reach the 95% threshold.

B.      The exact data availability is 99.7% and 99.8% for NMY and TRL, respectively, in 2017; for simplicity the values presented in Fig. S5 are rounded. Checks are indeed performed at these stations, both on the instrumental setup and the measurements collected (see for instance Weller et al., 2011 for more details about the procedures applied at NMY), but when these checks result in missing or excluded data for periods shorter than 1 h, there is not necessarily an impact on the availability of the data calculated from the hourly averages that is reported in this study.

C.      Indeed, there is an automatic wind-based filter applied to Alert data, but this information (which does not appear in the metadata available on EBAS), had not been communicated to us. We have therefore corrected the sentence accordingly:

"*As mentioned previously, such filtering occurs at SPO, BRW, ALT and MLO; for these four stations, the coverage criteria discussed here were not applied.*"

We have also added a more general comment on the data quality control procedure applied on the various sites at the end of Section 2:

"*To ensure the quality of the analysis, only the data marked as valid were used, similar to Asmi et al. (2011). It is important to note, however, that despite the procedures that are being implemented within the research networks with the aim of achieving consistency in data from different sites, different data submitters may flag their data differently both because of their scientific use of the data and the tools at their disposal. For instance, a very strict and automated wind screening criterion is applied at some stations (SPO, BRW, ALT and MLO), with an impact on data coverage, while for other sites such as NMY, data from a given wind sector are flagged but considered valid when there is no further indication for contamination (e.g. from concurrent black carbon measurements). Additional checking of the data was performed in collaboration with each instrument's principal investigator to ensure the homogeneity of the dataset, but we cannot exclude the possibility that specific treatment of the data applied at some sites (but not specified in the metadata available on EBAS) may have not been reported. As part of this quality control process, negative concentrations…*"

Comment 10: Page 17, line 8 - I tend to agree that 4-5 weeks shows little impact, but visually there is not much difference between the medians at 12 weeks and those at 20. The choice here of 12 weeks here appears to be more a matter of convenience than rigor.  How do you justify 12?

Reply 10: The authors do agree that the effects of removing 12 to 20 weeks leads to a continuous dispersion of the ratio between the medians. Focus on this specific gap length (i.e., 12 weeks) is explained by the fact that some ratios between the original and the corrected medians are larger than 1.5 for the removal of periods longer than 12 weeks. The manuscript clearly indicated by the symbol "~" that 12 weeks is an approximated threshold allowing an easy description of the results. This threshold is moreover not further used to discriminate results.

Comment 11: Page 18, line 10 - PAL and VAR are both close to the Arctic Circle and relatively close to each other. Alert is much more remote, offers another perspective and appears to qualify, based on Figure 2.  Why was it not chosen?

Reply 11: As mentioned already in Reply 9.A, data availability at Alert (84%) was below the 95% threshold used in this part of the analysis.

Comment 12: Page 19, lines 7-10 – Although I find the diel cycle analysis interesting, I don't really see how it helps global models. Somewhere in the text, can you offer some explanation for the importance of this on a global scale?

Reply 12: Dealing with diel variability is one of the difficulties in global models, therefore we believe that such analysis can contribute to the evaluation of model performances with respect to these aspects,

and that it may be of interest for regional or process level modellers as well. Getting variability on different time scales right in the models can be a quality indicator or even need for addressing certain research questions.

Comment 13: Page 23, line 5 – Should 6.a be 6.c?

Reply 13: Reported numbers seem correct:

- analysis of the seasonal contrast of $N_{tot}$ is reported in Fig. 6c, with, specifically, the values of SeasC shown in the left panel and indication on the seasons used for the determination of SeasC in the right panel;
- annual medians and quartiles of $N_{tot}$ are reported in Fig. 6.a.

Comment 14: Page 24, line 12 – The use of "generally" and "around" to characterize something so precisely defined as 42+/-14 seem a little out of place.

Reply 14: The above mentioned expressions have been removed from the referred sentence, which now reads as:
*"The corresponding PNSD are characterized by an Aitken mode located at 42±14 nm and an accumulation mode found, on average, at 149±37 nm (Table A1 and Fig. 9)."*

Comment 15: Page 24, lines 17-18 – Freud et al. (ACP, 2017) should be referenced in connection with this statement.

Reply 15: The abovementioned reference was added:

*"Enhanced concentrations of Aitken mode particles coinciding with the maximum of $N_{tot}$ during local summer more specifically appear as a common feature of the four polar sites equipped with a MPSS (Table A1 and Fig. 8). This is consistent with findings by Freud et al. (2017) for size distributions measured at five Arctic sites."*

Comment 16: Page 24, lines 19-20 – The Arctic and Antarctic are quite different in some respects, most notably the strong influence of southern-latitude anthropogenic sources on the Arctic in the winter and spring, also known as Arctic Haze. Arctic Haze is significant, and the number concentrations of accumulation-mode particles can extend through significant depths of the Arctic troposphere. Your generalized statement, based on measurements at the Antarctic, should not include the Arctic.

Reply 16: The referred statement is based on Figure 6.a and Table S2 which show lower $N_{tot}$ for most of the sites classified as polar compared to other station types, and, further, decreasing concentrations with increasing latitudes among polar sites, with, for instance, the lowest annual medians occurring for 3 Arctic sites (BRW, ALT, ZEP). It seems reasonable to attribute this to less anthropogenic influence. We otherwise agree with the reviewer that the impact of Arctic Haze can be seasonally significant in the Arctic and briefly discuss seasonality further down in this section.

Comment 17: Page 24, lines 28-29 – I don't understand the statement beginning "Larger diameters ... during JJA". One of the reasons that Ntot is higher at Arctic sites in the summer (JJA) is because the accumulation mode concentrations are much lower, which helps NPF (e.g., Freud et al., ACP, 2017; Croft et al., ACP, 2016; Leaitch et al., Elementa, 2013). This statement appears to say the opposite.

Reply 17: It seems that there is a misunderstanding, since the message we wanted to convey is in fact in perfect agreement with what is indicated by the Reviewer. We have therefore simplified the sentence in question to avoid any confusion, and we have added elements related to these aspects at the end of the section:

*"Larger diameters are more specifically seen for both modes during MAM at ZEP, and later during JJA at PAL and VAR, while the modes' diameters are, in contrast, almost constant at TRL. [...]*
*This is for instance the case during the pervasive annual episodes of Arctic haze observed across much of the region, which causes elevated number concentrations of accumulation mode particles during springtime in the Arctic region (Abbatt et al., 2019 and references therein), as reflected in the measurements collected at ZEP during this time of the year (Fig. A1). As reported in earlier studies (e.g. Croft et al., 2016; Freud et al., 2017), this likely affects NPF over this region, where the maximum frequency of occurrence of the process is observed during JJA, when the existing aerosol surface area is reduced, while this maximum is, in contrast, seen earlier during spring at sub-Arctic sites (Nieminen et al., 2018).*"

Comment 18: Page 24, lines 32-33 – "Transport…" – Yes, but for accumulation-mode particles the transport in Croft et al is accompanied by stronger wet deposition, which significantly reduces those particles.

Reply 18: The referred sentence was changed to *"Transport was for instance reported as an important source of Aitken and accumulation mode particles during summer at Arctic sites such as ZEP and ALT, while the accompanying wet deposition reduced the number concentration of accumulation mode particles (Croft et al., 2016)".*

Comment 19: Page 25, lines 6-8 - Arctic Haze, which has been studied for over 50 years, is pervasive during the winter and spring in the high Arctic, largely because of the reduced deposition of these particles in the Arctic during those times of the year. Your statement makes Arctic Haze sound like an occasional factor, which it is not.

Reply 19: In order to meet the reviewer's objection, the sentence concerned has been amended as follows:
*"Finally, some specific phenomena have also been previously reported to affect the PNSD. This is for instance the case during the pervasive annual episodes of Arctic haze observed across much of the region, which causes elevated number concentrations of accumulation mode particles during springtime."*

Comment 20: Page 28, lines 13-14 – It may be fair to say that the higher Ntot at mountain sites during summer are similar to polar sites, but the reasons may be (and likely are) very different, and this should be noted.

Reply 20: This is now specified explicitly to avoid any confusion:
*"Similar to polar sites (although for different reasons, as discussed later in this section), higher $N_{tot}$ are mostly found during local summer (6 sites)…".*

Comment 21: Page 29, line 14 – remove "instead"

Reply 21: The sentence was rephrased to:
*"... ; in contrast, $N_{tot}$ values are more representative of free tropospheric air masses and long range transport during winter…"*

Comment 22: Page 30, lines 1-2 - Can't the lower correlation be related to the fact that the TopoIndex is a calculated quantity, whereas altitude has far less uncertainty?

Reply 22: Indeed, the altitude has clearly a much lower uncertainty than the ABL-TopoIndex, which contains in contrast a higher density of information, but we believe this is not the main reason for the stronger correlation with altitude during summer. The ABL-TopoIndex corresponds (by its construction) to a probability for the ABL influence at high altitude sites. The aerosol concentration, optical properties and chemical composition at these stations will then depend not only on the degree of ABL influence

but also on the aerosol concentration and properties in the ABL, which are not included into the ABL-TopoIndex. For that purpose, inventories of aerosol sources should also be taken into account. The higher correlation between $N_{tot}$ and altitude can simply correspond to the standard decrease of all components with altitude and is also likely related to the dilution of aerosol throughout the complete ABL depth. These aspects are now clarified in the revised version of the manuscript:

"*We hypothesize that the absence of aerosol source inventories in the ABL-TopoIndex explains the lower Spearman rank correlation with $N_{tot}$ during maximal ABL influence in summer whereas the standard decrease of aerosol with altitude and ABL depth has a strong impact on the connection between $N_{tot}$ and altitude*."

Comment 23: Page 37, lines 9-11 – You suggest here that the diurnal variation at polar sites during the transition seasons is responsible for NPF events, but Nieminen clearly shows that the frequency of observation of NPF is predominantly a summertime phenomenon in the Arctic. This needs correction.

Reply 23: The Reviewer's interpretation does not correspond to the idea we originally intended to convey, so we have changed the referred sentence to try to clarify the message: "*As shown in Fig. 11.b, $D_{cy}$ values are in fact mainly reported during the transition seasons, when there is a day/night distinction; although they are less frequent than in summer, NPF events are also observed during this time of the year (e.g. Nieminen et al., 2018) which can contribute to the identified cycles.*"

Comment 24: Page 39, lines 5-6 – You need to note here that for 50-80 nm particles to be effective at nucleating cloud droplets, the N100 must be significantly lower. Competition for water vapour among the larger (and soluble) particles will not permit supersaturations to reach sufficient values to activate 50-80 nm particles unless the larger particle concentrations are quite low.

Reply 24: As mentioned already in the answer to Comment 8 and now clarified in the revised manuscript (Sect. 3.3), the present work aims at estimating the concentration of particles likely to act as CCN based on a simple approach. A study of the actual CDN concentration would not only include the number concentration of activated particles but also the updraft velocity and is beyond the scope of this work.

Comment 25: Page 39, lines 27-28 – It would be interesting to see how Ntot-N100 and N100 compare, and if there are locations/situations where they exhibit some degree of mutual exclusivity or correlate.

Reply 25: The study of N50 and N100 and the corresponding Nx/Ntot ratios was intended to provide a simple analysis of the population of potential CCN; in this context, we believe that additional analysis of other combinations of these quantities would be of limited interest, and would furthermore contribute to lengthening the paper which, as the Reviewer points out, is already relatively long.

Comment 26: Page 39, line 34 – interstitial spelling.

Reply 26: Correction was made!

Comment 27: Page 40, lines 1-3 – The beginning of this sentence seems to be missing something. Do you mean "in contrast to those observed"? Also, here you mention "when clouds are less prevalent and transport is most favoured", referring again to Croft. As above, Croft does indicate higher transport of accumulation-mode particles, but they also show substantially increased wet deposition. This sentence should be re-considered.

Reply 27: The referred statement was rephrased/completed as follows:
"*In contrast, the highest ratios between $N_{100}$ and $N_{tot}$ are observed during summer at these sites, when clouds are less prevalent and the transport (in connection to ABL dynamics at mountain sites) of CCN-sized particles is the most favoured (e.g. Croft et al., 2016; Herrmann et al., 2015). We cannot exclude, however, the possibility, that efficient wet deposition reported to reduce accumulation mode particles*"

*at some Arctic sites (Croft et al., 2016) could lead to observations specific to these sites, possibly contrasting with the average picture shown in Fig. 13.a.”*

Comment 28: Page 40, lines 14-15 – "additional source" relative to what?

Reply 28: The sentence has been slightly modified to clarify the message:
"*The winter maximum of the ratio between $N_{50}$ and $N_{tot}$ is however less marked than in the case of $N_{100}$ at lowland sites other than polar, supporting the existence of an additional source of particles larger than 100 nm in winter compared to other seasons at these stations*".

Comment 29: Page 40, lines 19-22 - Here you mention "contribution", but a slope is not necessarily a contribution. In Figure 13, you show the true contributions of N100 to Ntot to be typically less than 30%. Please explain why the contributions are so much higher here.

Reply 29: That is true, the slope does not translate to the value of the contribution of $N_{100}$ to $N_{tot}$, which is otherwise shown in the inserts of Fig. 13, but, as stated in the referred statement, the fact that the slope is close to 1 does indicate, however, that the ratio $N_{100}/N_{tot}$ is relatively constant over the whole range of $N_{tot}$. Indeed, if $\log(N_{100}) = a*\log(N_{tot}) + b$ (which is the form of the fit equation in Fig. 13) and a~1, then $N_{100}/N_{tot} \sim 10^b$, which is a constant.

Comment 30: Page 40, lines 26-27 – I don't understand how the summer and winter "behaviour"s are close; perhaps 'behaviour' is the problem here. In winter, the size distributions are for particles mostly larger than 80 nm. In summer the size distributions are skewed towards smaller particles, e.g., see Freud et al. (ACP, 2017).

Reply 30: We state that summer and winter behaviours are close at polar sites in that, in both seasons, the slope of the fit is close to 1, and therefore the contribution of $N_{100}$ to $N_{tot}$ is, in each case, constant over the whole range of Ntot. This does not imply, however, that the distributions are identical and/or that the contributions are equal across the seasons. The sentence was slightly rephrased to avoid any confusion:
"*The fits obtained for polar stations in summer have slopes approaching 1 (0.91 for $N_{100}$ and 0.99 for $N_{50}$), indicating rather constant contributions of $N_{100}$ and $N_{50}$ to $N_{tot}$ over the whole range of $N_{tot}$ in this season as well. This is also the case for mountain sites…*".

Comment 31: Page 40-41, lines 34-4 – You overlook the issue with the impact of particle concentrations on cloud supersaturation, which is just as important as the numbers of CCN.

Reply 31: As mentioned already in the answers to comments 8 and 24, the present work aims at evaluating the population of potential CCN based on a simple analysis of the PNSD. The point which is raised by the reviewer concerns the ability of these potential CCN to further activate into cloud droplets, which is beyond the scope of this analysis. Still, clarification was made at the end of Sect. 7 : "*... a more precise analysis would require information on other parameters that impact the activation diameter, such as the hygroscopicity of the sampled particles for each site, which probably varies seasonally according to the nature of the particles, the total number of activated particles, which reduces the supersaturation, as well as the updraft velocity (Schmale et al., 2018).*"

Comment 32: Page 44, lines 4-6 - Figure 5a indicates relatively uniform global coverage, although, as you mention, Africa, Russia, South America and oceans are clearly under-represented. Do you need all 39 European sites? Why not introduce a bit more global balance by conducting the analysis using only the sites in Figure 5a plus a smaller selection of European sites?

Reply 32: We believe that it was of particular interest for this study, which aims to provide as complete a picture as possible of particle number concentrations worldwide, to include as many sites as possible (within the selection criteria). Furthermore, the criteria that could have been used for the selection of

some European sites are not obvious, and certainly arbitrary. Also, this type of selection seems less justified in the context of this work as the results are not discussed at the scale of geographical areas (with the exception of polar sites) but according to the types of sites, independent of their location. However, as we cannot eliminate the possibility that inhomogeneities in the dataset (e.g., site location, station type and instruments in use) might affect our observations, these constraints are clearly stated at the beginning of the analysis (P6, L12-25), and, for some of them, recalled in the result sections (e.g. P25, L13, all urban stations included in the analysis are located in Europe).

Comment 33: Page 44, lines 22-23 – It may be true for some of the forest sites, mountain sites and polar sites, but it does not appear to be true for the urban or rural background sites.

Reply 33: It is true that higher concentrations are not systematically observed in spring and summer but as illustrated in Fig. 6 and discussed in Sect. 5.2, this is *often* (as in turn indicated in the referred statement) the case, including at urban (6 sites out of 9 shown in Fig. 6.c) and rural background (10 sites out of 12 shown in Fig. 6.c) sites.

Comment 34: Page 45, lines 2-3 – Based on section 3.2, it seems as if your analysis essentially dictated the bimodality. Is that false?

Reply 34: As explained in the answers to parts B. and C. of Comment 7, it is true that the fitting procedure used to study the modes representative of the distributions consists in systematically identifying 2 modes; in the case of distributions having a rather monomodal character, however, one of the two modes makes only a small contribution. Focussing specifically on polar sites, we believe that a simple visual inspection of the distributions shown in Fig. A1 allows us to conclude that most of them have a marked bimodal character, indicating that the reported bimodality is not just dictated by the fitting procedure.

Comment 35: Page 46, line 10 – Remove "Slight".

Reply 35: Removed.

**References :**

Freud, E., Krejci, R., Tunved, P., Leaitch, R., Nguyen, Q. T., Massling, A., Skov, H., and Barrie, L.: Pan-Arctic aerosol number size distributions: seasonality and transport patterns, Atmos. Chem. Phys., 17, 8101–8128, https://doi.org/10.5194/acp-17-8101-2017, 2017.

Motos, G., Schmale, J., Corbin, J. C., Modini, Rob. L., Karlen, N., Bertò, M., Baltensperger, U., and Gysel-Beer, M.: Cloud droplet activation properties and scavenged fraction of black carbon in liquid-phase clouds at the high-alpine research station Jungfraujoch (3580 m a.s.l.), Atmos. Chem. Phys., 19, 3833–3855, https://doi.org/10.5194/acp-19-3833-2019, 2019.

Rose, C., Collaud Coen, M., Andrews, E., Lin, Y., Bossert, I., Lund Myhre, C., Tuch, T., Wiedensohler, A., Fiebig, M., Aalto, P., Alastuey, A., Alonso-Blanco, E., Andrade, M., Artíñano, B., Arsov, T., Baltensperger, U., Bastian, S., Bath, O., Beukes, J. P., Brem, B. T., Bukowiecki, N., Casquero-Vera, J. A., Conil, S., Eleftheriadis, K., Favez, O., Flentje, H., Gini, M. I., Gómez-Moreno, F. J., Gysel-Beer, M., Hallar, A. G., Kalapov, I., Kalivitis, N., Kasper-Giebl, A., Keywood, M., Kim, J. E., Kim, S.-W., Kristensson, A., Kulmala, M., Lihavainen, H., Lin, N.-H., Lyamani, H., Marinoni, A., Martins Dos Santos, S., Mayol-Bracero, O. L., Meinhardt, F., Merkel, M., Metzger, J.-M., Mihalopoulos, N., Ondracek, J., Pandolfi, M., Pérez, N., Petäjä, T., Petit, J.-E., Picard, D., Pichon, J.-M., Pont, V., Putaud, J.-P., Reisen, F., Sellegri, K., Sharma, S., Schauer, G., Sheridan, P., Sherman, J. P., Schwerin, A., Sohmer, R., Sorribas, M., Sun, J., Tulet, P., Vakkari, V., van Zyl, P. G., Velarde, F., Villani, P., Vratolis,

S., Wagner, Z., Wang, S.-H., Weinhold, K., Weller, R., Yela, M., Zdimal, V., and Laj, P.: Seasonality of the particle number concentration and size distribution: a global analysis retrieved from the network of Global Atmosphere Watch (GAW) near-surface observatories, ACTRIS Data Centre, https://doi.org/10.21336/gen.sg1y-ay21, 2021.

Weller, R., Minikin, A., Wagenbach, D., and Dreiling, V.: Characterization of the inter-annual, seasonal, and diurnal variations of condensation particle concentrations at Neumayer, Antarctica, Atmos. Chem. Phys., 11, 13243–13257, https://doi.org/10.5194/acp-11-13243-2011, 2011.